# Nonparametric Evaluation of Noisy ICA Solutions

**Syamantak Kumar**[1]     **Purnamrita Sarkar**[2]     **Peter Bickel**[3]     **Derek Bean**[4]

[1]Department of Computer Science, UT Austin
[2]Department of Statistics and Data Sciences, UT Austin
[3]Department of Statistics, University of California, Berkeley
[4] Department of Statistics, University of Wisconsin, Madison
syamantak@utexas.edu, purna.sarkar@austin.utexas.edu,
bickel@stat.berkeley.edu, derekb@stat.wisc.edu

## Abstract

Independent Component Analysis (ICA) was introduced in the 1980's as a model for Blind Source Separation (BSS), which refers to the process of recovering the sources underlying a mixture of signals, with little knowledge about the source signals or the mixing process. While there are many sophisticated algorithms for estimation, different methods have different shortcomings. In this paper, we develop a nonparametric score to adaptively pick the right algorithm for ICA with arbitrary Gaussian noise. The novelty of this score stems from the fact that it just assumes a finite second moment of the data and uses the characteristic function to evaluate the quality of the estimated mixing matrix without any knowledge of the parameters of the noise distribution. In addition, we propose some new contrast functions and algorithms that enjoy the same fast computability as existing algorithms like FASTICA and JADE but work in domains where the former may fail. While these also may have weaknesses, our proposed diagnostic, as shown by our simulations, can remedy them. Finally, we propose a theoretical framework to analyze the local and global convergence properties of our algorithms.

## 1 Introduction

Independent Component Analysis (ICA) was introduced in the 1980's as a model for Blind Source Separation (BSS) [13, 12], which refers to the process of recovering the sources underlying a mixture of signals, with little knowledge about the source signals or the mixing process. It has become a powerful alternative to the Gaussian model, leading to PCA in factor analysis. A good account of the history, many results, and the role of dimensionality in noiseless ICA can be found in [28, 5]. In one of its first forms, ICA is based on observing $n$ independent samples from a $k$-dimensional population:

$$\boldsymbol{x} = B\boldsymbol{z} + \boldsymbol{g} \tag{1}$$

where $B \in \mathbb{R}^{d \times k}$ $(d \geq k)$ is an unknown mixing matrix, $\boldsymbol{z} \in \mathbb{R}^k$ is a vector of statistically independent mean zero "sources" or "factors" and $\boldsymbol{g} \sim N(\boldsymbol{0}, \Sigma)$ is a $d$ dimensional mean zero Gaussian noise vector with covariance matrix $\Sigma$, independent of $\boldsymbol{z}$. We denote $\mathrm{diag}\,(\boldsymbol{a}) := \mathrm{cov}\,(\boldsymbol{z})$ and $S := \mathrm{cov}\,(\boldsymbol{x}) = B\,\mathrm{diag}\,(\boldsymbol{a})\,B^T + \Sigma$. The goal of the problem is to estimate $B$.
ICA was initially developed in the context of $\boldsymbol{g} = 0$, now called noiseless ICA. In that form, it spawned an enormous literature [17, 39, 40, 1, 8, 27, 35, 37, 23, 11, 6, 42] and at least two well-established algorithms FASTICA [29] and JADE [9] which are widely used. The general model with nonzero $\boldsymbol{g}$ and unknown noise covariance matrix $\Sigma$, known as *noisy* ICA, has developed more slowly, with its own literature [4, 50, 49, 30, 31, 41].
In the following sections, we will show that various ICA and noisy ICA methods have distinct shortcomings. Some struggle with heavy-tailed distributions and outliers, while others require approximations of entropy-based objectives, which have their own challenges (see eg. [32]).

38th Conference on Neural Information Processing Systems (NeurIPS 2024).

Although methods for noiseless cases may sometimes work in noisy settings [49], they are not always reliable (e.g., see Figure 1 in [49] and Theorem 2 in [11]). Despite the plethora of algorithms for noisy and noiseless ICA, the literature has largely been missing a diagnostic to decide which algorithm to pick for a given dataset. This is our primary goal.

The paper is organized as follows. Section 2 contains background and notation. Section 3.1 contains the independence score, its properties, and a Meta algorithm 1 that utilizes this score. Section 3.3 introduces new contrast functions based on the natural logarithm of the characteristic function and moment generating function of $\boldsymbol{x}^T\boldsymbol{u}$. Section 4 presents the global and local convergence results for noisy ICA. Section 5 demonstrates the empirical performance of our new contrast functions as well as the Meta algorithm applied to a variety of methods for inferring $B$.

## 2 Background and overview

This section contains a general overview of the main concepts of ICA.

**Notation:** We denote vectors using bold font as $\boldsymbol{v} \in \mathbb{R}^d$. The dataset is represented as $\left\{\boldsymbol{x}^{(i)}\right\}_{i \in [n]}$ for $\boldsymbol{x}^{(i)} \in \mathbb{R}^d$. Scalar random variables and matrices are represented using upper case letters, respectively. $\boldsymbol{I}_k$ denotes the $k$- dimensional identity matrix. Entries of vector $\boldsymbol{v}$ (matrix $M$) are represented as $v_i$ ($M_{ij}$). $M(:,i)$ ($M(i,:)$) represents the $i^{\text{th}}$ column (row) of matrix $M$. $M^\dagger$ represents the Moore–Penrose inverse of $M$. $\|.\|$ represents the Euclidean $l_2$ norm for vectors and operator norm for matrices. $\|.\|_F$ similarly represents the Frobenius norm for matrices. $\mathbb{E}[.]$ ($\widehat{\mathbb{E}}[.]$) denotes the expectation (empirical average); $\boldsymbol{x} \perp\!\!\!\perp \boldsymbol{y}$ denotes statistical independence. $\|.\|_{\psi_2}$ represents the subgaussian Orlicz norm (see Def. 2.5.6 in [46]). $X_n = O_P(a_n)$ implies that $\frac{X_n}{a_n}$ is stochastically bounded i.e, $\forall \epsilon > 0$, there exists a finite $M$ such that $\sup_n \mathbb{P}\left(\left|\frac{X_n}{a_n}\right| > M\right) \le \epsilon$ (See [45]).

**Identifiability:** One key issue in Eq 1 is the identifiability of $A := B^{-1}$[1], which holds up to permutation and scaling of the coordinates of $\boldsymbol{z}$ if, at most, one component of $\boldsymbol{z}$ is Gaussian (see [14, 43]). If $d = k$ and $\Sigma$ are unknown, it is possible, as we shall see later, to identify the canonical versions of $B$ and $\Sigma$. For $d > k$ and unknown $\Sigma$, it is possible (see [15]) to reduce to the $d = k$ case. In this work, we assume $d = k$. For classical factor models, when $\boldsymbol{z}$ is Gaussian, identifiability can only be identified up to rotation and scale, leading to non-interpretability. This suggests fitting strategies focusing on recovering $B^{-1}$ through nongaussianity as well as independence. In the noisy model (Eq 1), an additional difficulty is that recovery of the source signals becomes impossible even if $B$ is known exactly. This is because, the ICA models $\mathbf{x} = B(\mathbf{z} + \mathbf{r}) + \mathbf{g}$ and $\mathbf{x} = B\mathbf{z} + (B\mathbf{r} + \mathbf{g})$ are indistinguishable (see [48] pages 2-3). This is resolved in [48] via maximizing the Signal-to-Interference-plus-Noise ratio (SINR). In classical work on ICA, a large class of algorithms (see [37]) choose to optimize a measure of non-gaussianity, referred to as contrast functions.

**Contrast functions:** Methods for estimating $B$ typically optimize an empirical contrast function. Formally, we define a contrast function $f(\boldsymbol{u}|P)$ by $f : \mathcal{B}_k \times \mathcal{P} \to \mathbb{R}$ where $\mathcal{B}_k$ is the unit ball in $\mathbb{R}^k$ and $\mathcal{P}$ is essentially the class of mean zero distributions on $\mathbb{R}^d$. More concretely, for suitable choices of $g$, $f(\boldsymbol{u}|P) \equiv f(\boldsymbol{u}|\boldsymbol{x}) = \mathbb{E}_{\boldsymbol{x} \sim P}[g(\boldsymbol{u}^T\boldsymbol{x})]$ for any random variable $\boldsymbol{x} \sim P$. The most popular example of a contrast function is the kurtosis, which is defined as $f(\boldsymbol{u}|P) = \mathbb{E}_{\boldsymbol{x} \sim P}(\boldsymbol{u}^T\boldsymbol{x})^4 / \mathbb{E}\left[(\boldsymbol{u}^T\boldsymbol{x})^2\right]^2 - 3$. The aim is to maximize an empirical estimate, $\hat{f}(\boldsymbol{u}|P)$. Notable contrast functions are the negentropy, mutual information (used by INFOMAX [7]), the $\tanh$ function, etc. (See [37] for further details).

**Fitting strategies:** There are two broad categories of the existing fitting strategies. (I) Find $A$ such that the components of $A\boldsymbol{x}$ are both as nongaussian and as independent as possible. See, for example, the multivariate cumulant-based method JADE [9] and characteristic function-based methods [21, 11]. The latter we shall call the PFICA method. (II) Find successively the $j^{th}$ row of $A$, denoted by $A(j,:) \in \mathbb{R}^d$, $j = 1, \ldots, k$ such that $A(j,:)^T\boldsymbol{x}$ are independent and each as nongaussian as possible, that is, estimate $A(1,:)$, project, and then apply the method again on the residuals successively. The chief representative of these methods is FastICA [27] based on univariate kurtosis.

**From noiseless to noisy ICA:** In the noiseless case one can first prewhiten the dataset using the empirical variance-covariance matrix, $\hat{\Sigma}$, of $\boldsymbol{x}^{(i)}$, using $\hat{\Sigma}^{-1/2}$. Therefore, WLOG, $B$ can be assumed to be orthogonal. Searching over orthogonal matrices not only simplifies algorithm design for fitting

---

[1]If $B$ is not invertible, this can be replaced by the Moore-Penrose inverse.

strategy (I) but also makes strategy (II) meaningful. It also greatly simplifies the analysis of kurtosis-based contrast functions (see [18]). For noisy ICA, prewhitening requires knowledge of the noise covariance, and therefore many elegant methods [4, 48, 49] avoid this step.

**Individual shortcomings of different contrast functions and fitting strategies:** To our knowledge, no single ICA algorithm or contrast function works universally across all source distributions. Adaptively determining which algorithm is useful in a given setting would be an important tool in a practitioner's toolbox. Key challenges include:

**Cumulants:** Even though contrast functions based on *even cumulants* have no spurious local optima (Theorem 3, [49]), they can vanish for some non-Gaussian distributions. Our experiments (Section 5) show that kurtosis-based contrast functions can perform poorly even when there are a few independent components with zero kurtosis. They also suffer under heavy-tailed source distributions [3, 11, 26].

**PFICA:** PFICA can outperform cumulant-based methods in some situations (Section 5). However, it is computationally much slower, and poorly performs for Bernoulli($p$) sources with small $p$.

**FastICA:** [32] show that FastICA's use of Negentropy approximations for computational efficiency may result in a contrast function where optimal directions do not correspond to directions of low entropy. tanh is another popular smooth contrast function used by FastICA. However, in [52], the authors show that this tends to return spurious solutions for some distributions [52].

Contrast functions are usually nonconvex and may have spurious local optima. However, they may still, under suitable conditions, converge to maximal directions. We introduce such a contrast function which vanishes iff $\boldsymbol{x}$ is Gaussian in Section 3.3 and does not require higher moments.

**Our contributions:**

1) Using Theorem 1 we obtain a computationally efficient scoring method for evaluating the $B$ estimated by different algorithms. Our score can also be extended to evaluate each demixing direction in sequential settings, which we do not pursue here.

2) We propose new contrast functions that work under scenarios where cumulant-based methods fail. We propose a fast sequential algorithm as in [49] to optimize these and further provide theoretical guarantees for convergence.

## 3 Main contributions

In this section, we present the "Independence score" and state the key result that justifies it. We conclude with two new contrast functions.

### 3.1 Independence score

While there are many tests for independence [22, 34, 42, 6], we choose the Characteristic function-based one because it is easy to compute and has been widely used in normality testing [19, 20, 25] and ICA [11, 21, 44]. Let us start with noiseless ICA to build intuition. Say we have estimated the inverse of the mixing matrix, i.e. $B^{-1}$ using a matrix $F$. Ideally, if $F = B^{-1}$, then $F\boldsymbol{x} = \boldsymbol{z}$, where $\boldsymbol{z}$ is a vector of independent random variables. Kac's theorem [33] tells us that $\mathbb{E}\left[\exp(it^T\boldsymbol{z})\right] = \prod_{j=1}^{k} \mathbb{E}\left[\exp(it_j z_j)\right]$ if and only if $z_i$ are independent. In [21], a novel characteristic function-based objective (CHFICA), is further analyzed and studied by [11] (PFICA). Here one minimizes $|\mathbb{E}\left[\exp(it^T F\boldsymbol{x})\right] - \prod_{j=1}^{k} \mathbb{E}\left[\exp(it_j(F\boldsymbol{x})_j)\right]|$ over $F$. We refer to this objective as the *uncorrected* independence score. We propose to adapt the CHFICA objective using *estimable parameters*, $S$, to the noisy ICA setting. We will minimize:

$$\Delta(\boldsymbol{t}, F|P) := \left| \underbrace{\mathbb{E}\left[\exp(i\boldsymbol{t}^T F\boldsymbol{x})\right] \exp\left(\frac{-\boldsymbol{t}^T \operatorname{diag}(FSF^T)\boldsymbol{t}}{2}\right)}_{\text{JOINT}} - \underbrace{\prod_{j=1}^{k} \mathbb{E}\left[\exp(it_j(F\boldsymbol{x})_j)\right] \exp\left(\frac{-\boldsymbol{t}^T FSF^T \boldsymbol{t}}{2}\right)}_{\text{PRODUCT}} \right|$$

(2)

where $S = \mathbb{E}[\boldsymbol{x}\boldsymbol{x}^T]$ and can be estimated using the sample covariance matrix. Hence, we do not require knowledge of any model parameters. We refer to this score as the (*corrected*) independence score. The second terms in JOINT and PRODUCT (Eq 2) *corrects* the original score by canceling out the additional terms resulting from the Gaussian noise using the covariance matrix S of the

data (estimated via the sample covariance). See [21] for a related score requiring knowledge of the unknown Gaussian covariance matrix $\Sigma$. We are now ready to present our first theoretical result for consistency of $\Delta(\boldsymbol{t}, F|P)$.

**Theorem 1.** *If $F \in \mathbb{R}^{k \times k}$ is invertible and the joint and marginal characteristic functions of all independent components, $\{z_i\}_{i \in [k]}$, are twice-differentiable, then $\forall \boldsymbol{t} \in \mathbb{R}^k$, $\Delta(\boldsymbol{t}, F|P) = 0$ iff $F = DB^{-1}$ where $D$ is a permutation of an arbitrary diagonal matrix.*

Unfortunately, $F$ is not uniquely defined if $\Sigma$ is unknown as we have noted in Section 1. The proof of Theorem 1 is provided in the Appendix Section A.1. When using this score in practice, $S$ is replaced with the sample covariance matrix of the data, and the expectations are replaced by the empirical estimates of the characteristic function. The convergence of this empirical score, $\Delta(\mathbf{t}, F|\hat{P})$, to the population score, $\Delta(\mathbf{t}, F|P)$, is given by the following theorem.

**Theorem 2.** *Let $\mathcal{F} := \{F \in \mathbb{R}^{k \times k} : \|F\| \leq 1\}$, $\boldsymbol{x} \sim subgaussian(\sigma)$ and $C_k := \max(1, k \log(n) \operatorname{Tr}(S))$. Then, we have*

$$\sup_{F \in \mathcal{F}} |\mathbb{E}_{\boldsymbol{t} \in N(0, I_k)} \Delta(\boldsymbol{t}, F|P) - \mathbb{E}_{\boldsymbol{t} \in N(0, I_k)} \Delta(\boldsymbol{t}, F|\hat{P})| = O_P \left( \sqrt{\frac{k^2 \|S\| \max(k, \sigma^4 \|S\|) \log^2(nC_k)}{n}} \right)$$

This bound shows that uniformly over $F$, the empirical average $\mathbb{E}_{\boldsymbol{t} \in N(0, I_k)} \Delta(\boldsymbol{t}, F|\hat{P})$, is close to the population score. This guarantees that as long as the difference between the population scores of the two candidate algorithms is not too small, the meta-algorithm can pick up the better score.

**Remark 1** (Subgaussianity assumption). *The subgaussianity assumption in Theorem 2 simply ensures the concentration of the sample covariance matrix since it is used in the score (see Theorem A.3, line 825). It can be relaxed if the concentration in the operator norm is ensured. Appendix Section A.2.1 contains experiments with more super-Gaussian source signals.*

The proof of this result is deferred to the Appendix section A.1.1. It is important to note that $\Delta(\boldsymbol{t}, F|\hat{P})$ is not easy to optimize. As we show in Section 3.2, objective functions that are computationally more amenable to optimize for ICA, e.g., cumulants, satisfy some properties (see Assumption 1). The independence score does not satisfy the first property (Assumption 1 (a)). In the *noiseless* case, whitening reduces the problem to $B$ being orthogonal. This facilitates efficient gradient descent in the Stiefel manifold of orthogonal matrices [11]. However, in the noisy setting, the prewhitening matrix contains the unknown noise covariance $\Sigma$, making optimization hard. Furthermore, as noted in Remark 2, in practice, it is better to use $\mathbb{E}_{\boldsymbol{t} \sim N(0, I_k)} \Delta(\boldsymbol{t}, F|\hat{P})$ [11, 21, 25] instead of using a fixed vector, $\boldsymbol{t}$, to evaluate $\Delta(\boldsymbol{t}, F|\hat{P})$. Any iterative optimization method requires repeatedly estimating this expectation at each gradient step. Therefore, instead of using this score directly as the optimization objective, we use it to choose between different contrast functions after extracting the demixing matrix with each. This process is referred to as the Meta algorithm (Algorithm 1).

---

**Algorithm 1** Meta-algorithm for choosing best candidate algorithm.

---

**Input**: Algorithm list $\mathcal{L}$, Data $X \in \mathbb{R}^{n \times k}$
**for** $j$ in range[1, size$(\mathcal{L})$] **do**
    $B_j \leftarrow \mathcal{L}_j(X)$ {Extract mixing matrix $B_j$ using $j^{th}$ candidate algorithm}
    $\delta_j \leftarrow \widehat{\mathbb{E}}_{\boldsymbol{t} \sim \mathcal{N}(0, \boldsymbol{I}_k)} \left[ \Delta \left( \mathbf{t}, B_j^{-1} | \hat{P} \right) \right]$
**end for**
$i_* \leftarrow \arg\min_{j \in [\text{size}(\mathcal{L})]} [\delta_j]$
**return** $B_{i_*}$

---

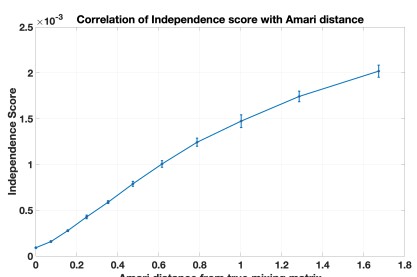

Figure 1: Correlation of independence score (with std. dev.) with Amari error between $B' = \epsilon B + (1 - \epsilon) I$ and $B$, averaged over 10 random runs.

**Remark 2** (Averaging over $\boldsymbol{t}$). *Algorithm 1 averages the independence score over $\boldsymbol{t} \sim \mathcal{N}(0, \boldsymbol{I}_k)$. While Eq 2 defines the independence score for one direction $\boldsymbol{t}$, in practice, however, there may be some directions such that a non-gaussian signal has a small score in that direction. Hence, it is desirable to average over $\boldsymbol{t}$, following the convention in [11, 21, 25].*

To gain an intuition of the independence score, we conduct an experiment where we use the dataset mentioned in Section 5 (Figure 2b) and compute the independence score for $B' = \epsilon B + (1 - \epsilon) I, \epsilon \in (0.5, 1)$. As we increase $\epsilon$, $B'$ approaches $B$, and hence the Amari error $d_{B',B}$ (see Eq 8) decreases. Figure 1 shows the independence score versus Amari error, indicating that the independence score accurately predicts solution quality even without knowing the true $B$.

## 3.2 Desired properties of contrast functions

The following properties are often useful for designing provable optimization algorithms for ICA.

**Assumption 1** (Properties of contrast functions). *Let $f$ be a contrast function defined for a mean zero random variable $X$. Then,*

*(a) $f(u|X + Y) = f(u|X) + f(u|Y)$ for $X \perp\!\!\!\perp Y$*

*(b) $f(0|X) = 0, f'(0|X) = 0, f''(0|X) = 0$*

*(c) $f(u|G) = 0, \forall u$ for $G \sim \mathcal{N}(0, 1)$*

*(d) WLOG, $f(u|X) = f(-u|X)$ (symmetry)*

Properties (a) and (c) ensure that $f(u|G + X) = f(u|X)$ for non-gaussian $X$ and independent non-gaussian $G$, which means that the additive independent Gaussian noise does not change $f$. Property (c) ensures that $|f(u|G)|$ is minimal for a Gaussian. Property (d) holds without loss of generality because one can always symmetrize the distribution. [2]

## 3.3 New contrast functions

In Section 2, we provided a discussion of the individual shortcomings of different contrast functions for existing contrast functions. Before we introduce new contrast functions in this section, we revisit the algorithmic issues posed by the added Gaussian noise with unknown $\Sigma$ in Eq 1.

Prewhitening the data is challenging for noisy ICA because $\mathbb{E}[\boldsymbol{xx}^T]$ includes the unknown Gaussian covariance matrix $\Sigma$. [4] show that it is enough to *quasi* orthogonalize the data, i.e., multiply it by a matrix, which makes the data have a diagonal covariance matrix (not necessarily the identity). Subsequent work [50, 49] uses cumulants and the Hessian of $f(\boldsymbol{u})$ to construct a matrix $C := BDB^T$ where $D$ is a diagonal matrix, and then use this matrix to achieve quasi-orthogonalization. In general, $D$ may not be positive semidefinite (e.g. when components of $\boldsymbol{z}$ have kurtosis of different signs). To remedy this, in [49], the authors propose computing the $C$ matrix using the Hessian of $f(\boldsymbol{u})$ at some fixed unit vector and then perform an elegant power method in the pseudo-Euclidean space:

$$\boldsymbol{u}_t \leftarrow \nabla f(C^\dagger \boldsymbol{u}_{t-1}|P) \big/ \|\nabla f(C^\dagger \boldsymbol{u}_{t-1}|P)\| \tag{3}$$

A pseudo-Euclidean space is a generalization of the Euclidean space used by [48]. Here the produce between vectors $\boldsymbol{u}, \boldsymbol{v}$ is given as $u^\top A v$,

---

**Algorithm 2** Pseudo-Euclidean Power Iteration for ICA (Algorithm 2 in [48]) $\tilde{B}$ is the recovered matrix for the ICA model in Eq 1. $\tilde{A}$ is a running estimate of $\tilde{B}^\dagger$.

---

**Input**: $C \in \mathbb{R}^{k \times k}, \nabla f$
$\tilde{A} \leftarrow 0, \tilde{B} \leftarrow 0$
**for** $j$ in range$[1, k]$ **do**
    Draw $\mathbf{u}$ uniformly at random from $\mathcal{S}^{k-1}$
    **while** Convergence (up to sign) **do**
        $\mathbf{u} \leftarrow \tilde{B}\tilde{A}\mathbf{u}$
        $\mathbf{u} \leftarrow \nabla f\left(C^\dagger \mathbf{u}\right) \big/ \left\|\nabla f\left(C^\dagger \mathbf{u}\right)\right\|_2$
    **end while**
    $\tilde{B}_j \leftarrow \mathbf{u}, \tilde{A}_j \leftarrow \left[C^\dagger \tilde{B}_j \big/ \left(\left(C^\dagger \tilde{B}_j\right)^\top \tilde{B}_j\right)\right]^\top$
**end for**
**return** $\tilde{B}, \tilde{A}$

---

[2]Note that $y_i := x_i - x_{\lfloor n/2 \rfloor + i}, \ i = 1 \cdots \lfloor n/2 \rfloor$ has a symmetric distribution, and also remains a noisy version of a linear combination of source signals with the same mixing matrix.

In this section, we present two new contrast functions based on the logarithm of the characteristic function (CHF) and the cumulant generating function (CGF). These do not depend on a particular cumulant, and the first only requires a finite second moment, which makes it suitable for heavy-tailed source signals. We will use $f(\boldsymbol{u}|P)$ or $f(\boldsymbol{u}|\boldsymbol{x})$ where $\boldsymbol{x} \sim P$ interchangeably to represent the population contrast function. In constructing both of the following contrast functions, we use the fact that, like the cumulants, the CGF and CHF-based contrast functions satisfy Assumption 1 (a). To satisfy Assumption 1 (c), we subtract out the part resulting from the Gaussian noise, which leads to the additional terms involving $\boldsymbol{u}^\top S \boldsymbol{u}$.

**CHF-based contrast function:** Recall that $S$ denotes the covariance matrix of $\boldsymbol{x}$. We maximize the *absolute value* of following:

$$\begin{aligned} f(\boldsymbol{u}|P) &= \log \mathbb{E} \exp(i\boldsymbol{u}^T \boldsymbol{x}) + \log \mathbb{E} \exp(-i\boldsymbol{u}^T \boldsymbol{x}) + \boldsymbol{u}^T S \boldsymbol{u} \\ &= \log(\mathbb{E}\cos(\boldsymbol{u}^T \boldsymbol{x}))^2 + \log(\mathbb{E}\sin(\boldsymbol{u}^T \boldsymbol{x}))^2 + \boldsymbol{u}^T S \boldsymbol{u} \end{aligned} \quad (4)$$

The intuition is that this is exactly zero for zero mean Gaussian data and maximizing this leads to extracting non-gaussian signal from the data. It also satisfies Assumption 1 a), b) and c).

**CGF-based contrast function:** The cumulant generating function also has similar properties (see [53] for the noiseless case). In this case, we maximize the *absolute value* of following:

$$f(\boldsymbol{u}|P) = \log \mathbb{E}\exp(\boldsymbol{u}^T \boldsymbol{x}) - \frac{1}{2}\boldsymbol{u}^T S \boldsymbol{u} \quad (5)$$

Like CHF, this vanishes iff $\boldsymbol{x}$ is mean zero Gaussian, and satisfies Assumption 1 a), b) and c). However, it cannot be expected to behave well in the heavy-tailed case. In the Appendix (Section A.1, Theorem A.1), we show that the Hessian of both functions obtained from Eq 4 and Eq 5 evaluated at some $\boldsymbol{u}$ is, in fact, of the form $BDB^T$ where $D$ is some diagonal matrix.

# 4 Global and local Convergence: loss landscape of noisy ICA

Here we present sufficient conditions for global and local convergence of a broad class of contrast functions. This covers the cumulant-based contrast functions and our CHF and CGF-based proposals.

## 4.1 Global convergence

The classical argument for global optima of kurtosis for noiseless ICA in [18] assumes WLOG that $B$ is orthogonal. This reduces the optimization over $\boldsymbol{u}$ into one for $\boldsymbol{v} = B\boldsymbol{u}$. Since $B$ is orthogonal, $\|\boldsymbol{u}\|= 1$ translates into $\|\boldsymbol{v}\|= 1$. It may seem that this idea should extend to the noisy case due to property 1 a) and c) by optimizing over $\boldsymbol{v} = B\boldsymbol{u}$ over potentially non-orthogonal mixing matrices $B$.

However, for non-orthogonal $B$, the norm constraint on $\boldsymbol{v}$ is no longer $\boldsymbol{v}^T\boldsymbol{v} = 1$, *rendering the original argument invalid*. In what follows, we extend the original argument to the noisy ICA setting by introducing pseudo-euclidean constraints. Our framework includes a large family of contrast functions, including cumulants.

To our knowledge, this is the first work to characterize the loss functions in the pseudo-euclidean space. In contrast, [50] provides global convergence of the cumulant-based methods by a convergence argument of the power method itself.

Consider the contrast function $f(\boldsymbol{u}|P) := \mathbb{E}_{\boldsymbol{x}\sim P}\left[g\left(\boldsymbol{x}^T\boldsymbol{u}\right)\right]$ and recall the definition of the quasi orthogonalization matrix $C = BDB^T$, where $D$ is a diagonal matrix. For simplicity, WLOG let us assume that $D$ is invertible. We now aim to find the optimization objective that leads to the pseudo-Euclidean update in Eq 3. For $f\left(C^{-1}\boldsymbol{u}|P\right) = \mathbb{E}_{\boldsymbol{x}\sim P}\left[g\left(\boldsymbol{u}^T C^{-1}\boldsymbol{x}\right)\right]$, consider the following:

$$f\left(C^{-1}\boldsymbol{u}|P\right) = \sum_{i\in[k]} \mathbb{E}\left[g\left(\left(B^{-1}\boldsymbol{u}\right)_i z_i/D_{ii}\right)\right] = \sum_{i\in[k]} \mathbb{E}\left[g\left(\alpha_i\tilde{z}_i\right)\right] = \sum_{i\in[k]} f\left(\alpha_i/D_{ii}|z_i\right) \quad (6)$$

where we define $\boldsymbol{\alpha} := B^{-1}\boldsymbol{u}$ and $\tilde{z}_i = z_i/D_{ii}$. We now examine $f(C^{-1}\boldsymbol{u})$ subject to the "pseudo" norm constraint $\boldsymbol{u}^T C^{-1}\boldsymbol{u} = 1$. The key point is that for suitably defined $f$, one can construct a matrix $C = BDB^T$ from the data even when $B$ is unknown. So our new objective is to optimize

$$f\left(C^{-1}\boldsymbol{u}|P\right) \text{ s.t. } \boldsymbol{u}^T C^{-1}\boldsymbol{u} = 1$$

Using the Lagrange multiplier method, optimizing the above simply gives the power method update (Eq 3) $\boldsymbol{u} \propto \nabla f(C^{-1}\boldsymbol{u})$. Furthermore, optimizing Eq 6 leads to the following transformed objective:

$$\max_{\boldsymbol{\alpha}} \left| \sum_i f\left(\alpha_i/D_{ii} \,\middle|\, z_i\right) \right| \quad \text{s.t.} \quad \sum_i \alpha_i^2/D_{ii} = 1 \tag{7}$$

Now we provide our theorem about maximizing $f(C^{-1}\boldsymbol{u}|P)$. Analogous results hold for minimization. We will denote the corresponding derivatives evaluated at $t_0$ as $f'(t_0|z_i)$ and $f''(t_0|z_i)$.

**Theorem 3.** *Let $C$ be a matrix of the form $BDB^T$, where $d_i := D_{ii} = f''(u_i|z_i)$ for some random $u_i$. Let $S_+ = \{i : d_i > 0\}$. Consider a contrast function $f : \mathbb{R} \to \mathbb{R}$. Assume that for every non-Gaussian independent component, Assumption 1 holds and the third derivative, $f'''(u|X)$, does not change the sign in the half-lines $[0, \infty)$, $(\infty, 0]$. Then $f\left(C^{-1}\boldsymbol{u} \,\middle|\, \boldsymbol{x}\right)$ with the constraint of $\langle \boldsymbol{u}, \boldsymbol{u} \rangle_{C^{-1}} = 1$ has local maxima at $B^{-1}\boldsymbol{u} = \boldsymbol{e}_i, i \in S_+$. All solutions satisfying $\boldsymbol{e}_i^T B^{-1}\boldsymbol{u} \neq 0, \ \forall i \in [1, k]$ are minima, and all solutions with $|\{i : \boldsymbol{e}_i^T B^{-1}\boldsymbol{u} \neq 0\}| < k$ are saddle points. These are the only fixed points.*

Note that the above result immediately implies that for $\tilde{C} = -C$, the same optimization in Theorem 3 will have maxima at all $\boldsymbol{u}$ such that $B^{-1}\boldsymbol{u} = \boldsymbol{e}_i, i \in S_-$.

**Corollary 3.1.** *$2k^{th}$ cumulant-based contrast functions for $k > 1$, have maxima and minima at the directions of the columns of $B$, provided $\{\boldsymbol{z}_i\}_{i \in [k]}$ have a corresponding non-zero cumulant.*

*Proof.* This proof (see Appendix Section A.1) is immediate since cumulants satisfy (a), (c). For (c) and (d), note that $f(u|Z)$ is simply $u^{2j}\kappa_{2j}(Z)$, where $\kappa_{2j}(Z)$ is the $2j^{th}$ cumulant of $Z$. □

Theorem 3 can be easily extended to the case where $C$ is not invertible. The condition on the third derivative, $f'''(u|X)$, may seem strong, and it is not hard to construct examples of random variables $Z$ where this fails for suitable contrast functions (see [36]). However, for such settings, it is hard to separate $Z$ from a Gaussian. See the Appendix A.4.1 for more details.

### 4.2 Local convergence

In this section, we establish a local convergence result for the power iteration-based method described in Eq 3. Define $\forall t \in \mathbb{R}, \forall i \in [d]$, the functions $q_i(t) := f'\left(\frac{\alpha_i}{D_{ii}} \,\middle|\, z_i\right)$. Then, without loss of generality, we have the following result for the first corner:

**Theorem 4.** *Let $\alpha_i = B^{-1}\boldsymbol{u}_i$ and $\boldsymbol{\alpha}^* := \boldsymbol{e}_1$. Let the contrast function $f(.|X)$, satisfy assumptions 1 (a), (b), (c), and (d). Let $\mathcal{B} := [-\|B^{-1}\|_2, \|B^{-1}\|_2]$ and define $c_1, c_2, c_3 > 0$ such that $\forall i \in [d]$, $\sup_{x \in \mathcal{B}} \left\{ \frac{|q_i(x)|}{c_1}, \frac{|q_i'(x)|}{c_2}, \frac{|q_i''(x)|}{c_3} \right\} \leq 1$. Define $\epsilon := \frac{\|B\|_F}{\|B\boldsymbol{e}_1\|^2} \max \left\{ \frac{c_3 + c_2\|B\boldsymbol{e}_1\|}{|q_1(\alpha_1^*)|}, \frac{c_2^2}{|q_1(\alpha_1^*)|^2}, \frac{c_3\|B\boldsymbol{e}_1\|}{|q_1'(\alpha_1^*)|} \right\}$. Let $\|\boldsymbol{\alpha}_0 - \boldsymbol{\alpha}^*\|_2 \leq R$, $R \leq \max \left\{ \frac{c_2}{c_3}, \frac{1}{5\epsilon} \right\}$. Then, we have $\forall t \geq 1$, $\frac{\|\boldsymbol{\alpha}_{t+1} - \boldsymbol{\alpha}^*\|}{\|\boldsymbol{\alpha}_t - \boldsymbol{\alpha}^*\|} \leq \frac{1}{2}$.*

Therefore, we can establish linear convergence without the third derivative constraint required for global convergence (see Theorem 3), by assuming some mild regularity conditions on the contrast functional and a sufficiently close initialization. The proof is included in Appendix Section A.1.2.

**Remark 3.** *Let $\boldsymbol{\alpha} = B^{-1}\boldsymbol{u}$ denote the demixed direction, $\boldsymbol{u}$. Theorem 4 shows that if the initial $\boldsymbol{u}_0$ is such that $\boldsymbol{\alpha}_0$ is close to the ground truth $\boldsymbol{\alpha}^*$ (which is WLOG taken to be $\boldsymbol{e}_1$), then there is geometric convergence to $\boldsymbol{\alpha}^*$. $|q_1(\alpha_1^*)|$ and $|q_1'(\alpha_1^*)|$ essentially quantify how non-gaussian the corresponding independent component is since they are zero for a mean zero Gaussian. When these are large quantities, $\epsilon$ is small, and the initialization radius $\|\boldsymbol{\alpha}_0 - \boldsymbol{\alpha}^*\|_2$ can be relatively larger.*

## 5  Experiments

In this section, we provide experiments to compare the fixed-point algorithms based on the characteristic function (CHF), the cumulant generating function (CGF) (Eqs 4 and 5) with the kurtosis-based algorithm (PEGI-$\kappa_4$ [49]). We also compare against noiseless ICA[3] algorithms - FastICA, JADE, and PFICA. These six algorithms are included as candidates for the Meta algorithm (see Algorithm

---

[3]MATLAB implementations (under the GNU General Public License) can be found at - FastICA and JADE. The code for PFICA was provided on request by the authors.

1). We also present the *Uncorrected Meta* algorithm (*Unc. Meta*) to denote the Meta algorithm with the uncorrected independence score proposed in CHFICA [21]. Table 1 and Figure 2 provide experiments on simulated data, and Figure 3 provides an application for image demixing [16]. We provide additional experiments on denoising MNIST images in the Appendix Section A.2.

**Experimental setup:** Similar to [49], the mixing matrix $B$ is constructed as $B = U\Lambda V^T$, where $U, V$ are $k$ dimensional random orthonormal matrices, and $\Lambda$ is a diagonal matrix with $\Lambda_{ii} \in [1, 3]$. The covariance matrix $\Sigma$ of the noise $\boldsymbol{g}$ follows the Wishart distribution and is chosen to be $\frac{\rho}{k} RR^T$, where $k$ is the number of sources, and $R$ is a random Gaussian matrix. Higher values of the *noise power* $\rho$ can make the noisy ICA problem harder. Keeping $B$ fixed, we report the median of 100 random runs of data generated from a given distribution (different for different experiments). The quasi-orthogonalization matrix for CHF and CGF is initialized as $\hat{B}\hat{B}^T$ using the mixing matrix, $\hat{B}$, estimated via PFICA. The performance of CHF and CGF is based on a single random initialization of the vector in the power method (see Algorithm 2). Our experiments were performed on a Macbook Pro M2 2022 CPU with 8 GB RAM.

**Error Metrics:** Due to the inherent ambiguity in signal recovery discussed in Section 2, we measure error using the accuracy of estimating $B$. We report the widely used Amari error [2, 24, 11, 10] for our results. For estimated demixing matrix $\widehat{B}$, define $W = \widehat{B}^{-1}B$, after normalizing the rows of $\widehat{B}^{-1}$ and $B^{-1}$. Then we measure $d_{\widehat{B},B}$ as -

$$d_{\widehat{B},B} := \frac{1}{k} \left( \sum_{i=1}^{k} \frac{\sum_{j=1}^{k}|W_{ij}|}{\max_j|W_{ij}|} + \sum_{j=1}^{k} \frac{\sum_{i=1}^{k}|W_{ij}|}{\max_i|W_{ij}|} \right) - 2 \tag{8}$$

**Variance reduction using Meta:** We first show that the independence score can also be used to pick the best solution from many random initializations. In Figure 2 (a), the top panel has a histogram of 40 runs of CHF, each with one random initialization. The bottom panel shows the histogram of 40 experiments, where, for each experiment, the *best out of 30 random initializations* are picked using the independence score. The top and bottom panels have (mean, standard deviation) (0.51, 0.51) and (0.39, 0.34) respectively. This shows a reduction in variance and overall better performance.

**Effect of varying kurtosis:** For our second experiment (see Table 1), we use $k = 5$ independent components, sample size $n = 10^5$ and noise power $\rho = 0.2$ from a Bernoulli($p$) distribution, where we vary $p$ from 0.001 to $0.5 - 1/\sqrt{12}$. The last parameter makes kurtosis zero. Different algorithms

| Scaled $\kappa_4$ / Algorithm | 994 | 194 | 95 | 15 | 5 | 2 | 0.8 | 0.13 | 0 |
|---|---|---|---|---|---|---|---|---|---|
| **Meta** | **0.007** | **0.010** | **0.011** | **0.010** | **0.011** | **0.011** | **0.0128** | **0.01981** | **0.023** |
| **CHF** | 1.524 | 0.336 | **0.011** | **0.010** | **0.011** | **0.011** | **0.0129** | **0.0213** | 0.029 |
| **CGF** | **0.007** | 0.011 | **0.011** | 0.016 | 0.029 | 0.044 | 0.05779 | 0.06521 | 0.071 |
| **PEGI** | **0.007** | **0.010** | **0.011** | **0.010** | 0.012 | 0.017 | 0.02795 | 0.13097 | 1.802 |
| **PFICA** | 1.525 | 0.885 | 0.540 | 0.024 | 0.023 | 0.023 | 0.0212 | 0.0224 | **0.024** |
| **JADE** | 0.021 | 0.022 | 0.021 | 0.022 | 0.022 | 0.023 | 0.029 | 0.089 | 1.909 |
| **FastICA** | 0.024 | 0.027 | 0.026 | 0.026 | 0.026 | 0.027 | 0.02874 | 0.0703 | - |
| **Unc. Meta** | 1.52 | 0.049 | 0.041 | 0.0408 | 0.0413 | 0.0414 | 0.0413 | 0.0416 | 0.0419 |

Table 1: Median Amari error with varying $p$ in Bernoulli($p$) data. The scaled kurtosis is given as $\kappa_4 := (1 - 6p(1-p))/(p(1-p))$. Observe that the Meta algorithm (shaded in red) performs at par or better than the best candidate algorithms. FastICA did not converge for zero-kurtosis data.

perform differently (best candidate algorithm is highlighted in bold font) for different $p$. In particular, characteristic function-based methods like PFICA and CHF perform poorly for small values of $p$. We attribute this to the fact that the characteristic function is close to one for small $p$. Kurtosis-based algorithms like PEGI, JADE, and FASTICA perform poorly for kurtosis close to zero. Furthermore, the Uncorrected Meta algorithm performs worse than the Meta algorithm since it shadows PFICA.

**Effect of varying noise power:** For our next two experiments, we use $k = 9$, out of which 3 are from Uniform $\left(-\sqrt{3}, \sqrt{3}\right)$, 3 are from Exponential(5) and 3 are from $\left(\text{Bernoulli}\left(\frac{1}{2} - \sqrt{\frac{1}{12}}\right)\right)$ and hence have zero kurtosis. In these experiments, we vary the sample size (Figure 2b), $n$ fixing $\rho = 0.2$, and

the noise power (Figure 2a), $\rho$ fixing $n = 10^5$, respectively. Note that with most mixture distributions, it is easily possible to have low or zero kurtosis. We include such signals in our data to highlight some limitations of PEGI-$\kappa_4$ and show that Algorithm 1 can choose adaptively to obtain better results. Figure 2a shows that large noise power leads to worse performance for all methods. PEGI, JADE,

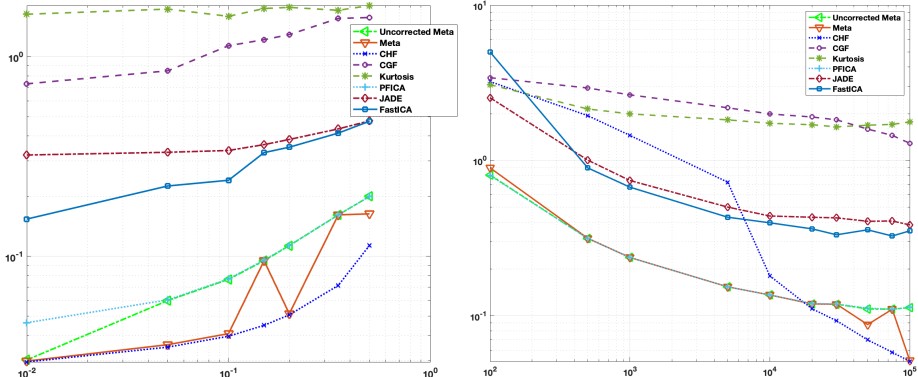

(a) Variation of performance with noise power    (b) Variation of performance with sample size

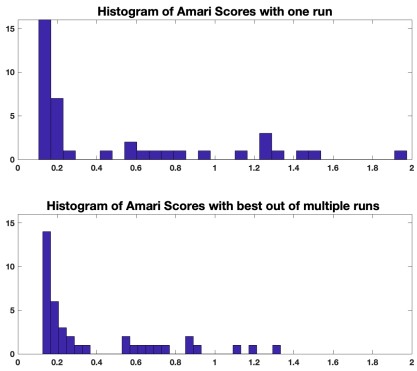

(c) Histograms of Amari error with $n = 10^4$ and noise power $\rho = 0.2$

Figure 2: Amari error in the log-scale on $y$ axis and varying noise powers (for $n = 10^5$) and varying sample sizes (for $\rho = 0.2$) on $x$ axis for figures 2a and 2b respectively. For figure 2c, the top panel contains a histogram of 40 runs with one random initialization. The bottom panel contains a histogram of 40 runs, each of which is the best independence score out of 30 random initializations.

and FastICA perform poorly consistently, because of some zero kurtosis components. CGF suffers because of heavy-tailed components. However, CHF and PFICA perform well consistently. The Meta algorithm mostly picks the best algorithm except for a few points, where the difference between the two leading algorithms is small. The Uncorrected Meta algorithm (which uses the independence score without the additional correction term introduced for noise) performs identically to PFICA.

**Effect of varying sample size:** Figure 2b shows that the noiseless ICA methods have good performance at smaller sample sizes. However, they suffer a bias in performance compared to their noiseless counterparts as the sample size increases. The Meta algorithm can consistently pick the best option amongst the candidates, irrespective of the distribution, leading to significant improvements in performance. What is interesting is that up to a sample size of $10^4$, PFICA dominates other algorithms and Meta performs like PFICA. However, after that CHF dominates, and one can see that Meta starts to have a similar trend as CHF. We also see that the Uncorrected Meta algorithm (which uses the independence score without the additional correction term introduced for noise) has a near identical error as PFICA and has a bias for large $n$.

# 6 Conclusion

ICA is a classical problem that aims to extract independent non-Gaussian components. The vast literature on both noiseless and noisy ICA introduces many different inference methods based on a

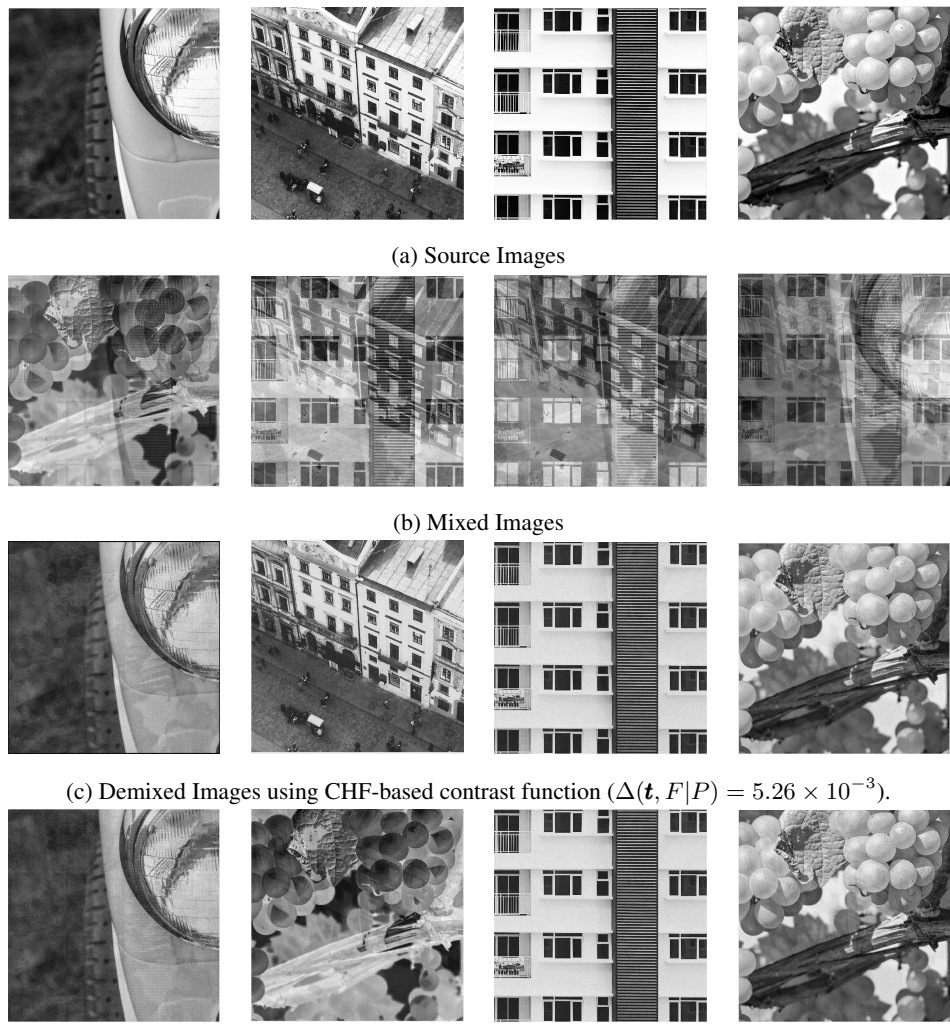

(a) Source Images

(b) Mixed Images

(c) Demixed Images using CHF-based contrast function ($\Delta(\boldsymbol{t}, F|P) = 5.26 \times 10^{-3}$).

(d) Demixed Images using Kurtosis-based contrast function ($\Delta(\boldsymbol{t}, F|P) = 2.48 \times 10^{-2}$)

Figure 3: We demix images using ICA by flattening and linearly mixing them with a $4 \times 4$ matrix $B$ (i.i.d entries $\sim \mathcal{N}(0, 1)$) and Wishart noise ($\rho = 0.001$). The CHF-based method (c) recovers the original sources well, upto sign. The Kurtosis-based method (d) fails to recover the second source. This is consistent with its higher independence score. The Meta algorithm selects CHF from candidates CHF, CGF, Kurtosis, FastICA, and JADE. Appendix Section A.2 provides results for other contrast functions and their independence scores.

variety of contrast functions for separating the non-Gaussian signal from the Gaussian noise. Each has its own set of shortcomings. We aim to identify the best method for a given dataset in a data-driven fashion. In this paper, we propose a nonparametric score, which is used to evaluate the quality of the solution of any inference method for the noisy ICA model. Using this score, we design a Meta algorithm, which chooses among a set of candidate solutions of the demixing matrix. We also provide new contrast functions and a computationally efficient optimization framework. While they also have shortcomings, we show that our diagnostic can remedy them. We provide uniform convergence properties of our score and theoretical results for local and global convergence of our methods. Simulated and real-world experiments show that our Meta-algorithm matches the accuracy of the best candidate across various distributional settings.

## Acknowledgments and Disclosure of Funding

The authors thank Aiyou Chen for many important discussions that helped shape this paper and for sharing his code of PFICA. The authors also thank Joe Neeman for sharing his valuable insights and the anonymous reviewers for their helpful suggestions in improving the exposition of the paper. SK and PS were partially supported by NSF grants 2217069, 2019844, and DMS 2109155.

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

# A  Appendix

The Appendix is organized as follows -

- Section A.1 proves Theorems 1, 2, A.1, 3 and 4
- Section A.2 provides additional experiments for noisy ICA
- Section A.3 provides the algorithm to compute independence scores via a sequential procedure
- Section A.4 explores the third derivative constraint in Theorem 3 and provides examples where it holds
- Section A.5 contains surface plots of the loss landscape for noisy ICA using the CHF-based contrast functions

## A.1  Proofs

*Proof of Theorem 1.* We will give an intuitive argument for the easy direction for $k = 2$. We will show that if $F = DB^{-1}$ for some permutation of a diagonal matrix $D$, then after projecting using that matrix, the data is of the form $\boldsymbol{z} + \boldsymbol{g}'$ for some Gaussian vector $\boldsymbol{g}'$. Let $k = 2$. Let the entries of this vector be $z_1 + g_1'$ and $z_2 + g_2'$, where $z_i$, $i \in \{1, 2\}$ are mean zero *independent* non-Gaussian random variables and $g_i'$, $i \in \{1, 2\}$ are mean zero *possibly dependent* Gaussian variables. The Gaussian variables are independent of the non-Gaussian random variables. Let $t_i$, $i \in \{1, 2\}$ be arbitrary but fixed real numbers. Let $\boldsymbol{t} = (t_1, t_2)$ and let $\Sigma_{g'}$ denote the covariance matrix of the Gaussian $g'$. Assume for simplicity $\mathrm{var}(z_i) = 1$ for $i \in \{1, 2\}$. Denote by $\Lambda := \mathrm{cov}(F\boldsymbol{x}) = FSF^T = I + \Sigma_{g'}$. The JOINT part of the score is given by:

$$\text{JOINT} = \mathbb{E}\left[it_1 z_1\right] \mathbb{E}\left[it_2 z_2\right] \exp\left(-\frac{\boldsymbol{t}^T(\Sigma_{g'} + \mathrm{diag}(\Lambda))\boldsymbol{t}}{2}\right)$$

The PRODUCT part of the score is given by

$$\text{PRODUCT} = \mathbb{E}\left[it_1 z_1\right] \mathbb{E}\left[it_2 z_2\right] \exp\left(-\frac{\boldsymbol{t}^T(\mathrm{diag}(\Sigma_{g'}) + \Lambda)\boldsymbol{t}}{2}\right)$$

It is not hard to see that JOINT equals PRODUCT. The same argument generalizes to $k > 2$. We next provide proof for the harder direction here. Suppose that $\Delta_F(\boldsymbol{t}|P) = 0$, i.e,

$$\mathbb{E}\left[\exp(i\boldsymbol{t}^T F\boldsymbol{x})\right] \exp\left(-\frac{\boldsymbol{t}^T \mathrm{diag}\left(FSF^T\right)\boldsymbol{t}}{2}\right) - \prod_{j=1}^{k} \mathbb{E}\left[\exp(it_j(F\boldsymbol{x})_j)\right] \exp\left(-\frac{\boldsymbol{t}^T FSF^T\boldsymbol{t}}{2}\right) = 0 \tag{A.9}$$

We then prove that $F$ must be of the form $DB^{-1}$ for $D$ being a permutation of a diagonal matrix. Then, taking the logarithm, we have

$$\ln\left(\mathbb{E}\left[\exp(i\boldsymbol{t}^T F\boldsymbol{x})\right]\right) - \sum_{j=1}^{k} \ln\left(\mathbb{E}\left[\exp(it_j(F\boldsymbol{x})_j)\right]\right) = \frac{1}{2}\boldsymbol{t}^T\left(\mathrm{diag}\left(FSF^T\right) - FSF^T\right)\boldsymbol{t}$$

$$= \frac{1}{2}\boldsymbol{t}^T\left(\mathrm{diag}\left(F\Sigma F^T\right) - F\Sigma F^T\right)\boldsymbol{t}$$

Under the ICA model we have, $\boldsymbol{x} = B\boldsymbol{z} + \boldsymbol{g}$. Therefore, from Eq A.9, using the definition of the characteristic function of a Gaussian random variable and noting that $\mathrm{cov}(\boldsymbol{g}) = \Sigma$, we have

$$\ln\left(\mathbb{E}\left[\exp(i\boldsymbol{t}^T F\boldsymbol{x})\right]\right) = \ln\left(\mathbb{E}\left[\exp(i\boldsymbol{t}^T FB\boldsymbol{z})\right]\right) + \ln\left(\mathbb{E}\left[\exp(i\boldsymbol{t}^T F\boldsymbol{g})\right]\right)$$

$$= \ln\left(\mathbb{E}\left[\exp(i\boldsymbol{t}^T FB\boldsymbol{z})\right]\right) - \frac{1}{2}\boldsymbol{t}^T F\Sigma F^T\boldsymbol{t}$$

and similarly,

$$\sum_{j=1}^{k} \ln\left(\mathbb{E}\left[\exp(it_j(F\boldsymbol{x})_j)\right]\right) = \sum_{j=1}^{k} \ln\left(\mathbb{E}\left[\exp(it_j(FB\boldsymbol{z})_j)\right]\right) - \frac{1}{2}\boldsymbol{t}^T \mathrm{diag}\left(F\Sigma F^T\right)\boldsymbol{t}$$

Therefore, from Eq A.9,

$$g_F\left(\boldsymbol{t}\right) := \ln\left(\mathbb{E}\left[\exp(i\boldsymbol{t}^T FB\boldsymbol{z})\right]\right) - \sum_{j=1}^{k}\ln\left(\mathbb{E}\left[\exp(it_j(FB\boldsymbol{z})_j)\right]\right)$$

must be a quadratic function of $\boldsymbol{t}$. It is important to note that the second term is an additive function w.r.t $t_1, t_2, \cdots t_k$. For simplicity, consider the case of $k = 2$ and assume that the joint and marginal characteristic functions of all signals are twice differentiable. Consider $\frac{\partial^2\left(g_F(\boldsymbol{t})\right)}{\partial t_1 \partial t_2}$, then

$$\sum_{k=1}^{2}\frac{\partial^2\left(g_F\left(\boldsymbol{t}\right)\right)}{\partial t_1 \partial t_2} \equiv const. \tag{A.10}$$

To simplify the notation, let $\psi_j(t) := \mathbb{E}\left[e^{itz_j}\right]$, $f_j \equiv \log\psi_j$, $M := FB$. The above is then equivalent to

$$\sum_{k=1}^{2} M_{1k}M_{2k}f_k''(M_{1k}t_1 + M_{2k}t_2) \equiv const. \tag{A.11}$$

Note that $M$ is invertible since $F, B$ are invertible. Therefore, let $\mathbf{s} \equiv M^T\mathbf{t}$, then

$$\sum_{k=1}^{2} M_{1k}M_{2k}f_k''(s_k) \equiv const.$$

which implies that either $M_{1k}M_{2k} = 0$ or $f_k''(s_k) \equiv const$, for $k = 1, 2$. For any $k \in \{1, 2\}$, since $z_k$ is non-Gaussian, its logarithmic characteristic function, i.e. $f_k$ cannot be a quadratic function, so

$$M_{1k}M_{2k} = 0$$

which implies that each column of $M$ has a zero. Since $M$ is invertible, thus $M$ is either a diagonal matrix or a permutation of a diagonal matrix. Now to consider $k > 2$, we just need to apply the above $k = 2$ argument to pairwise entries of $\{t_1, \cdots, t_k\}$ with other entries fixed. Hence proved. $\qquad\square$

*Proof of Theorem 3.* We start with considering the following constrained optimization problem -

$$\sup_{\boldsymbol{u}^T C^{-1}\boldsymbol{u}=1} f\left(C^{-1}\boldsymbol{u}|P\right)$$

By an application of lagrange multipliers, for the optima $\boldsymbol{u}_{\text{opt}}$, we have

$$\nu C^{-1}\boldsymbol{u}_{opt} = C^{-1}\nabla f\left((C^{-1})^T\boldsymbol{u}_{\text{opt}}|P\right), \quad \boldsymbol{u}_{\text{opt}}^T C^{-1}\boldsymbol{u}_{\text{opt}} = 1$$

with multiplier $\nu \in \mathbb{R}$. This leads to a fixed-point iteration very similar to the one in [49], which is the motivation for considering such an optimization framework. We now introduce some notations to simplify the problem.

Let $\boldsymbol{u} = B\boldsymbol{\alpha}$, $z_i' := \frac{z_i}{d_i}$ and $a_i' := \text{var}(z_i') = \frac{a_i}{d_i^2}$, where $a_i := \text{var}(z_i)$. Then,

$$\sup_{\boldsymbol{u}^T C^{-1}\boldsymbol{u}=1} f\left(C^{-1}\boldsymbol{u}|P\right) = \sup_{\boldsymbol{\alpha}^T D^{-1}\boldsymbol{\alpha}=1} f\left(D^{-1}\boldsymbol{\alpha}|\boldsymbol{z}\right)$$

We will use $f(\boldsymbol{u}|P)$ or $f(\boldsymbol{u}|\boldsymbol{x})$ where $\boldsymbol{x} \sim P$ interchangeably. By Assumption 1-(a), we see that $f\left(D^{-1}\boldsymbol{\alpha}|\boldsymbol{z}\right) = \sum_{i=1}^{k} f\left(\frac{\alpha_i}{d_i}|z_i\right)$. For simplicity of notation, we will use $h_i(\alpha_i/d_i) = f\left(\alpha_i/d_i|z_i\right)$, since the functional form can be different for different random variables $z_i$. Therefore, we seek to characterize the fixed points of the following optimization problem -

$$\sup_{\boldsymbol{\alpha}} \sum_{i=1}^{k} h_i(\alpha_i/d_i) \tag{A.12}$$

$$s.t \quad \sum_{i=1}^{k}\frac{\alpha_i^2}{d_i} = 1,$$

$$d_i \neq 0 \ \ \forall\, i \in [k]$$

where $\boldsymbol{\alpha}, \boldsymbol{d} \in \mathbb{R}^k$. We have essentially moved from optimizing in the $\langle ., .\rangle_{C^{-1}}$ space to the $\langle ., .\rangle_{D^{-1}}$ space. We find stationary points of the Lagrangian

$$\mathcal{L}(\boldsymbol{\alpha}, \lambda) := \sum_{i=1}^{k} h_i \left( \frac{\alpha_i}{d_i} \right) - \lambda \left( \sum_{i=1}^{k} \frac{\alpha_i^2}{d_i} - 1 \right) \tag{A.13}$$

The components of the gradient of $\mathcal{L}(\boldsymbol{\alpha}, \lambda)$ w.r.t $\boldsymbol{\alpha}$ are given as

$$\frac{\partial}{\partial \alpha_j} \mathcal{L}(\boldsymbol{\alpha}, \lambda) = \frac{1}{d_j} h_j' \left( \frac{\alpha_j}{d_j} \right) - \lambda \frac{\alpha_j}{d_j},$$

At the fixed point $(\boldsymbol{\alpha}, \lambda)$ we have,

$$\forall j \in [k], h_j' \left( \frac{\alpha_j}{d_j} \right) - \lambda \alpha_j = 0$$

By Assumption 1-(b), for $\alpha_i = 0$, the above is automatically zero. However, for $\{j : \alpha_j \neq 0\}$, we have,

$$\lambda = \frac{h_j' \left( \frac{\alpha_j}{d_j} \right)}{\alpha_j} \tag{A.14}$$

So, for all $\{j : \alpha_j \neq 0\}$, we must have the same value and sign of $\frac{1}{\alpha_j} h_j' \left( \frac{\alpha_j}{d_j} \right)$. By assumption in the theorem statement, $h_i'''(x)$ does not change sign on the half-lines $x > 0$ and $x < 0$. Along with Assumption 1-(b), this implies that

$$\forall x \in [0, \infty), \ \mathrm{sgn}(h_i(x)) = \mathrm{sgn}(h_i'(x)) = \mathrm{sgn}(h_i''(x)) = \mathrm{sgn}(h_i'''(x)) = \kappa_1 \tag{A.15}$$

$$\forall x \in (-\infty, 0], \ \mathrm{sgn}(h_i(x)) = -\mathrm{sgn}(h_i'(x)) = \mathrm{sgn}(h_i''(x)) = -\mathrm{sgn}(h_i'''(x)) = \kappa_2 \tag{A.16}$$

where $\kappa_1, \kappa_2 \in \{-1, 1\}$ are constants. Furthermore, we note that Assumption 1-(d) ensures that $h_i(x)$ is a symmetric function. Therefore, $\forall x \in \mathbb{R}, \mathrm{sgn}(h_i(x)) = \mathrm{sgn}(h_i''(x)) = \kappa, \kappa \in \{-1, 1\}$. Then, since $d_i = h_i''(u_i)$,

$$\mathrm{sgn}(d_i) = \mathrm{sgn}(h_i(x)), \forall x \in \mathbb{R} \tag{A.17}$$

Now, using A.14,

$$\begin{aligned}
\mathrm{sgn}(\lambda) &= \mathrm{sgn}\left( h_j' \left( \frac{\alpha_j}{d_j} \right) \right) \times \mathrm{sgn}(\alpha_j) \\
&= \mathrm{sgn}\left( \frac{\alpha_j}{d_j} \right) \times \mathrm{sgn}\left( h_j \left( \frac{\alpha_j}{d_j} \right) \right) \times \mathrm{sgn}(\alpha_j), \text{using Eq } A.15 \text{ and } A.16 \\
&= \mathrm{sgn}(d_j) \times \mathrm{sgn}\left( h_j \left( \frac{\alpha_j}{d_j} \right) \right) \\
&= 1, \text{using Eq } A.17
\end{aligned} \tag{A.18}$$

Keeping this in mind, we now compute the Hessian, $H \in \mathbb{R}^{k \times k}$, of the lagrangian, $\mathcal{L}(\boldsymbol{\alpha}, \lambda)$ at the fixed point, $(\boldsymbol{\alpha}, \lambda)$. Recall that, we have $h_i'(0) = 0$ and $h_i''(0) = 0$. Thus, for $\{i : \alpha_i = 0\}$,

$$H_{ii} = -\lambda / d_i \tag{A.19}$$

This implies that $\mathrm{sgn}(d_i H_{ii}) = \mathrm{sgn}(\lambda) = -1$ for $\{i : \alpha_i = 0\}$, using Eq A.18.

For $\{i : \alpha_i \neq 0\}$, we have

$$\begin{aligned}
H_{ij} = \left. \frac{\partial^2}{\partial \alpha_i \partial \alpha_j} \mathcal{L}(\boldsymbol{\alpha}, \lambda) \right|_{\boldsymbol{\alpha}} &= \mathbb{1}(i = j) \left[ \frac{h_i'' \left( \frac{\alpha_i}{d_i} \right)}{d_i^2} - \frac{\lambda}{d_i} \right] \\
&= \mathbb{1}(i = j) \left[ \frac{h_i'' \left( \frac{\alpha_i}{d_i} \right)}{d_i^2} - \frac{h_i' \left( \frac{\alpha_i}{d_i} \right)}{\alpha_i d_i} \right], \qquad \text{for } \alpha_i \neq 0, \text{ using } A.14 \\
&= \mathbb{1}(i = j) \frac{1}{d_i \alpha_i} \left[ \frac{\alpha_i}{d_i} h_i'' \left( \frac{\alpha_i}{d_i} \right) - h_i' \left( \frac{\alpha_i}{d_i} \right) \right], \qquad \text{for } \alpha_i \neq 0
\end{aligned}$$

We consider the pseudo inner product space $\langle .,. \rangle_{D^{-1}}$ for optimizing $\boldsymbol{\alpha}$. Furthermore, since we are in this pseudo-inner product space, we have

$$\langle \boldsymbol{v}, H\boldsymbol{v} \rangle_{D^{-1}} = \boldsymbol{v} D^{-1} H \boldsymbol{v}$$

So we will consider the positive definite-ness of the matrix $\tilde{H} := D^{-1}H$ to characterize the fixed points. Recall that for a differentiable convex function $f$, we have $\forall\, x, y \in dom(f) \subseteq \mathbb{R}^n$

$$f(y) \geq f(x) + \nabla f(x)^T (y - x)$$

Therefore, for $\{i : \alpha_i \neq 0\}$, we have

$$
\begin{aligned}
\operatorname{sgn}(d_i H_{ii}) &= \operatorname{sgn}(d_i) \times \operatorname{sgn}(d_i \alpha_i) \times \operatorname{sgn}\left( \frac{\alpha_i}{d_i} h_i''\left(\frac{\alpha_i}{d_i}\right) - h_i'\left(\frac{\alpha_i}{d_i}\right) \right) \\
&= \operatorname{sgn}(\alpha_i) \times \operatorname{sgn}\left( h_i'''\left(\frac{\alpha_i}{d_i}\right) \right), \text{ using convexity/concavity of } h'(.) \\
&= \operatorname{sgn}(\alpha_i) \times \operatorname{sgn}\left( h_i'\left(\frac{\alpha_i}{d_i}\right) \right), \text{ using } A.15 \text{ and } A.16 \\
&= \operatorname{sgn}(\alpha_i) \times \operatorname{sgn}(\lambda) \times \operatorname{sgn}(\alpha_i), \text{ using } A.14 \\
&= \operatorname{sgn}(\lambda) \\
&= 1, \text{ using Eq } A.17 \quad\quad\quad\quad\quad (A.20)
\end{aligned}
$$

Let $S := \{i : \alpha_i \neq 0\}$. We then have the following cases -

1. Assume that only one $\alpha_i$ is nonzero, then $\forall \boldsymbol{v}$ orthogonal to this direction, $\langle \boldsymbol{v}, H\boldsymbol{v} \rangle_{D^{-1}} < 0$ by Eq A.19. Thus this gives a local maxima.

2. Assume more than one $\alpha_i$ are nonzero, but $|S| < k$. Then we have for $i \notin S$, $\tilde{H}_{jj} < 0$ using A.19. For $i \in S$, $\tilde{H}_{ii} > 0$ from Eq A.20. Hence these are saddle points.

3. Assume we have $S = [k]$, i.e. $\forall i, \alpha_i \neq 0$. In this case, $\tilde{H} \succ 0$. So, we have a local minima.

This completes our proof.

$\square$

**Theorem A.1.** *Consider the data generated from the noisy ICA model (Eq 1). If $f(\boldsymbol{u}|P)$ is defined as in Eq 4 or Eq 5, we have $\nabla^2 f(\boldsymbol{u}|P) = BD_u B^T$, for some diagonal matrix $D_u$, which can be different for the differerent contrast functions.*

**Remark 4.** *The above theorem is useful because it shows that the Hessian of the contrast functions based on the CHF (eq 4 or CGF (eq 5) is of the form $BDB^T$, where $D$ is some diagonal matrix. This matrix can be precomputed at some vector $\boldsymbol{u}$ and used as the matrix $C$ in the power iteration update (see eq 3) in Algorithm 2 of [49].*

*Proof of Theorem A.1.* First, we show that the CHF-based contrast function (see Eq 4), the Hessian, is of the correct form.

**CHF-based contrast function**

$$f(\boldsymbol{u}|P) = \log \mathbb{E} \exp(i\boldsymbol{u}^T B\boldsymbol{z}) + \log \mathbb{E} \exp(-i\boldsymbol{u}^T B\boldsymbol{z}) + \boldsymbol{u}^T B \operatorname{diag}(\boldsymbol{a}) B^T \boldsymbol{u},$$

where $\operatorname{diag}(\boldsymbol{a})$ is the covariance matrix of $\boldsymbol{z}$. For simplicity of notation, define $\phi(\boldsymbol{z}; \boldsymbol{u}) := \mathbb{E}\left[\exp(i\boldsymbol{u}^T B\boldsymbol{z})\right]$.

$$\nabla f(\boldsymbol{u}|P) = iB \left( \underbrace{\frac{\mathbb{E}\left[\exp(i\boldsymbol{u}^T B\boldsymbol{z})\boldsymbol{z}\right]}{\phi(\boldsymbol{z}; \boldsymbol{u})}}_{\mu(\boldsymbol{u};\boldsymbol{z})} - i\frac{\mathbb{E}\left[\exp(-i\boldsymbol{u}^T \boldsymbol{z})\boldsymbol{z}\right]}{\phi(\boldsymbol{z}; -\boldsymbol{u})} \right) + 2B \operatorname{diag}(\boldsymbol{a}) B^T \boldsymbol{u}$$

Define

$$H(\boldsymbol{u}; \boldsymbol{z}) := \frac{\mathbb{E}\left[\exp(i\boldsymbol{u}^T B \boldsymbol{z}) \boldsymbol{z}\boldsymbol{z}^T\right]}{\phi(\boldsymbol{z}; \boldsymbol{u})} - \mu(\boldsymbol{u}; \boldsymbol{z})\mu(\boldsymbol{u}; \boldsymbol{z})^T.$$

Taking a second derivative, we have:

$$\nabla^2 f(\boldsymbol{u}|P) = B\left(-H(\boldsymbol{u}; \boldsymbol{z}) - H(\boldsymbol{z}; -\boldsymbol{u}) + 2\operatorname{diag}(\boldsymbol{a})\right) B^T$$

All we have to show at this point is that $H(\boldsymbol{u}; \boldsymbol{z})$ is diagonal. We will evaluate the $k, \ell$ entry. Let $B_i$ denote the $i^{th}$ column of $B$. Let $Y_{\setminus k, \ell} := \boldsymbol{u}^T B \sum_{j \neq k, \ell} z_j$. Using independence of the components of $\boldsymbol{z}$, we have, for $k \neq \ell$,

$$\frac{\mathbb{E}\left[\exp(i\boldsymbol{u}^T B_k z_k + i\boldsymbol{u}^T B_\ell z_\ell + Y_{\setminus k\ell})z_k z_\ell\right]}{\phi(\boldsymbol{z}; \boldsymbol{u})} = \frac{\mathbb{E}\left[\exp(i\boldsymbol{u}^T B_k z_k)z_k\right]\mathbb{E}\left[\exp(i\boldsymbol{u}^T B_\ell z_\ell)z_\ell\right]}{\mathbb{E}\left[\exp(i\boldsymbol{u}^T B_k z_k)\right]\mathbb{E}\left[\exp(i\boldsymbol{u}^T B_\ell z_\ell)\right]}$$

(A.21)

Now we evaluate:

$$\boldsymbol{e}_k^T \mu(\boldsymbol{u}; \boldsymbol{z}) := \frac{\mathbb{E}\left[\exp(i\boldsymbol{u}^T B \boldsymbol{z})z_k\right]}{\phi(\boldsymbol{z}; \boldsymbol{u})} = \frac{\mathbb{E}\left[\exp(i\boldsymbol{u}^T B_k z_k)z_k\right]}{\mathbb{E}\left[\exp(i\boldsymbol{u}^T B_k z_k)\right]}$$

(A.22)

Using Eqs A.21 and A.22, we see that indeed $H(\boldsymbol{u}; \boldsymbol{z})$ is diagonal.

Now, we prove the statement about the CGF-based contrast function.

**CGF-based contrast function**

We have,

$$\nabla f(\boldsymbol{u}|P) = \frac{\mathbb{E}\left[\exp\left(\boldsymbol{u}^T B \boldsymbol{z}\right) B \boldsymbol{z}\right]}{\mathbb{E}\left[\exp\left(\boldsymbol{u}^T B \boldsymbol{z}\right)\right]} - B\operatorname{diag}\left(\boldsymbol{a}\right) B^T \boldsymbol{u},$$

$$\nabla^2 f(\boldsymbol{u}|P) = \frac{\mathbb{E}\left[\exp\left(\boldsymbol{u}^T B \boldsymbol{z}\right) B \boldsymbol{z}\boldsymbol{z}^T B^T\right]}{\mathbb{E}\left[\exp\left(\boldsymbol{u}^T B \boldsymbol{z}\right)\right]} - \frac{\mathbb{E}\left[\exp\left(\boldsymbol{u}^T B \boldsymbol{z}\right) B \boldsymbol{z}\right]\mathbb{E}\left[\exp\left(\boldsymbol{u}^T B \boldsymbol{z}\right) \boldsymbol{z}^T B^T\right]}{\mathbb{E}\left[\exp\left(\boldsymbol{u}^T B \boldsymbol{z}\right)\right]^2} - B\operatorname{diag}\left(\boldsymbol{a}\right) B^T$$

$$= B\left[\underbrace{\frac{\mathbb{E}\left[\exp\left(\boldsymbol{u}^T B \boldsymbol{z}\right) \boldsymbol{z}\boldsymbol{z}^T\right]}{\mathbb{E}\left[\exp\left(\boldsymbol{u}^T B \boldsymbol{z}\right)\right]} - \frac{\mathbb{E}\left[\exp\left(\boldsymbol{u}^T B \boldsymbol{z}\right) \boldsymbol{z}\right]\mathbb{E}\left[\exp\left(\boldsymbol{u}^T B \boldsymbol{z}\right) \boldsymbol{z}^T\right]}{\mathbb{E}\left[\exp\left(\boldsymbol{u}^T B \boldsymbol{z}\right)\right]^2}}_{\text{Covariance of } \boldsymbol{z} \sim \mathbb{P}(\boldsymbol{z}) \cdot \frac{\exp\left(\boldsymbol{u}^T B \boldsymbol{z}\right)}{\mathbb{E}\left[\exp\left(\boldsymbol{u}^T B \boldsymbol{z}\right)\right]}} - \operatorname{diag}\left(\boldsymbol{a}\right)\right] B^T$$

The new probability density $\mathbb{P}(\boldsymbol{z}) \cdot \frac{\exp(\boldsymbol{u}^T B \boldsymbol{z})}{\mathbb{E}[\exp(\boldsymbol{u}^T B \boldsymbol{z})]}$ is an exponential tilt of the original pdf, and since $\{z_i\}_{i=1}^d$ are independent, and the new tilted density also factorizes over the $z_i$'s, therefore, the covariance under this tilted density is also diagonal. $\qquad \square$

### A.1.1 Uniform convergence

In this section, we provide the proof for Theorem 2. First, we state some preliminary results about the uniform convergence of smooth function classes which will be useful for our proofs.

#### A.1.1.1 Preliminaries

Let $D := \sup_{\theta, \tilde{\theta} \in \mathbb{T}} \rho_X\left(\theta, \tilde{\theta}\right)$ denote the diameter of set $\mathbb{T}$, and let $\mathcal{N}_X(\delta; \mathbb{T})$ denote the $\delta$-covering number of $\mathbb{T}$ in the $\rho_X$ metric. Then, we have the following standard result:

**Proposition A.2.** *(Proposition 5.17 from [51]) Let $\{X_\theta, \theta \in \mathbb{T}\}$ be a zero-mean subgaussian process with respect to the metric $\rho_X$. Then for any $\delta \in [0, D]$ such that $\mathcal{N}_X(\delta; \mathbb{T}) \geq 10$, we have*

$$\mathbb{E}\left[\sup_{\theta, \tilde{\theta} \in \mathbb{T}} \left(X_\theta - X_{\tilde{\theta}}\right)\right] \leq 2\mathbb{E}\left[\sup_{\gamma, \gamma' \in \mathbb{T}; \rho_X(\gamma, \gamma') \leq \delta} \left(X_\theta - X_{\tilde{\theta}}\right)\right] + 4\sqrt{D^2 \log\left(\mathcal{N}_X(\delta; \mathbb{T})\right)}$$

**Remark 5.** *For zero-mean subgaussian processes, Proposition A.2 implies,* $\mathbb{E}\left[\sup_{\theta \in \mathbb{T}} X_\theta\right] \leq \mathbb{E}\left[\sup_{\theta, \tilde{\theta} \in \mathbb{T}}\left(X_\theta - X_{\tilde{\theta}}\right)\right]$

**Proposition A.3.** *(Theorem 4.10 from [51]) For any b-uniformly bounded class of functions $\mathcal{C}$, any positive integer $n \geq 1$, any scalar $\delta \geq 0$, and a set of i.i.d datapoints $\left\{X^{(i)}\right\}_{i \in [n]}$ we have*

$$\sup_{f \in \mathcal{C}}\left|\frac{1}{n}\sum_{i=1}^n f\left(X^{(i)}\right) - \mathbb{E}\left[f\left(X\right)\right]\right| \leq 2\frac{1}{n}\mathbb{E}_{X,\epsilon}\left[\sup_{f \in \mathcal{C}}\left|\sum_{i=1}^n \epsilon_i f\left(X^{(i)}\right)\right|\right] + \delta$$

*with probability at least $1 - \exp\left(-\frac{n\delta^2}{2b^2}\right)$. Here $\{\epsilon_i\}_{i \in [n]}$ are i.i.d rademacher random variables.*

### A.1.1.2 Proof of theorem 2

For dataset $\left\{\boldsymbol{x}^{(i)}\right\}_{i \in [n]}, \boldsymbol{x}^{(i)} \in \mathbb{R}^k$, consider the following definitions:

$$\phi(\boldsymbol{t}, F|P) := \mathbb{E}_{\boldsymbol{x}}\left[\exp(i\boldsymbol{t}^T F\boldsymbol{x})\right], \quad \phi(\boldsymbol{t}, F|\hat{P}) := \frac{1}{n}\sum_{j=1}^n \exp(i\boldsymbol{t}^T F\boldsymbol{x}^{(j)}) \tag{A.23}$$

$$\psi(\boldsymbol{t}, F|P) := \prod_{j=1}^k \mathbb{E}_{\boldsymbol{x}}\left[\exp(it_j(F\boldsymbol{x})_j)\right], \quad \psi(\boldsymbol{t}, F|\hat{P}) := \frac{1}{n}\prod_{j=1}^k\sum_{r=1}^n \exp(it_j(F\boldsymbol{x}^{(r)})_j) \tag{A.24}$$

Let $\Delta(\boldsymbol{t}, F|\hat{P})$ be the empirical version where the expectation is replaced by sample averages. Now define:

$$\Delta(\boldsymbol{t}, F|P) = \left|\phi(\boldsymbol{t}, F|P)\exp\left(-\boldsymbol{t}^T \operatorname{diag}(FSF^T)\boldsymbol{t}\right) - \psi(\boldsymbol{t}, F|P)\exp\left(-\boldsymbol{t}^T FSF^T\boldsymbol{t}\right)\right|$$

$$\Delta(\boldsymbol{t}, F|\hat{P}) = \left|\phi(\boldsymbol{t}, F|\hat{P})\exp\left(-\boldsymbol{t}^T \operatorname{diag}(F\widehat{S}F^T)\boldsymbol{t}\right) - \psi(\boldsymbol{t}, F|\hat{P})\exp\left(-\boldsymbol{t}^T F\widehat{S}F^T\boldsymbol{t}\right)\right|$$

**Theorem A.4.** *Let $\mathcal{F} := \{F \in \mathbb{R}^{k \times k} : \|F\| \leq 1\}$. Assume that $\boldsymbol{x} \sim subgaussian(\sigma)$. We have:*

$$\sup_{F \in \mathcal{F}}|\mathbb{E}_{\boldsymbol{t} \in N(0, I_k)}\Delta(\boldsymbol{t}, F|P) - \mathbb{E}_{\boldsymbol{t} \in N(0, I_k)}\Delta(\boldsymbol{t}, F|\hat{P})| = O_P\left(\sqrt{\frac{k^2\|S\|\max(k, \sigma^4\|S\|)\log^2(nC_k)}{n}}\right)$$

*where $C_k := \max(1, k\log(n)\operatorname{Tr}(S))$.*

*Proof.* Define $\mathcal{E} = \{\boldsymbol{t} : \|\boldsymbol{t}\| \leq \sqrt{2k\log n}\}$. We will use the fact that $P(\mathcal{E}^c) \leq 2/n$ (Example 2.12, [51]). Also note that by construction, $\Delta(\boldsymbol{t}, F|P) \leq 1$. Observe that:

$$\sup_{F \in \mathcal{F}}|\mathbb{E}_{\boldsymbol{t} \in N(0, I_k)}\Delta(\boldsymbol{t}, F|P) - \mathbb{E}_{\boldsymbol{t} \in N(0, I_k)}\Delta(\boldsymbol{t}, F|\hat{P})|$$

$$\leq \sup_{F \in \mathcal{F}}\mathbb{E}_{\boldsymbol{t} \in N(0, I_k)}|\Delta(\boldsymbol{t}, F|P) - \Delta(\boldsymbol{t}, F|\hat{P})|$$

$$\leq \mathbb{E}_{\boldsymbol{t} \in N(0, I_k)}\sup_{F \in \mathcal{F}}|\Delta(\boldsymbol{t}, F|P) - \Delta(\boldsymbol{t}, F|\hat{P})|$$

$$\leq \mathbb{E}_{\boldsymbol{t} \in N(0, I_k)}\left[\sup_{F \in \mathcal{F}}\left|\Delta(\boldsymbol{t}, F|P) - \Delta(\boldsymbol{t}, F|\hat{P})\right|\Big|\mathcal{E}\right] + \mathbb{E}_{\boldsymbol{t} \in N(0, I_k)}\left[\sup_{F \in \mathcal{F}}|\Delta(\boldsymbol{t}, F|P) - \Delta(\boldsymbol{t}, F|\hat{P})|\Big|\mathcal{E}^c\right]P(\mathcal{E}^c)$$

$$\leq \mathbb{E}_{\boldsymbol{t} \in N(0, I_k)}\left[\sup_{F \in \mathcal{F}}\left|\Delta(\boldsymbol{t}, F|P) - \Delta(\boldsymbol{t}, F|\hat{P})\right|\Big|\mathcal{E}\right] + 2/n$$

$$= O_P\left(\sqrt{\frac{k^2\|S\|\max(k, \sigma^4\|S\|)\log^2(nC_k)}{n}}\right) + 2/n, \text{ using Theorem } A.5$$

The second inequality follows from Jensen's inequality, the fourth follows from $\mathbb{E}[\sup(.)] \leq \sup(\mathbb{E}[.])$. $\qquad\square$

**Theorem A.5.** *Let $\mathcal{F} := \{F \in \mathbb{R}^{k \times k} : \|F\| \leq 1\}$. Assume that $\boldsymbol{x} \sim subgaussian(\sigma)$. We have:*

$$\sup_{F \in \mathcal{F}} |\Delta(\boldsymbol{t}, F|P) - \Delta(\boldsymbol{t}, F|\hat{P})| = O_P\left(\|\boldsymbol{t}\| \sqrt{\frac{k\|S\|\max(k, \sigma^4\|S\|)\log(nC_t)}{n}}\right)$$

*where $C_t := \max(1, \|\boldsymbol{t}\|^2 \mathrm{Tr}(S))$.*

*Proof.*

$$\begin{aligned}
\Delta(\boldsymbol{t}, F|P) - \Delta(\boldsymbol{t}, F|\hat{P}) &\leq \left|\phi(\boldsymbol{t}, F|P) - \phi(\boldsymbol{t}, F|\hat{P})\right| \exp\left(-\boldsymbol{t}^T \operatorname{diag}(FSF^T)\boldsymbol{t}\right) \\
&\quad + \phi(\boldsymbol{t}, F|\hat{P}) \left|\exp\left(-\boldsymbol{t}^T \operatorname{diag}(F\widehat{S}F^T)\boldsymbol{t}\right) - \exp\left(-\boldsymbol{t}^T \operatorname{diag}(FSF^T)\boldsymbol{t}\right)\right| \\
&\quad + \left|\psi(\boldsymbol{t}, F|P) - \psi(\boldsymbol{t}, F|\hat{P})\right| \exp\left(-\boldsymbol{t}^T FSF^T\boldsymbol{t}\right) \\
&\quad + \psi(\boldsymbol{t}, F|\hat{P}) \left|\exp\left(-\boldsymbol{t}^T F\widehat{S}F^T\boldsymbol{t}\right) - \exp\left(-\boldsymbol{t}^T FSF^T\boldsymbol{t}\right)\right|
\end{aligned}$$

Finally, for some $S' = \lambda S + (1 - \lambda)\hat{S}$,

$$\begin{aligned}
|\exp(-\boldsymbol{t}^T FSF^T\boldsymbol{t}) - \exp(-\boldsymbol{t}^T F\hat{S}F^T\boldsymbol{t})| &= \left|\left\langle \partial_S \exp(-\boldsymbol{t}^T FSF^T\boldsymbol{t})\big|_{S'}, \hat{S} - S\right\rangle\right| \\
&\leq \exp(-\boldsymbol{t}^T FS'F^T\boldsymbol{t}) \left|\boldsymbol{t}^T F(S - \hat{S})F^T\boldsymbol{t}\right| \\
&\leq \|\boldsymbol{t}\|^2 \|S - \hat{S}\| = \|\boldsymbol{t}\|^2 O_P\left(\sigma^2 \|S\| \sqrt{\frac{k}{n}}\right)
\end{aligned}$$

where the last result follows from Theorem 4.7.1 in [47]. Now, for the second term, observe that:

$$\begin{aligned}
&\left|\phi(\boldsymbol{t}, F|\hat{P})\left(\exp\left(-\boldsymbol{t}^T \operatorname{diag}(F\widehat{S}F^T)\boldsymbol{t}\right) - \exp\left(-\boldsymbol{t}^T \operatorname{diag}(FSF^T)\boldsymbol{t}\right)\right)\right| \\
&\leq \exp\left(-\boldsymbol{t}^T \operatorname{diag}(FSF^T)\boldsymbol{t}\right) \left|\exp\left(-\boldsymbol{t}^T \operatorname{diag}(F(\widehat{S} - S)F^T)\boldsymbol{t}\right) - 1\right| \quad\quad\quad (\text{A.25})
\end{aligned}$$

We next have that, with probability at least $1 - 1/n$,

$$\left\|\operatorname{diag}\left(F(S - \widehat{S})F^T\right)\right\|_2 \leq \left\|F(S - \widehat{S})F^T\right\|_2 \leq C\left(\sigma^2 \|S\| \sqrt{\frac{k\log n}{n}}\right)$$

Thus with probability at least $1 - 1/n$, using the inequality $|1 - e^x| \leq 2|x|, \forall x \in [-1, 1]$, Eq A.25 leads to:

$$\left|\phi(\boldsymbol{t}, F|\hat{P})\left(\exp\left(-\boldsymbol{t}^T \operatorname{diag}(F\widehat{S}F^T)\boldsymbol{t}\right) - \exp\left(-\boldsymbol{t}^T \operatorname{diag}(FSF^T)\boldsymbol{t}\right)\right)\right| \leq 2C\|\boldsymbol{t}\|^2\left(\sigma^2 \|S\| \sqrt{\frac{k\log n}{n}}\right)$$

Note that the matrices $\operatorname{diag}\left(F(S - \widehat{S})F^T\right)$ and $FSF^T$ are positive semi-definite and $\boldsymbol{t}$ is unit-norm. Observe that, using Lemmas A.6 and A.7, we have:

$$\begin{aligned}
&\sup_{F \in \mathcal{F}} |\Delta(\boldsymbol{t}, F|P) - \Delta(\boldsymbol{t}, F|\hat{P})| \\
&\leq O_P\left(\|\boldsymbol{t}\| \sqrt{\frac{k\,\mathrm{Tr}(S)\log(nC_t)}{n}}\right) + O_P\left(\|\boldsymbol{t}\|\sigma^2 \|S\| \sqrt{\frac{k\log n}{n}}\right) + O_P\left(\|\boldsymbol{t}\| \sqrt{\frac{k^2\|S\|\log n}{n}}\right) \\
&= O_P\left(\|\boldsymbol{t}\| \sqrt{\frac{k\|S\|\max(k, \sigma^4\|S\|)\log(nC_t)}{n}}\right)
\end{aligned}$$

$\square$

**Lemma A.6.** *Define* $\mathcal{F} := \{F \in \mathbb{R}^{k \times k} : \|F\| \le 1\}$. *Let* $\{\boldsymbol{x}^{(i)}\}_{i \in [n]}$ *be i.i.d samples from a subgaussian*$(\sigma)$ *distribution. We have:*

$$\sup_{F \in \mathcal{F}} \left| \frac{1}{n} \sum_{j=1}^{n} \exp(i\boldsymbol{t}^T F \boldsymbol{x}^{(j)}) - \mathbb{E}_{\boldsymbol{x}}\left[\exp(i\boldsymbol{t}^T F \boldsymbol{x})\right] \right| = O_P\left( \|\boldsymbol{t}\| \sqrt{\frac{k \operatorname{Tr}(S) \log(nC_t)}{n}} \right)$$

*where* $C_t := \max(1, \|\boldsymbol{t}\|^2 \operatorname{Tr}(S))$.

*Proof.* Let $\phi(\boldsymbol{t}, F; \boldsymbol{x}^{(i)}) = \exp(i\boldsymbol{t}^T F \boldsymbol{x}^{(i)})$. Next, we note that $\|F\|_2 \le 1$ imply $\left\| F^T \boldsymbol{t} \right\| \le \|\boldsymbol{t}\|$. Therefore,

$$\sup_{F \in \mathcal{F}} \left| \frac{1}{n} \sum_{j=1}^{n} \exp(i\boldsymbol{t}^T F \boldsymbol{x}^{(j)}) - \mathbb{E}_{\boldsymbol{x}}\left[\exp(i\boldsymbol{t}^T F \boldsymbol{x})\right] \right| \le \sup_{\boldsymbol{u} \in \mathbb{R}^k, \|\boldsymbol{u}\| \le 1} \left| \frac{1}{n} \sum_{j=1}^{n} \exp(i\boldsymbol{u}^T \boldsymbol{x}^{(j)}) - \mathbb{E}_{\boldsymbol{x}}\left[\exp(i\boldsymbol{u}^T \boldsymbol{x})\right] \right|$$

Let $f_X(\boldsymbol{u}) = \sum_i \epsilon_i \xi(\boldsymbol{u}; \boldsymbol{x}^{(i)})$. We will argue for unit vectors $\boldsymbol{u}$, and later scale them by $\|\boldsymbol{t}\|$. Define $\mathcal{B}_k := \{\boldsymbol{u} \in \mathbb{R}^k, \|\boldsymbol{u}\| \le 1\}$, $\xi(\boldsymbol{u}; \boldsymbol{x}) := \exp(i\boldsymbol{u}^T \boldsymbol{x})$ and let

$$d(\boldsymbol{u}, \boldsymbol{u}')^2 := \sum_i (\xi(\boldsymbol{u}; \boldsymbol{x}^{(i)}) - \xi(\boldsymbol{u}'; \boldsymbol{x}^{(i)}))^2$$

Then we have,

$$\|\nabla_{\boldsymbol{u}} \exp(i\boldsymbol{u}^T \boldsymbol{x})\|_2 = \|\boldsymbol{x}\| |-\sin(\boldsymbol{u}^T \boldsymbol{x}) + i\cos(\boldsymbol{u}^T \boldsymbol{x})| \le \|\boldsymbol{x}\|$$

Then,

$$\left| \xi(\boldsymbol{u}; \boldsymbol{x}^{(i)}) - \xi(\boldsymbol{t}; \boldsymbol{x}^{(i)}) \right| \le \min\left( \|\nabla_{\boldsymbol{u}''} \xi(\boldsymbol{u}; \boldsymbol{x}^{(i)})\|_2 \|\boldsymbol{u} - \boldsymbol{u}'\|, 2 \right)$$
$$\le \min\left( \|\boldsymbol{x}^{(i)}\|_2 \|\boldsymbol{u} - \boldsymbol{u}'\|, 2 \right)$$

Let $\widehat{S} := \sum_i \boldsymbol{x}^{(i)} \left(\boldsymbol{x}^{(i)}\right)^T / n$ and $\tau_n := n \operatorname{Tr}\left(\widehat{S}\right)$. We next have,

$$D^2 := \max_{\boldsymbol{u}, \boldsymbol{u}'} d(\boldsymbol{u}, \boldsymbol{u}')^2 \le \min\left\{ 4n, \max_{\boldsymbol{u}, \boldsymbol{u}'} \sum_i \|\boldsymbol{x}^{(i)}\|_2^2 \|\boldsymbol{u} - \boldsymbol{u}'\|^2 \right\}$$
$$\le 4 \min\{n, \tau_n\}$$

Next, we bound the covering number $N(\delta; \mathcal{B}_k, d_{\boldsymbol{u}})$. Note that $N(\delta; \mathcal{B}_k, d_{\boldsymbol{u}}) \le N(\delta/\sqrt{\tau_n}; \mathcal{B}_k, \|.\|_2)$ since,

$$d\left(\boldsymbol{u}, \boldsymbol{u}'\right)^2 = \sum_i (\xi(\boldsymbol{u}; \boldsymbol{x}^{(i)}) - \xi(\boldsymbol{u}'; \boldsymbol{x}^{(i)}))^2$$
$$\le \sum_i \|\boldsymbol{x}^{(i)}\|_2^2 \|\boldsymbol{u} - \boldsymbol{u}'\|^2$$
$$= \sum_i \operatorname{Tr}\left( \boldsymbol{x}^{(i)} \left(\boldsymbol{x}^{(i)}\right)^T \right) \|\boldsymbol{u} - \boldsymbol{u}'\|^2$$

Using Cauchy-Schwarz, we have:

$$|f_X(\boldsymbol{u}) - f_X(\boldsymbol{u}')| \le \sum_{i=1}^{n} \epsilon_i |\xi(\boldsymbol{u}; \boldsymbol{x}^{(i)}) - \xi(\boldsymbol{u}; \boldsymbol{x}^{(i)})| \le \sqrt{nd(\boldsymbol{u}, \boldsymbol{u}')^2}$$

Therefore, we have $N(\delta; \mathcal{B}_k, d_{\boldsymbol{u}}) \leq \left(\frac{3\sqrt{\tau_n}}{\delta}\right)^k$. Therefore, using Proposition A.2,

$$\mathbb{E}_{\boldsymbol{x},\epsilon}\left[\sup_{F\in\mathcal{F}}|f_X(\boldsymbol{u}) - f_X(\boldsymbol{u}')|\right] \leq 2\mathbb{E}_{\boldsymbol{x}}\left[\sup_{d(\boldsymbol{u},\boldsymbol{u}')\leq\delta}|f_X(\boldsymbol{u}) - f_X(\boldsymbol{u}')|\right] + 4\mathbb{E}_{\boldsymbol{x}}\left[D\sqrt{\log N(\delta; \mathcal{B}_k, d_{\boldsymbol{u}})}\right]$$

$$\leq 2\mathbb{E}_{\boldsymbol{x}}\left[\inf_{\delta>0}\left\{\delta\sqrt{n} + 8\sqrt{\min\{n,\tau_n\}}\sqrt{2k\log\left(\frac{3\sqrt{\tau_n}}{\delta}\right)}\right\}\right]$$

$$= O\left(\mathbb{E}_{\boldsymbol{x}}\sqrt{(k\min(n,\tau_n)\log(\max(\tau_n,n)))}\right), \text{ setting } \delta := \sqrt{\min(1,\tau_n/n)}$$

$$= \sqrt{kn}O\left(\sqrt{\min(1,\mathrm{Tr}(S))\log(n\max(1,\mathrm{Tr}(S)))}\right), \text{ Cauchy-Schwarz inequality}$$

Now we recall that $\|F^T\boldsymbol{t}\| \leq \|\boldsymbol{t}\|$, since all vectors $\boldsymbol{t}$ appear as $\boldsymbol{t}^T F^T \boldsymbol{x}^{(i)}$, this simply leads to scaling $\tau$ and $\|S\|$ by $\|\boldsymbol{t}\|^2$. Dividing both sides by $n$, we have

$$\frac{1}{n}\mathbb{E}_{\boldsymbol{x},\epsilon}\left[\sup_{F\in\mathcal{F}}|f_X(\boldsymbol{t}) - f_X(\boldsymbol{t}')|\right] \leq \sqrt{\frac{k}{n}}O\left(\sqrt{\min(1,\|\boldsymbol{t}\|^2\mathrm{Tr}(S))\log(n\max(1,\|\boldsymbol{t}\|^2\mathrm{Tr}(S)))}\right)$$

Thus, using Proposition A.3 and using the definition of $C_t$, we have,

$$\sup_{F\in\mathcal{F}}|\phi(\boldsymbol{t}, F|\hat{P}) - \phi(\boldsymbol{t}, F|P)| = \sqrt{\frac{k}{n}}O\left(\sqrt{\|\boldsymbol{t}\|^2\mathrm{Tr}(S)\log(nC_t)}\right) = \|\boldsymbol{t}\|\sqrt{\frac{k}{n}}O\left(\sqrt{\mathrm{Tr}(S)\log(nC_t)}\right)$$

$$\square$$

**Lemma A.7.** *Let $\mathcal{F} = \{F \in \mathbb{R}^{k\times k} : \|F\| \leq 1\}$. We have:*

$$\sup_{F\in\mathcal{F}}\left|\psi(\boldsymbol{t}, F|\hat{P}) - \psi(\boldsymbol{t}, F|P)\right| \leq O_P\left(\|\boldsymbol{t}\|\sqrt{\frac{k^2\|S\|\log n}{n}}\right)$$

*Proof.* Let $\mathcal{B}_k := \{\boldsymbol{u} \in \mathbb{R}^k, \|\boldsymbol{u}\| \leq 1\}$ Now define $\psi_j(\boldsymbol{t}, F; \boldsymbol{x}) = \mathbb{E}[\exp(it_j F_j^T\boldsymbol{x})]$. Also define $f_X^{(j)}(F) = \frac{1}{n}\sum_i \epsilon_i\psi_j(\boldsymbol{t}, F; \boldsymbol{x}^{(i)})$. Thus, using the same argument as in Lemma A.6 for $\boldsymbol{t} \leftarrow \|\boldsymbol{t}\|\boldsymbol{e}_j$ and $\boldsymbol{x}^{(i)} \leftarrow t_j\boldsymbol{x}^{(i)}$,

$$\sup_{F\in\mathcal{F}}|\psi_j(\boldsymbol{t}, F|\hat{P}) - \psi_j(\boldsymbol{t}, F|P)| = O_P\left(|t_j|\sqrt{\frac{k\|S\|\log n}{n}}\right)$$

Finally, we see that:

$$\sup_{F\in\mathcal{F}}\left|\psi(\boldsymbol{t}, F|\hat{P}) - \psi(\boldsymbol{t}, F|P)\right| \leq \sum_j O_P\left(|t_j|\sqrt{\frac{k\|S\|\log n}{n}}\right)$$

$$= O_P\left(\sqrt{\frac{k\|S\|\log n}{n}}\right)\left(\sum_{j=1}^k |t_j|\right) = \|\boldsymbol{t}\|O_P\left(\sqrt{\frac{k^2\|S\|\log n}{n}}\right)$$

The above is true because, for $|a_i|, |b_i| \leq 1$, $i = 1, \ldots, k$,

$$\left|\prod_{i=1}^k a_i - \prod_{i=1}^k b_i\right| = \left|\sum_{j=0}^{k-1}\prod_{i\leq j}b_i(a_{j+1} - b_{j+1})\prod_{i=j+2}^k a_i\right| \leq \sum_{j=0}^{k-1}|a_{j+1} - b_{j+1}|$$

$$\square$$

### A.1.2 Local convergence

In this section, we provide the proof of Theorem 4. Recall the ICA model from Eq 1,

$$\boldsymbol{x} = B\boldsymbol{z} + \boldsymbol{g},$$

$$\mathrm{diag}(\boldsymbol{a}) := \mathrm{cov}(\boldsymbol{z}), \quad \Sigma := \mathrm{cov}(\boldsymbol{g}) = B\,\mathrm{diag}(\boldsymbol{a})\,B^T + \Sigma$$

We provide the proof for the CGF-based contrast function. The proof for the CHF-based contrast function follows similarly. From the Proof of Theorem A.1 we have,

$$\nabla f\left(\boldsymbol{u}|P\right) = \frac{\mathbb{E}\left[\exp\left(\boldsymbol{u}^T B\boldsymbol{z}\right)B\boldsymbol{z}\right]}{\mathbb{E}\left[\exp\left(\boldsymbol{u}^T B\boldsymbol{z}\right)\right]} - B\operatorname{diag}\left(\boldsymbol{a}\right)B^T\boldsymbol{u},$$

$$\nabla^2 f\left(\boldsymbol{u}|P\right) = \frac{\mathbb{E}\left[\exp\left(\boldsymbol{u}^T B\boldsymbol{z}\right)B\boldsymbol{z}\boldsymbol{z}^T B^T\right]}{\mathbb{E}\left[\exp\left(\boldsymbol{u}^T B\boldsymbol{z}\right)\right]} - \frac{\mathbb{E}\left[\exp\left(\boldsymbol{u}^T B\boldsymbol{z}\right)B\boldsymbol{z}\right]\mathbb{E}\left[\exp\left(\boldsymbol{u}^T B\boldsymbol{z}\right)\boldsymbol{z}^T B^T\right]}{\mathbb{E}\left[\exp\left(\boldsymbol{u}^T B\boldsymbol{z}\right)\right]^2} - B\operatorname{diag}\left(\boldsymbol{a}\right)B^T$$

$$= B\left[\underbrace{\frac{\mathbb{E}\left[\exp\left(\boldsymbol{u}^T B\boldsymbol{z}\right)\boldsymbol{z}\boldsymbol{z}^T\right]}{\mathbb{E}\left[\exp\left(\boldsymbol{u}^T B\boldsymbol{z}\right)\right]} - \frac{\mathbb{E}\left[\exp\left(\boldsymbol{u}^T B\boldsymbol{z}\right)\boldsymbol{z}\right]\mathbb{E}\left[\exp\left(\boldsymbol{u}^T B\boldsymbol{z}\right)\boldsymbol{z}^T\right]}{\mathbb{E}\left[\exp\left(\boldsymbol{u}^T B\boldsymbol{z}\right)\right]^2}}_{\text{Covariance of } \boldsymbol{z} \sim \frac{\mathbb{P}\left(\boldsymbol{z}\right)\exp\left(\boldsymbol{u}^T B\boldsymbol{z}\right)}{\mathbb{E}\left[\exp\left(\boldsymbol{u}^T B\boldsymbol{z}\right)\right]}} - \operatorname{diag}\left(\boldsymbol{a}\right)\right]B^T$$

The new probability density $\dfrac{\mathbb{P}\left(\boldsymbol{z}\right)\exp\left(\boldsymbol{u}^T B\boldsymbol{z}\right)}{\mathbb{E}\left[\exp\left(\boldsymbol{u}^T B\boldsymbol{z}\right)\right]}$ is an exponential tilt of the original pdf, and since $\{z_i\}_{i=1}^d$ are independent, and the new tilted density also factorizes over the $z_i$'s, therefore, the covariance under this tilted density is also diagonal.

Let $C := BD_0 B^T$. We denote the $i^{th}$ column of B as $B_{.i}$. Define functions

$$r_i\left(\boldsymbol{u}\right) := \ln\left(\mathbb{E}\left[\exp\left(\boldsymbol{u}^T B_{.i}z_i\right)\right]\right) - \operatorname{Var}\left(z_i\right)\frac{\boldsymbol{u}^T B_{.i}B_{.i}^T \boldsymbol{u}}{2}$$

$$g_i\left(\boldsymbol{u}\right) := \nabla r_i\left(\boldsymbol{u}\right)$$

Then $f\left(\boldsymbol{u}|P\right) = \sum_{i=1}^d r_i\left(\boldsymbol{u}\right)$, $\nabla f\left(\boldsymbol{u}|P\right) = \sum_{i=1}^d g_i\left(\boldsymbol{u}\right)$. For fixed point iteration, $\boldsymbol{u}_{k+1} = \dfrac{\nabla f\left(C^{-1}\boldsymbol{u}_k|P\right)}{\|\nabla f\left(C^{-1}\boldsymbol{u}_k|P\right)\|}$. Consider the function $\nabla f\left(C^{-1}\boldsymbol{u}|P\right)$. Let $\boldsymbol{e}_i$ denote the $i^{th}$ axis-aligned unit basis vector of $\mathbb{R}^n$. Since $B$ is full-rank, we can denote $\boldsymbol{\alpha} = B^{-1}\boldsymbol{u}$. We have,

$$\nabla f\left(C^{-1}\boldsymbol{u}|P\right) = \sum_{i=1}^d g_i\left(C^{-1}\boldsymbol{u}\right)$$

$$= \sum_{i=1}^d \frac{\mathbb{E}\left[\exp\left(\boldsymbol{u}^T\left(C^{-1}\right)^T B_{.i}z_i\right)B_{.i}z_i\right]}{\mathbb{E}\left[\exp\left(\boldsymbol{u}^T\left(C^{-1}\right)^T B_{.i}z_i\right)\right]} - \operatorname{Var}\left(z_i\right)B_{.i}B_{.i}^T C^{-1}\boldsymbol{u}$$

$$= \sum_{i=1}^d \frac{\mathbb{E}\left[\exp\left(\boldsymbol{u}^T\left(B^T\right)^{-1}D_0^{-1}\boldsymbol{e}_i z_i\right)B_{.i}z_i\right]}{\mathbb{E}\left[\exp\left(\boldsymbol{u}^T\left(B^T\right)^{-1}D_0^{-1}\boldsymbol{e}_i z_i\right)\right]} - \operatorname{Var}\left(z_i\right)B_{.i}\boldsymbol{e}_i^T D_0^{-1}B^{-1}u$$

$$= \sum_{i=1}^d \left(\frac{\mathbb{E}\left[\exp\left(\frac{\alpha_i}{(D_0)_{ii}}z_i\right)z_i\right]}{\mathbb{E}\left[\exp\left(\frac{\alpha_i}{(D_0)_{ii}}z_i\right)\right]} - \operatorname{Var}\left(z_i\right)\frac{\alpha_i}{(D_0)_{ii}}\right)B_{.i}$$

For $t \in \mathbb{R}$, define the function $q_i\left(t\right) : \mathbb{R} \to \mathbb{R}$ as -

$$q_i\left(t\right) := \frac{\mathbb{E}\left[\exp\left(\frac{t}{(D_0)_{ii}}z_i\right)z_i\right]}{\mathbb{E}\left[\exp\left(\frac{t}{(D_0)_{ii}}z_i\right)\right]} - \operatorname{Var}\left(z_i\right)\frac{t}{(D_0)_{ii}}$$

Note that $q_i\left(0\right) = \mathbb{E}\left[z_i\right] = 0$. For $\boldsymbol{t} = \left(t_1, t_2, \cdots t_d\right) \in \mathbb{R}^d$, let $\boldsymbol{q}\left(\boldsymbol{t}\right) := \left[q_1\left(t_1\right), q_2\left(t_2\right)\cdots q_d\left(t_d\right)\right]^T \in \mathbb{R}^d$. Then,

$$\nabla f\left(C^{-1}u|P\right) = B\boldsymbol{q}\left(\boldsymbol{\alpha}\right)$$

Therefore, if $\boldsymbol{u}_{t+1} = B\boldsymbol{\alpha}_{t+1}$, then

$$B\boldsymbol{\alpha}_{t+1} = \frac{B\boldsymbol{q}(\boldsymbol{\alpha}_t)}{\|B\boldsymbol{q}(\boldsymbol{\alpha}_t)\|}$$

$$\implies \boldsymbol{\alpha}_{t+1} = \frac{\boldsymbol{q}(\boldsymbol{\alpha}_t)}{\|B\boldsymbol{q}(\boldsymbol{\alpha}_t)\|}$$

$$\implies \forall i \in [d], \ (\boldsymbol{\alpha}_{t+1})_i = \frac{q_i((\boldsymbol{\alpha}_t)_i)}{\|B\boldsymbol{q}(\boldsymbol{\alpha}_t)\|}$$

At the fixed point, $\boldsymbol{\alpha}^* = \dfrac{\boldsymbol{e}_1}{\|B\boldsymbol{e}_1\|}$ and $\dfrac{B\boldsymbol{q}(\boldsymbol{\alpha}^*)}{\|B\boldsymbol{q}(\boldsymbol{\alpha}^*)\|_2} = B\boldsymbol{e}_1$. Therefore,

$$\forall i \in [2, d], (\boldsymbol{\alpha}_{t+1})_i - \alpha_i^* = (\boldsymbol{\alpha}_{t+1})_i = \frac{q_i((\boldsymbol{\alpha}_t)_i)}{\|B\boldsymbol{q}(\boldsymbol{\alpha}_t)\|} \tag{A.26}$$

$$\text{for } i = 1, (\boldsymbol{\alpha}_{t+1})_1 - \alpha_1^* = \frac{q_1((\boldsymbol{\alpha}_t)_i)}{\|B\boldsymbol{q}(\boldsymbol{\alpha}_t)\|} - \frac{1}{\|B\boldsymbol{e}_1\|} \tag{A.27}$$

Note the smoothness assumptions on $q_i(.)$ mentioned in Theorem 4, $\forall i \in [d]$,

1. $\sup_{t \in [-\|B^{-1}\|_2, \|B^{-1}\|_2]} |q_i(t)| \le c_1$
2. $\sup_{t \in [-\|B^{-1}\|_2, \|B^{-1}\|_2]} |q_i'(t)| \le c_2$
3. $\sup_{t \in [-\|B^{-1}\|_2, \|B^{-1}\|_2]} |q_i''(t)| \le c_3$

Since $\forall k, \ \|\boldsymbol{u}_k\| = 1$, therefore, $\forall k, \ \|\boldsymbol{\alpha}_t\| \le \|B^{-1}\|_2$. We seek to prove that the sequence $\{\boldsymbol{\alpha}_t\}_{k=1}^n$ converges to $\boldsymbol{\alpha}^*$.

### A.1.2.1  Taylor expansions

Consider the function $g_i(\boldsymbol{y}) := \dfrac{q_i(y_i)}{\|B\boldsymbol{q}(\boldsymbol{y})\|}$ for $\boldsymbol{y} \in \mathbb{R}^d$. We start by computing the gradient, $\nabla_{\boldsymbol{y}} g_i(\boldsymbol{y})$.

**Lemma A.8.** *Let* $g_i(\boldsymbol{y}) := \dfrac{q_i(y_i)}{\|B\boldsymbol{q}(\boldsymbol{y})\|}$, *then we have*

$$[\nabla_{\boldsymbol{y}} g_i(\boldsymbol{y})]_j = \begin{cases} \dfrac{1}{\|B\boldsymbol{q}(\boldsymbol{y})\|} \dfrac{\partial q_i(y_i)}{\partial y_i} - \dfrac{q_i(y_i)}{\|B\boldsymbol{q}(\boldsymbol{y})\|^2} \dfrac{\partial \|B\boldsymbol{q}(\boldsymbol{y})\|}{\partial q_i(y_i)} \dfrac{\partial q_i(y_i)}{\partial y_i}, & \text{for } j = i \\[3mm] -\dfrac{q_j'(y_j)}{\|B\boldsymbol{q}(\boldsymbol{y})\|} \dfrac{q_i(y_i) \boldsymbol{e}_j^T B^T B\boldsymbol{q}(\boldsymbol{y})}{\|B\boldsymbol{q}(\boldsymbol{y})\|^2}, & \text{for } j \neq i \end{cases}$$

*Proof.* The derivative w.r.t $y_i$ is given as -

$$\frac{\partial g_i(\boldsymbol{y})}{\partial y_i} = \frac{1}{\|B\boldsymbol{q}(\boldsymbol{y})\|} \frac{\partial q_i(y_i)}{\partial y_i} - \frac{q_i(y_i)}{\|B\boldsymbol{q}(\boldsymbol{y})\|^2} \frac{\partial \|B\boldsymbol{q}(\boldsymbol{y})\|}{\partial y_i}$$

$$= \frac{1}{\|B\boldsymbol{q}(\boldsymbol{y})\|} \frac{\partial q_i(y_i)}{\partial y_i} - \frac{q_i(y_i)}{\|B\boldsymbol{q}(\boldsymbol{y})\|^2} \frac{\partial \|B\boldsymbol{q}(\boldsymbol{y})\|}{\partial q_i(y_i)} \frac{\partial q_i(y_i)}{\partial y_i}$$

Note that

$$\nabla_{\boldsymbol{y}} \|A\boldsymbol{y}\|_2 = \frac{1}{\|A\boldsymbol{y}\|} A^T A\boldsymbol{y}$$

Therefore,

$$\frac{\partial g_i(\boldsymbol{y})}{\partial y_i} = \frac{1}{\|B\boldsymbol{q}(\boldsymbol{y})\|} \frac{\partial q_i(y_i)}{\partial y_i} - \frac{q_i(y_i)}{\|B\boldsymbol{q}(\boldsymbol{y})\|^2} \frac{1}{\|B\boldsymbol{q}(\boldsymbol{y})\|} \boldsymbol{e}_i^T B^T B\boldsymbol{q}(\boldsymbol{y}) \frac{\partial q_i(y_i)}{\partial y_i}$$

$$= \frac{q_i'(y_i)}{\|B\boldsymbol{q}(\boldsymbol{y})\|} \left[ 1 - \frac{q_i(y_i) \boldsymbol{e}_i^T B^T B\boldsymbol{q}(\boldsymbol{y})}{\|B\boldsymbol{q}(\boldsymbol{y})\|^2} \right] \tag{A.28}$$

For $j \neq i$, the derivative w.r.t $y_j$ is given as -

$$\frac{\partial g_i(\boldsymbol{y})}{\partial y_j} = -\frac{q_i(y_i)}{\|B\boldsymbol{q}(\boldsymbol{y})\|^2} \frac{\partial \|B\boldsymbol{q}(\boldsymbol{y})\|}{\partial q_j(y_j)} \frac{\partial q_j(y_j)}{\partial y_j}$$

$$= -\frac{q_j'(y_j)}{\|B\boldsymbol{q}(\boldsymbol{y})\|} \frac{q_i(y_i) \boldsymbol{e}_j^T B^T B\boldsymbol{q}(\boldsymbol{y})}{\|B\boldsymbol{q}(\boldsymbol{y})\|^2} \tag{A.29}$$

$\square$

Next, we bound $q_i(t)$ and $q_i'(t)$.

**Lemma A.9.** *Under the smoothness assumptions on $q_i(.)$ mentioned in Theorem 4, we have for* $t \in [-\|B^{-1}\|_2, \|B^{-1}\|_2]$,

1. $|q_1(t) - q_1(\alpha_1^*)| \leq c_2 |t - \alpha_1^*|, \quad |q_1'(t) - q_1'(\alpha_1^*)| \leq c_3 |t - \alpha_1^*|$

2. $\forall i, |q_i(t)| \leq \frac{c_3 t^2}{2}, \quad |q_i'(t)| \leq c_3 |t|$

*Proof.* First consider $q_i(t)$ and $q_i'(t)$ for $i \neq 1$. Using Taylor expansion around $t = 0$, we have for some $c \in (0, t)$ and $\forall i \neq 1$,

$$q_i(t) = q_i(0) + t q_i'(0) + \frac{t^2}{2} q_i''(c) \text{ and}$$

$$q_i'(t) = q_i'(0) + t q_i''(0)$$

Now, we know that $q_i(0) = q_i'(0) = 0$. Then using Assumption 1, we have, for $t \in [-\|B^{-1}\|_2, \|B^{-1}\|_2]$,

$$|q_i(t)| \leq \frac{c_3 t^2}{2} \text{ and} \tag{A.30}$$

$$|q_i'(t)| \leq c_3 |t| \tag{A.31}$$

Similarly, using Taylor expansion around $\alpha_1^* = \frac{1}{\|B\boldsymbol{e}_1\|_2}$ for $q_1(.)$ we have, for some $c' \in (0, t)$

$$q_1(t) = q_1(\alpha_1^*) + (t - \alpha_1^*) q_1'(c') \text{ and}$$

$$q_1'(t) = q_1'(\alpha_1^*) + (t - \alpha_1^*) q_1''(c')$$

Therefore, using Assumption 4, we have, for $t \in [-\|B^{-1}\|_2, \|B^{-1}\|_2]$

$$|q_1(t) - q_1(\alpha_1^*)| \leq c_2 |t - \alpha_1^*| \text{ and} \tag{A.32}$$

$$|q_1'(t) - q_1'(\alpha_1^*)| \leq c_3 |t - \alpha_1^*| \tag{A.33}$$

$\square$

### A.1.2.2 Using convergence radius

In this section, we use the Taylor expansion results for $q_i(.)$ and $q_i'(.)$ in the previous section to analyze the following functions for $\boldsymbol{y} \in \mathbb{R}^d$,

1. $w(\boldsymbol{y}) := \|B\boldsymbol{y}\|$

2. $v(\boldsymbol{y}; i) := \frac{\boldsymbol{e}_i^T B^T B\boldsymbol{q}(\boldsymbol{y})}{\|B\boldsymbol{q}(\boldsymbol{y})\|^2}$

Under the constraints specified in the theorem statement we have,

$$\|\boldsymbol{y} - \boldsymbol{\alpha}^*\|_2 \leq R, \ R \leq \max\{c_2/c_3, 1\},$$

$$\epsilon := \frac{\|B\|_F}{\|B\boldsymbol{e}_1\|} \max\left\{\frac{c_2}{|q_1(\alpha_1^*)|}, \frac{c_3}{|q_1'(\alpha_1^*)|}\right\}, \ \epsilon R \leq \frac{1}{10} \tag{A.34}$$

We start with $w(\boldsymbol{y})$.

**Lemma A.10.** $\forall\, \boldsymbol{y} \in \mathbb{R}^d$, *satisfying* (A.34), *let* $\delta\left(\boldsymbol{y}\right) := \boldsymbol{q}\left(\boldsymbol{y}\right) - q_1\left(\alpha_1^*\right)\boldsymbol{e}_1$. *Then, we have*

1. $\left(1 - \epsilon\|\boldsymbol{y} - \boldsymbol{\alpha}^*\|\right)|q_1\left(\alpha_1^*\right)|\,\|B\boldsymbol{e}_1\| \leq \|B\boldsymbol{q}\left(\boldsymbol{y}\right)\| \leq \left(1 + \epsilon\|\boldsymbol{y} - \boldsymbol{\alpha}^*\|\right)|q_1\left(\alpha_1^*\right)|\,\|B\boldsymbol{e}_1\|$

2. $\|\delta\left(\boldsymbol{y}\right)\| \leq c_2\|\boldsymbol{y} - \boldsymbol{\alpha}^*\|$

*Proof.* Consider the vector $\delta\left(\boldsymbol{y}\right) := \boldsymbol{q}\left(\boldsymbol{y}\right) - q_1\left(\alpha_1^*\right)\boldsymbol{e}_1$. Then,

$$|(\delta\left(\boldsymbol{y}\right))_\ell| \leq \begin{cases} c_2\,|y_1 - \alpha_1^*|, & \text{for } \ell = 1, \text{ using Eq A.32} \\ \frac{c_3}{2}\,(y)_\ell^2, & \text{for } \ell \neq 1, \text{ using Eq A.30} \end{cases} \tag{A.35}$$

Note that

$$\|\delta\left(\boldsymbol{y}\right)\| \leq c_2\|\boldsymbol{y} - \boldsymbol{\alpha}^*\| \tag{A.36}$$

Now consider $w\left(\boldsymbol{y}\right)$. Using the mean-value theorem on the Euclidean norm we have,

$$\|B\boldsymbol{q}\left(\boldsymbol{y}\right)\| = |q_1\left(\alpha_1^*\right)|\,\|B\boldsymbol{e}_1\| + \frac{1}{\|B\gamma\left(\boldsymbol{y}\right)\|}\left(B^T B\gamma\left(\boldsymbol{y}\right)\right)^T \delta\left(\boldsymbol{y}\right), \tag{A.37}$$

where $\gamma\left(\boldsymbol{y}\right) = \mu\boldsymbol{q}\left(\boldsymbol{y}\right) + \left(1 - \mu\right)q_1\left(\alpha_1^*\right)\boldsymbol{e}_1,\ \mu \in (0, 1)$. Then,

$$\begin{aligned}
\left\|\frac{1}{\|B\gamma\left(\boldsymbol{y}\right)\|}\left(B^T B\gamma\left(\boldsymbol{y}\right)\right)^T \delta\left(\boldsymbol{y}\right)\right\| &\leq \frac{1}{\|B\gamma\left(\boldsymbol{y}\right)\|}\,\|B^T B\gamma\left(\boldsymbol{y}\right)\|\,\|\delta\left(\boldsymbol{y}\right)\| \\
&\leq \frac{1}{\|B\gamma\left(\boldsymbol{y}\right)\|}\,\|B\|\,\|B\gamma\left(\boldsymbol{y}\right)\|\,\|\delta\left(\boldsymbol{y}\right)\| \\
&= \|B\|\,\|\delta\left(\boldsymbol{y}\right)\| \\
&\leq c_2\,\|B\|\,\|\boldsymbol{y} - \boldsymbol{\alpha}^*\|, \text{ using Eq A.36} \\
&\leq c_2\,\|B\|_F\,\|\boldsymbol{y} - \boldsymbol{\alpha}^*\|, \\
&\leq \epsilon\,|q_1\left(\alpha_1^*\right)|\,\|B\boldsymbol{e}_1\|\,\|\boldsymbol{y} - \boldsymbol{\alpha}^*\|, \text{ using Eq A.34}
\end{aligned} \tag{A.38}$$

Therefore using A.37,

$$\left(1 - \epsilon\|\boldsymbol{y} - \boldsymbol{\alpha}^*\|\right)|q_1\left(\alpha_1^*\right)|\,\|B\boldsymbol{e}_1\| \leq \|B\boldsymbol{q}\left(\boldsymbol{y}\right)\| \leq \left(1 + \epsilon\|\boldsymbol{y} - \boldsymbol{\alpha}^*\|\right)|q_1\left(\alpha_1^*\right)|\,\|B\boldsymbol{e}_1\| \tag{A.39}$$

$\square$

Finally, we consider the function $v\left(\boldsymbol{y}; i\right) := \dfrac{\boldsymbol{e}_i^T B^T B\boldsymbol{q}\left(\boldsymbol{y}\right)}{\|B\boldsymbol{q}\left(\boldsymbol{y}\right)\|^2}$.

**Lemma A.11.** $\forall\, \boldsymbol{y} \in \mathbb{R}^d$ *satisfying* (A.34), *we have*

1. *For* $i = 1$, $1 - 5\epsilon\|\boldsymbol{y} - \boldsymbol{\alpha}^*\| \leq q_1\left(\alpha_1^*\right)v\left(\boldsymbol{y}|1\right) \leq 1 + 5\epsilon\|\boldsymbol{y} - \boldsymbol{\alpha}^*\|$

2. *For* $i \neq 1$, $|q_1\left(\alpha_1^*\right)v\left(\boldsymbol{y}; i\right)| \leq \left(1 + 5\epsilon\|\boldsymbol{y} - \boldsymbol{\alpha}^*\|\right)\dfrac{\|B\boldsymbol{e}_i\|}{\|B\boldsymbol{e}_1\|}$

*Proof.* Define $\theta := \epsilon\|\boldsymbol{y} - \boldsymbol{\alpha}^*\|$ for convenience of notation. We have,

$$\frac{\boldsymbol{e}_i^T B^T B\boldsymbol{q}\left(\boldsymbol{y}\right)}{\|B\boldsymbol{q}\left(\boldsymbol{y}\right)\|^2} = q_1\left(\alpha_1^*\right)\frac{\boldsymbol{e}_i^T B^T B\boldsymbol{e}_1}{\|B\boldsymbol{q}\left(\boldsymbol{y}\right)\|^2} + \frac{\boldsymbol{e}_i^T B^T B\delta\left(\boldsymbol{y}\right)}{\|B\boldsymbol{q}\left(\boldsymbol{y}\right)\|^2}, \text{ using definition of } \delta\left(\boldsymbol{y}\right) \tag{A.40}$$

Therefore, if $i = 1$, then using Lemma A.10,

$$\frac{1}{\left(1 + \theta\right)^2} + \frac{q_1\left(\alpha_1^*\right)\boldsymbol{e}_1^T B^T B\delta\left(\boldsymbol{y}\right)}{\|B\boldsymbol{q}\left(\boldsymbol{y}\right)\|^2} \leq \frac{q_1\left(\alpha_1^*\right)\boldsymbol{e}_1^T B^T B\boldsymbol{q}\left(\boldsymbol{y}\right)}{\|B\boldsymbol{q}\left(\boldsymbol{y}\right)\|^2} \leq \frac{1}{\left(1 - \theta\right)^2} + \frac{q_1\left(\alpha_1^*\right)\boldsymbol{e}_1^T B^T B\delta\left(\boldsymbol{y}\right)}{\|B\boldsymbol{q}\left(\boldsymbol{y}\right)\|^2} \tag{A.41}$$

Using the following inequalities -

$$\frac{1}{\left(1 + \theta\right)^2} \geq 1 - 2\theta, \theta \geq 0, \text{ and, } \frac{1}{\left(1 - \theta\right)^2} \leq 1 + 5\theta, \theta \leq \frac{1}{5}$$

and observing that using Eq A.34, and Lemma A.10,

$$\left| \frac{q_1\left(\alpha_1^*\right)\boldsymbol{e}_1^T B^T B \delta\left(\boldsymbol{y}\right)}{\|B\boldsymbol{q}\left(\boldsymbol{y}\right)\|^2} \right| \leq \left|q_1\left(\alpha_1^*\right)\right| \frac{\|B\boldsymbol{e}_1\|}{\left(1-\theta\right)^2 \left|q_1\left(\alpha_1^*\right)\right|^2 \|B\boldsymbol{e}_1\|^2} \|B\| \|\delta\left(\boldsymbol{y}\right)\|$$

$$\leq \frac{1}{\left(1-\theta\right)^2} \frac{c_2\|B\|}{\left|q_1\left(\alpha_1^*\right)\right| \|B\boldsymbol{e}_1\|} \|\boldsymbol{y} - \boldsymbol{\alpha}^*\|$$

$$\leq \frac{\epsilon}{\left(1-\theta\right)^2} \|\boldsymbol{y} - \boldsymbol{\alpha}^*\|$$

Therefore we have from Eq A.41,

$$1 - 5\theta \leq q_1\left(\alpha_1^*\right) v\left(\boldsymbol{y}; i\right) \leq 1 + 5\theta \tag{A.42}$$

where we used $\theta := \epsilon \|\boldsymbol{y} - \boldsymbol{\alpha}^*\|$. For the case of $i \neq 1$, using Lemma A.10 we have

$$\left|q_1\left(\alpha_1^*\right)\right| \left|v\left(\boldsymbol{y}; i\right)\right| = \left|q_1\left(\alpha_1^*\right)\right| \left| \frac{\boldsymbol{e}_i^T B^T B\boldsymbol{q}\left(\boldsymbol{y}\right)}{\|B\boldsymbol{q}\left(\boldsymbol{y}\right)\|^2} \right|$$

$$\leq \left|q_1\left(\alpha_1^*\right)\right| \left| \frac{\|B\boldsymbol{e}_i\|}{\|B\boldsymbol{q}\left(\boldsymbol{y}\right)\|} \right|$$

$$\leq \frac{1}{\left(1-\theta\right)^2} \frac{\|B\boldsymbol{e}_i\|}{\|B\boldsymbol{e}_1\|}$$

$$\leq \left(1 + 5\theta\right) \frac{\|B\boldsymbol{e}_i\|}{\|B\boldsymbol{e}_1\|} \tag{A.43}$$

$\square$

We now operate under the assumption that Eq A.34 holds for $\boldsymbol{y} = \boldsymbol{\alpha}_t$ and inductively show that it holds for $\boldsymbol{y} = \boldsymbol{\alpha}_{t+1}$ as well. Recall the function $g_i\left(\boldsymbol{y}\right) := \frac{q_i\left(y_i\right)}{\|B\boldsymbol{q}\left(\boldsymbol{y}\right)\|}$. By applying the mean-value theorem for $g_i\left(.\right)$ for the points $\boldsymbol{\alpha}_t$ and $\boldsymbol{\alpha}^*$, we have from Eq A.26 and A.27 -

$$\left|\left(\boldsymbol{\alpha}_{t+1}\right)_i - \alpha_i^*\right| = \left|g_i\left(\boldsymbol{\alpha}_t\right) - g_i\left(\boldsymbol{\alpha}^*\right)\right|$$

$$= \left|\nabla g_i\left(\boldsymbol{\beta}_i\right)^T \left(\boldsymbol{\alpha}_t - \boldsymbol{\alpha}^*\right)\right| \text{ for } \boldsymbol{\beta}_i := \left(1 - \lambda_i\right)\boldsymbol{\alpha}_t + \lambda_i \boldsymbol{\alpha}^*, \ \lambda_i \in \left(0, 1\right)$$

$$\leq \|\nabla g_i\left(\boldsymbol{\beta}_i\right)\| \|\boldsymbol{\alpha}_t - \boldsymbol{\alpha}^*\| \tag{A.44}$$

Note the induction hypothesis assumes that Eq A.34 is true for $x = \boldsymbol{\alpha}_t$. Since $\forall i, \lambda_i \in \left(0, 1\right)$, therefore Eq A.34 holds for all $\boldsymbol{\beta}_i$ as well. Squaring and adding Eq A.44 for $i \in \left[d\right]$ and taking a square-root, we have

$$\|\boldsymbol{\alpha}_{t+1} - \boldsymbol{\alpha}^*\| \leq \|\boldsymbol{\alpha}_t - \boldsymbol{\alpha}^*\| \sqrt{\sum_{i=1}^k \|\nabla g_i\left(\boldsymbol{\beta}_i\right)\|^2}$$

$$\leq \|\boldsymbol{\alpha}_t - \boldsymbol{\alpha}^*\| \sqrt{\sum_{i=1}^k \sum_{j=1}^k \left(\left. \frac{\partial g_i\left(\boldsymbol{y}\right)}{\partial y_j} \right|_{\boldsymbol{\beta}_i}\right)^2} \tag{A.45}$$

Let us consider the expression $G_{ij} := \left. \frac{\partial g_i\left(\boldsymbol{y}\right)}{\partial y_j} \right|_{\boldsymbol{\beta}_i}, \forall i, j \in \left[k\right]$. For the purpose of the subsequent analysis, we define $\theta := \epsilon \|\boldsymbol{\alpha}_t - \boldsymbol{\alpha}^*\|$. We divide the analysis into the following cases -

**Case 1** : $i = 1, j = 1$

Using Lemma A.8,

$$\left|G_{11}\right| = \left| \frac{q_1'\left(\left(\boldsymbol{\beta}_1\right)_1\right)}{\|Bw\left(\boldsymbol{\beta}_1\right)\|} \left[1 - \frac{q_1\left(\left(\boldsymbol{\beta}_1\right)_1\right)\boldsymbol{e}_1^T B^T B\boldsymbol{q}\left(\boldsymbol{\beta}_1\right)}{\|B\boldsymbol{q}\left(\boldsymbol{\beta}_1\right)\|^2} \right] \right|$$

From Lemma A.9,

$$
\begin{aligned}
|q_1'\left((\boldsymbol{\beta}_1)_1\right)| &\leq |q_1'\left(\alpha_1^*\right)| + c_3 \left|(\boldsymbol{\beta}_1)_1 - \alpha_1^*\right| \\
&= |q_1'\left(\alpha_1^*\right)| \left(1 + c_3 \frac{\left|(\boldsymbol{\beta}_1)_1 - \alpha_1^*\right|}{|q_1'\left(\alpha_1^*\right)|}\right) \\
&\leq |q_1'\left(\alpha_1^*\right)| \left(1 + c_3 \frac{\|\boldsymbol{\alpha}_t - \boldsymbol{\alpha}^*\|}{|q_1'\left(\alpha_1^*\right)|}\right) \\
&\leq |q_1'\left(\alpha_1^*\right)| \left(1 + \theta\right)
\end{aligned}
$$

and,

$$
\begin{aligned}
q_1\left((\boldsymbol{\beta}_1)_1\right) &\leq |q_1\left(\alpha_1^*\right)| + c_2 \left|(\boldsymbol{\beta}_1)_1 - \alpha_1^*\right| \\
&= |q_1\left(\alpha_1^*\right)| \left(1 + c_2 \frac{\left|(\boldsymbol{\beta}_1)_1 - \alpha_1^*\right|}{|q_1\left(\alpha_1^*\right)|}\right) \\
&\leq |q_1\left(\alpha_1^*\right)| \left(1 + c_2 \frac{\|\boldsymbol{\alpha}_t - \boldsymbol{\alpha}^*\|}{|q_1\left(\alpha_1^*\right)|}\right) \\
&\leq |q_1\left(\alpha_1^*\right)| \left(1 + \theta\right)
\end{aligned}
$$

From Lemma A.10,

$$
\|B\boldsymbol{q}\left(\boldsymbol{\beta}_1\right)\| \geq (1 - \theta) |q_1\left(\alpha_1^*\right)| \|B\boldsymbol{e}_1\|
$$

From Lemma A.11 and $\theta \leq \frac{1}{10}$,

$$
-6\theta \leq 1 - \frac{q_1\left((\boldsymbol{\beta}_1)_1\right) \boldsymbol{e}_1^T B^T Bw\left(\boldsymbol{\beta}_1\right)}{\|Bw\left(\boldsymbol{\beta}_1\right)\|^2} \leq 6\theta
$$

Therefore,

$$
\begin{aligned}
|G_{11}| &\leq 6\left(\frac{1 + \theta}{1 - \theta}\right) \frac{|q_1'\left(\alpha_1^*\right)|\theta}{\|B\boldsymbol{e}_1\||q_1\left(\alpha_1^*\right)|} \\
&\leq \frac{7.5\epsilon |q_1'\left(\alpha_1^*\right)|}{\|B\boldsymbol{e}_1\||q_1\left(\alpha_1^*\right)|} \|\boldsymbol{\alpha}_t - \boldsymbol{\alpha}^*\|, \text{ since } \theta \leq \frac{1}{10}
\end{aligned}
$$

**Case 2** : $i = 1, j \neq 1$

Using Lemma A.8,

$$
|G_{1j}| = \left| \frac{q_j'\left((\boldsymbol{\beta}_1)_j\right)}{\|B\boldsymbol{q}\left(\boldsymbol{\beta}_1\right)\|} \frac{q_1\left((\boldsymbol{\beta}_1)_1\right) \boldsymbol{e}_j^T B^T B\boldsymbol{q}\left(\boldsymbol{\beta}_1\right)}{\|B\boldsymbol{q}\left(\boldsymbol{\beta}_1\right)\|^2} \right|
$$

From Lemma A.9,

$$
\left|q_j'\left((\boldsymbol{\beta}_1)_j\right)\right| \leq c_3 \left|(\boldsymbol{\beta}_1)_j - \alpha_j^*\right| \leq c_3 \left|(\boldsymbol{\alpha}_t)_j - \alpha_j^*\right|, \text{ since } \alpha_j^* = 0
$$

From Lemma A.10,

$$
\|B\boldsymbol{q}\left(\boldsymbol{\beta}_1\right)\| \geq (1 - \theta) |q_1\left(\alpha_1^*\right)| \|B\boldsymbol{e}_1\|
$$

From Lemma A.9,

$$
\begin{aligned}
|q_1\left((\boldsymbol{\beta}_1)_1\right)| &\leq |q_1\left(\alpha_1^*\right)| + c_2 \left|(\boldsymbol{\beta}_1)_1 - \alpha_1^*\right| \\
&= |q_1\left(\alpha_1^*\right)| \left(1 + c_2 \frac{\left|(\boldsymbol{\beta}_1)_1 - \alpha_1^*\right|}{|q_1\left(\alpha_1^*\right)|}\right) \\
&\leq |q_1\left(\alpha_1^*\right)| \left(1 + \theta\right)
\end{aligned}
$$

From Lemma A.11,

$$
\left| \frac{\boldsymbol{e}_j^T B^T B\boldsymbol{q}\left(\boldsymbol{\beta}_1\right)}{\|B\boldsymbol{q}\left(\boldsymbol{\beta}_1\right)\|^2} \right| \leq \frac{(1 + 5\theta)}{|q_1\left(\alpha_1^*\right)|} \frac{\|B\boldsymbol{e}_j\|}{\|B\boldsymbol{e}_1\|}
$$

Therefore,

$$|G_{1j}| \le c_3 \left( \frac{1+\theta}{1-\theta} \right) (1+5\theta) \frac{\|B\mathbf{e}_j\|}{|q_1\left(\alpha_1^*\right)| \|B\mathbf{e}_1\|^2} \left| (\boldsymbol{\alpha}_t)_j - \alpha_j^* \right| \le 2c_3 \frac{\|B\mathbf{e}_j\|}{|q_1\left(\alpha_1^*\right)| \|B\mathbf{e}_1\|^2} \left| (\boldsymbol{\alpha}_t)_j - \alpha_j^* \right|$$

**Case 3** : $i \neq 1, j = 1$

Using Lemma A.8,

$$|G_{i1}| = \left| \frac{q_1'\left((\boldsymbol{\beta}_i)_1\right)}{\|B\mathbf{q}\left(\boldsymbol{\beta}_i\right)\|} \frac{q_i\left((\boldsymbol{\beta}_i)_i\right) \mathbf{e}_1^T B^T B\mathbf{q}\left(\boldsymbol{\beta}_i\right)}{\|B\mathbf{q}\left(\boldsymbol{\beta}_i\right)\|^2} \right|$$

From Lemma A.9,

$$
\begin{aligned}
|q_1'\left((\boldsymbol{\beta}_i)_1\right)| &\le |q_1'\left(\alpha_1^*\right)| + c_3 |(\boldsymbol{\beta}_i)_1 - \alpha_1^*| \\
&= |q_1'\left(\alpha_1^*\right)| \left( 1 + c_3 \frac{|(\boldsymbol{\beta}_i)_1 - \alpha_1^*|}{|q_1'\left(\alpha_1^*\right)|} \right) \\
&\le |q_1'\left(\alpha_1^*\right)| (1+\theta)
\end{aligned}
$$

From Lemma A.10,

$$\|B\mathbf{q}\left(\boldsymbol{\beta}_i\right)\| \ge (1-\theta) |q_1\left(\alpha_1^*\right)| \|B\mathbf{e}_1\|$$

From Lemma A.9,

$$|q_i\left((\boldsymbol{\beta}_i)_i\right)| \le \frac{c_3}{2} \left((\boldsymbol{\beta}_i)_i - \alpha_i^*\right)^2 \le \frac{c_3}{2} \left((\boldsymbol{\alpha}_t)_i - \alpha_i^*\right)^2$$

From Lemma A.11,

$$\left| \frac{\mathbf{e}_1^T B^T B\mathbf{q}\left(\boldsymbol{\beta}_i\right)}{\|B\mathbf{q}\left(\boldsymbol{\beta}_i\right)\|^2} \right| \le \frac{1+5\theta}{|q_1\left(\alpha_1^*\right)|}$$

Therefore,

$$
\begin{aligned}
|G_{i1}| &\le \frac{c_3}{2} \left( \frac{(1+\theta)(1+5\theta)}{1-\theta} \right) \frac{|q_1'\left(\alpha_1^*\right)|}{|q_1\left(\alpha_1^*\right)|^2 \|B\mathbf{e}_1\|} \left((\boldsymbol{\alpha}_t)_i - \alpha_i^*\right)^2 \\
&\le c_3 \frac{|q_1'\left(\alpha_1^*\right)|}{|q_1\left(\alpha_1^*\right)|^2 \|B\mathbf{e}_1\|} \left((\boldsymbol{\alpha}_t)_i - \alpha_i^*\right)^2 \\
&\le c_3 R \frac{|q_1'\left(\alpha_1^*\right)|}{|q_1\left(\alpha_1^*\right)|^2 \|B\mathbf{e}_1\|} |(\boldsymbol{\alpha}_t)_i - \alpha_i^*| \\
&\le c_2 \frac{|q_1'\left(\alpha_1^*\right)|}{|q_1\left(\alpha_1^*\right)|^2 \|B\mathbf{e}_1\|} |(\boldsymbol{\alpha}_t)_i - \alpha_i^*|
\end{aligned}
$$

**Case 4** : $i \neq 1, j \neq 1, i = j$

Using Lemma A.8,

$$|G_{ii}| = \left| \frac{q_i'\left((\boldsymbol{\beta}_i)_i\right)}{\|B\mathbf{q}\left(\boldsymbol{\beta}_i\right)\|} \left[ 1 - \frac{q_i\left((\boldsymbol{\beta}_i)_i\right) \mathbf{e}_i^T B^T B\mathbf{q}\left(\boldsymbol{\beta}_i\right)}{\|B\mathbf{q}\left(\boldsymbol{\beta}_i\right)\|^2} \right] \right|$$

From Lemma A.9,

$$|q_i'\left((\boldsymbol{\beta}_i)_i\right)| \le c_3 \left|(\boldsymbol{\beta}_i)_i - \alpha_j^*\right| \le c_3 |(\boldsymbol{\alpha}_t)_i - \alpha_i^*|$$

From Lemma A.10,

$$\|B\mathbf{q}\left(\boldsymbol{\beta}_i\right)\| \ge (1-\theta) |q_1\left(\alpha_1^*\right)| \|B\mathbf{e}_1\|$$

From Lemma A.9,

$$|q_i\left((\boldsymbol{\beta}_i)_i\right)| \le \frac{c_3}{2} \left((\boldsymbol{\beta}_i)_i - \alpha_i^*\right)^2 \le \frac{c_3}{2} \left((\boldsymbol{\alpha}_t)_i - \alpha_i^*\right)^2 \le c_3 R \|\boldsymbol{\alpha}_t - \boldsymbol{\alpha}^*\| \le c_2 \|\boldsymbol{\alpha}_t - \boldsymbol{\alpha}^*\|$$

From Lemma A.11,

$$\left| \frac{\boldsymbol{e}_i^T B^T B \boldsymbol{q} \left( \boldsymbol{\beta}_i \right)}{\| B \boldsymbol{q} \left( \boldsymbol{\beta}_i \right) \|^2} \right| \le (1 + 5\theta) \frac{\| B \boldsymbol{e}_i \|}{|q_1 \left( \alpha_1^* \right)| \, \| B \boldsymbol{e}_1 \|}$$

Then,

$$
\begin{aligned}
\frac{q_i \left( (\boldsymbol{\beta}_i)_i \right) \boldsymbol{e}_i^T B^T B \boldsymbol{q} \left( \boldsymbol{\beta}_i \right)}{\| B \boldsymbol{q} \left( \boldsymbol{\beta}_i \right) \|^2} &\le (1 + 5\theta) \| \boldsymbol{\alpha}_t - \boldsymbol{\alpha}^* \| \frac{\| B \boldsymbol{e}_i \|}{\| B \boldsymbol{e}_1 \|} \frac{c_2}{|q_1 \left( \alpha_1^* \right)|} \\
&\le (1 + 5\theta) \| \boldsymbol{\alpha}_t - \boldsymbol{\alpha}^* \| \frac{\| B \|_F}{\| B \boldsymbol{e}_1 \|} \frac{c_2}{|q_1 \left( \alpha_1^* \right)|} \\
&\le (1 + 5\theta) \epsilon \| \boldsymbol{\alpha}_t - \boldsymbol{\alpha}^* \| \\
&\le 2\theta
\end{aligned}
$$

Therefore,

$$|G_{ii}| \le \frac{c_3 \, |(\boldsymbol{\alpha}_t)_i - \alpha_i^*|}{(1 - \theta) \, |q_1 \left( \alpha_1^* \right)| \, \| B \boldsymbol{e}_1 \|} \le \frac{2 c_3}{|q_1 \left( \alpha_1^* \right)| \, \| B \boldsymbol{e}_1 \|} \, |(\boldsymbol{\alpha}_t)_i - \alpha_i^*|$$

**Case 5** : $i \ne 1, j \ne 1, i \ne j$

Using Lemma A.8,

$$|G_{ij}| = \left| \frac{q_j' \left( (\boldsymbol{\beta}_i)_j \right)}{\| B \boldsymbol{q} \left( (\boldsymbol{\beta}_i) \right) \|} \frac{q_i \left( (\boldsymbol{\beta}_i)_i \right) \boldsymbol{e}_j^T B^T B \boldsymbol{q} \left( (\boldsymbol{\beta}_i) \right)}{\| B \boldsymbol{q} \left( (\boldsymbol{\beta}_i) \right) \|^2} \right|$$

From Lemma A.9,

$$
\begin{aligned}
\left| q_j' \left( (\boldsymbol{\beta}_i)_j \right) \right| &\le c_3 \left| (\boldsymbol{\beta}_i)_j - \alpha_j^* \right| \\
&\le c_3 \left| (\boldsymbol{\alpha}_t)_j - \alpha_j^* \right|
\end{aligned}
$$

From Lemma A.10,

$$\| B \boldsymbol{q} \left( \boldsymbol{\beta}_i \right) \| \ge (1 - \theta) \, |q_1 \left( \alpha_1^* \right)| \, \| B \boldsymbol{e}_1 \|$$

From Lemma A.9,

$$|q_i \left( (\boldsymbol{\beta}_i)_i \right)| \le \frac{c_3}{2} \left( (\boldsymbol{\beta}_i)_i - \alpha_i^* \right)^2 \le \frac{c_3}{2} \left( (\boldsymbol{\alpha}_t)_i - \alpha_i^* \right)^2 \le \frac{c_3 R}{2} |(\boldsymbol{\alpha}_t)_i - \alpha_i^*|$$

From Lemma A.11,

$$\left| \frac{\boldsymbol{e}_j^T B^T B \boldsymbol{q} \left( \boldsymbol{\beta}_i \right)}{\| B \boldsymbol{q} \left( \boldsymbol{\beta}_i \right) \|^2} \right| \le \frac{(1 + 5\theta)}{|q_1 \left( \alpha_1^* \right)|} \frac{\| B \boldsymbol{e}_j \|}{\| B \boldsymbol{e}_1 \|}$$

Therefore,

$$|G_{ij}| \le \frac{c_3^2 R}{2} \left( \frac{1 + 5\theta}{1 - \theta} \right) \frac{\| B \boldsymbol{e}_j \|}{|q_1 \left( \alpha_1^* \right)|^2 \, \| B \boldsymbol{e}_1 \|^2} \left| (\boldsymbol{\alpha}_t)_j - \alpha_j^* \right| |(\boldsymbol{\alpha}_t)_i - \alpha_i^*| \le \frac{c_3^2 R \, \| B \boldsymbol{e}_j \|}{|q_1 \left( \alpha_1^* \right)|^2 \, \| B \boldsymbol{e}_1 \|^2} \left| (\boldsymbol{\alpha}_t)_j - \alpha_j^* \right| |(\boldsymbol{\alpha}_t)_i - \alpha_i^*|$$

### A.1.2.3  Putting everything together

Putting all the cases together in Eq A.45, we have $\frac{\| \boldsymbol{\alpha}_{t+1} - \boldsymbol{\alpha}^* \|}{\| \boldsymbol{\alpha}_t - \boldsymbol{\alpha}^* \|} \le C \, \| \boldsymbol{\alpha}_t - \boldsymbol{\alpha}^* \|$, where

$$
C := \left[
\begin{aligned}
& \left( \frac{7.5 \epsilon \, |q_1' \left( \alpha_1^* \right)|}{\| B \boldsymbol{e}_1 \| \, |q_1 \left( \alpha_1^* \right)|} \right)^2 + \left( \frac{2 c_3 \, \| B \|_F}{|q_1 \left( \alpha_1^* \right)| \, \| B \boldsymbol{e}_1 \|^2} \right)^2 + \left( \frac{c_2 \, |q_1' \left( \alpha_1^* \right)|}{|q_1 \left( \alpha_1^* \right)|^2 \, \| B \boldsymbol{e}_1 \|} \right)^2 + \left( \frac{2 c_3}{|q_1 \left( \alpha_1^* \right)| \, \| B \boldsymbol{e}_1 \|} \right)^2 \\
& + \left( \frac{c_3^2 R^2 \, \| B \|_F}{|q_1 \left( \alpha_1^* \right)|^2 \, \| B \boldsymbol{e}_1 \|^2} \right)^2
\end{aligned}
\right]^{1/2}
$$

Then we have, $C \le \frac{5 \| B \|_F}{\| B \boldsymbol{e}_1 \|^2} \max \left\{ \frac{c_3}{|q_1 (\alpha_1^*)|}, \frac{c_2^2}{|q_1 (\alpha_1^*)|^2} \right\}$. For linear convergence, we require $CR < 1$, which is ensured by the condition mentioned in Theorem 4.

## A.2 Additional experiments for noisy ICA

### A.2.1 Experiments with super-gaussian source signals

Table 2 shows the Amari error in the presence of super-Gaussian signals. The signals are 1) a Uniform distribution $U(-\sqrt{3}, \sqrt{3})$, 2) Bernoulli$\left(\frac{1}{2} + \frac{1}{\sqrt{12}}\right)$, 3) Laplace(0, 0.05) (mean 0 and standard deviation 0.05), 4) Exponential(5), and 5) and 6) a Student's $t$-distribution with 3 and 5 degrees of freedom, respectively. The Meta algorithm closely follows the best candidate algorithm even in the presence of many super-Gaussian signals.

Table 2: Variation of Amari error with Sample Size for Heavy-Tailed distributions, averaged over 100 random runs. Noise power $\rho = 0.001$, number of sources $k = 6$.

| Algorithm $\quad n$ | 200 | 500 | 1000 | 5000 | 10000 |
|---|---|---|---|---|---|
| Meta | **0.44376** | **0.25215** | **0.20222** | **0.11435** | **0.0838** |
| CHF | 1.10103 | 0.70823 | 0.52529 | 0.21344 | **0.09194** |
| CGF | 1.84266 | 1.51216 | 1.3702 | 0.84351 | 0.58753 |
| PEGI | 2.27873 | 1.86561 | 1.71474 | 1.33709 | 1.23322 |
| PFICA | **0.39237** | **0.25468** | **0.21222** | **0.12878** | 0.12347 |
| JADE | 0.70174 | 0.39246 | 0.28199 | 0.12652 | 0.09686 |
| FastICA | 0.66441 | 0.3869 | 0.28419 | 0.13215 | 0.09946 |

### A.2.2 Image demixing experiments

In this section, we provide additional experiments for image-demixing using ICA. We mix images using flattening and linearly mixing them with a $4 \times 4$ matrix $B$ (i.i.d entries $\sim \mathcal{N}(0, 1)$) and Wishart noise ($\rho = 0.001$). Demixing is performed using the SINR-optimal demixing matrix (see Section 2) and the results are shown in Figure A.1 along with their corresponding independence scores. The CHF-based method recovers the original sources well, upto sign. The Kurtosis and CGF-based method fails to recover the second source. This is consistent with their higher independence score. The Meta algorithm selects CHF from candidates CHF, CGF, Kurtosis, FastICA, and JADE.

### A.2.3 Image denoising experiments

In this experiment, we use the ICA-based denoising technique proposed in [38] to compare candidate Noisy ICA algorithms and show that the Meta algorithm can pick the best-denoised image based on the independence score proposed in our work.

We use the noisy MNIST dataset and further add entrywise Gaussian noise with variance proportional to $\sin^2(c_1\pi(i + j))(c_1 = 50)$ to the $(i, j)^{\text{th}}$ pixel. Training images are flattened to create a 784-dimensional vector, and PCA is performed to reduce dimensionality to 25, on which subsequently ICA is performed. The original and denoised images along with their independence score are shown in Figure A.2. We note that CHF-based denoising provides qualitatively better results, while CGF-based denoising provides the worst results. This is consistent with their corresponding independence scores.

## A.3 Algorithm for sequential calculation of independence scores

In this section, we provide an algorithm for the computation of the independence score proposed in the manuscript, but in a sequential manner. The detailed algorithm is provided as Algorithm 3. We assume that we have access to a matrix of the form $C = BDB^T$.

The power method (see Eq 3) essentially extracts one column of $B$ (up to scaling) at every step. [49] provides an elegant way to use the pseudo-Euclidean space to successively project the data onto columns orthogonal to the ones extracted. This enables one to extract each column of the mixing matrix. For completeness, we present the steps of this projection. After extracting the $i^{th}$

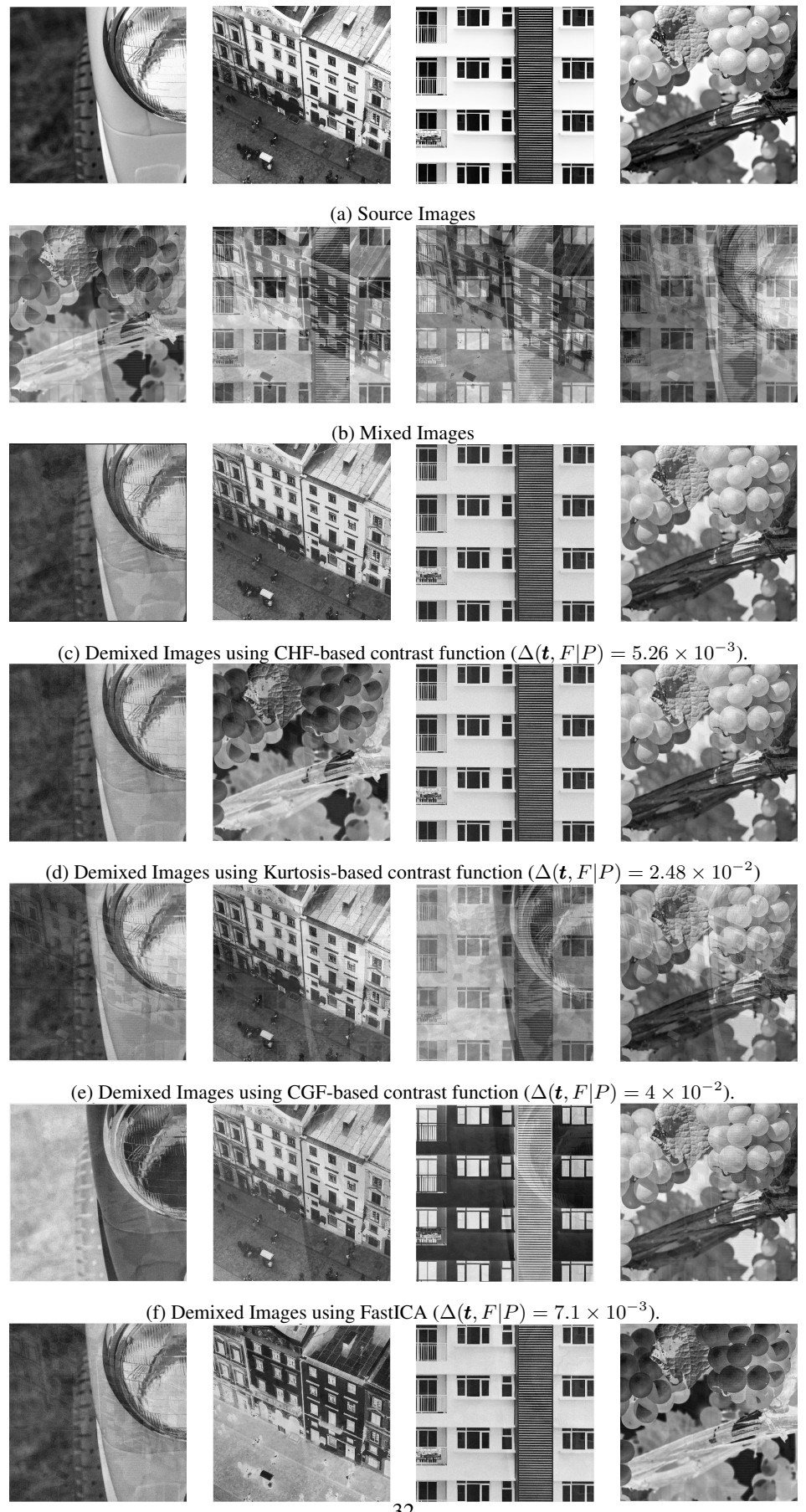

(a) Source Images

(b) Mixed Images

(c) Demixed Images using CHF-based contrast function ($\Delta(\boldsymbol{t}, F|P) = 5.26 \times 10^{-3}$).

(d) Demixed Images using Kurtosis-based contrast function ($\Delta(\boldsymbol{t}, F|P) = 2.48 \times 10^{-2}$)

(e) Demixed Images using CGF-based contrast function ($\Delta(\boldsymbol{t}, F|P) = 4 \times 10^{-2}$).

(f) Demixed Images using FastICA ($\Delta(\boldsymbol{t}, F|P) = 7.1 \times 10^{-3}$).

(g) Demixed Images using JADE ($\Delta(\boldsymbol{t}, F|P) = 6.8 \times 10^{-3}$).

Figure A.1: Image-Demixing using ICA

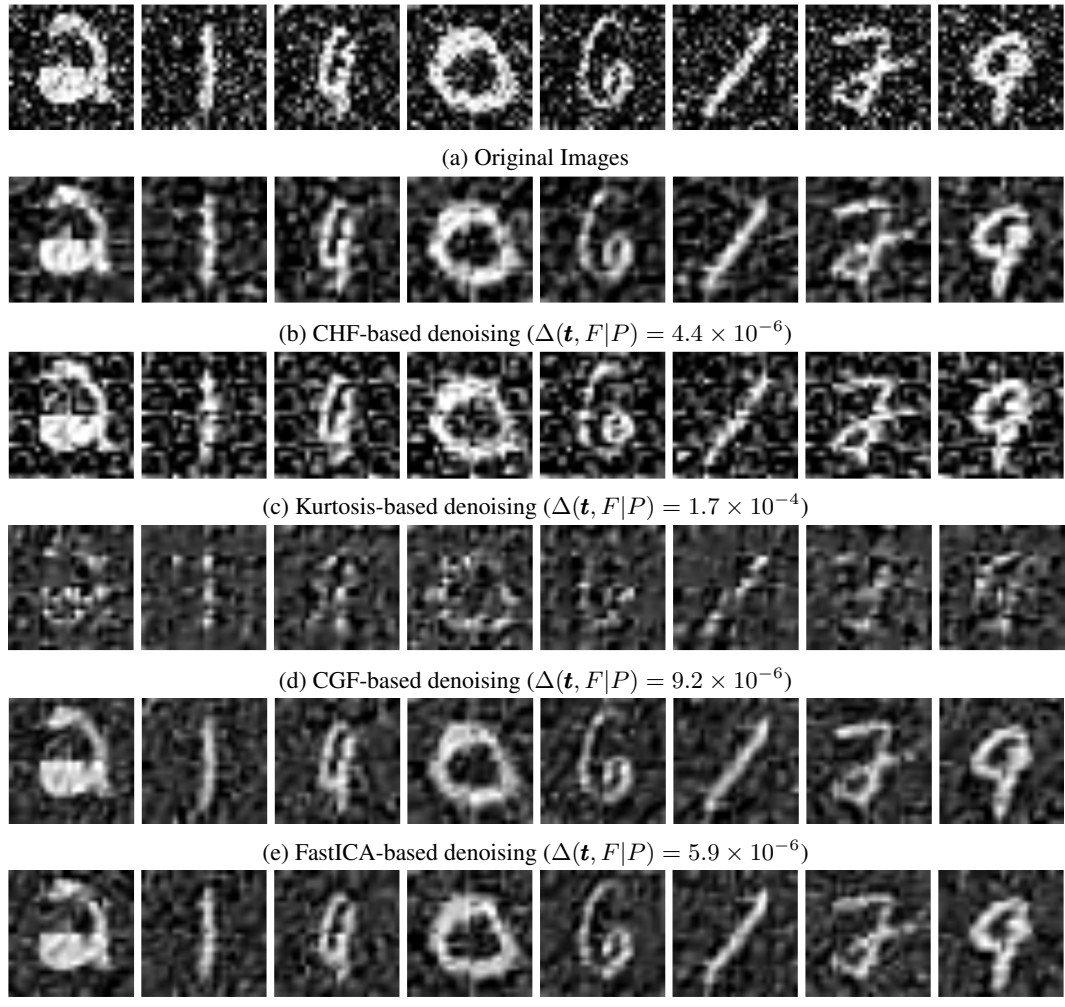

(a) Original Images

(b) CHF-based denoising ($\Delta(\boldsymbol{t}, F|P) = 4.4 \times 10^{-6}$)

(c) Kurtosis-based denoising ($\Delta(\boldsymbol{t}, F|P) = 1.7 \times 10^{-4}$)

(d) CGF-based denoising ($\Delta(\boldsymbol{t}, F|P) = 9.2 \times 10^{-6}$)

(e) FastICA-based denoising ($\Delta(\boldsymbol{t}, F|P) = 5.9 \times 10^{-6}$)

(f) JADE-based denoising ($\Delta(\boldsymbol{t}, F|P) = 4.6 \times 10^{-6}$)

Figure A.2: Image Denoising using ICA

column $\boldsymbol{u}^{(i)}$, the algorithm maintains two matrices. The first, denoted by $U$, estimates the mixing matrix $B$ one column at a time (up to scaling). The second, denoted by $V$ estimates $B^{-1}$ *one row* at a time. It is possible to extend the independence score in Eq 2 to the sequential setting as follows. Let $\boldsymbol{u}^{(j)}$ be the $j^{th}$ vector extracted by the power method (Eq 3) until convergence. After extracting $\ell$ columns, we project the data using $\min(\ell + 1, k)$ projection matrices. These would be mutually independent if we indeed extracted different columns of $B$. Let $C = BDB^T$ for some diagonal matrix $D$. For convenience of notation, denote $B_i \equiv B(:, i)$. Set -

$$\boldsymbol{v}^{(j)} = \frac{C^\dagger \boldsymbol{u}^{(j)}}{\boldsymbol{u}^{(j)T} C^\dagger \boldsymbol{u}^{(j)}} \tag{A.46}$$

When $\boldsymbol{u}^{(j)} = B_j / \|B_j\|$, $\boldsymbol{v}^{(j)} = \dfrac{(B^T)^{-1} D^\dagger \boldsymbol{e}_j}{\boldsymbol{e}_j^T D^\dagger \boldsymbol{e}_j} \|B_j\| = (B^T)^{-1} \boldsymbol{e}_j \|B_j\|$. Let $\boldsymbol{x}$ denote an arbitrary datapoint. Thus

$$U(:, j)V(j, :)\boldsymbol{x} = B_j z_j + U(:, j)V(j, :)\boldsymbol{g} \tag{A.47}$$

Thus the projection on all other columns $j > \ell$ is given by:

$$(I - UV)\boldsymbol{x} = \sum_{i=1}^{k} B_i z_i - \sum_{j=1}^{\ell} B_j z_j + \tilde{\boldsymbol{g}} = \sum_{j=\ell+1}^{k} B_j z_j + \tilde{\boldsymbol{g}}$$

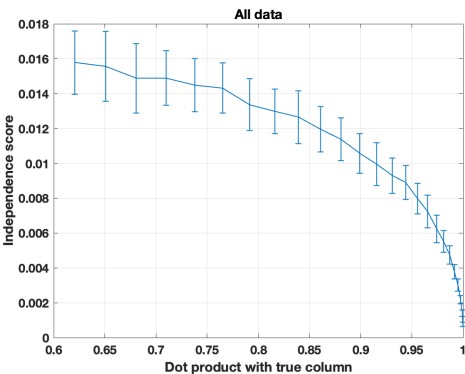
Figure A.3: Mean independence score with errorbars from 50 random runs

where $\tilde{\boldsymbol{g}} = F\boldsymbol{g}$, where $F$ is some $k \times k$ matrix. So we have $\min(\ell, k)$ vectors of the form $z_i B_i + \tilde{\boldsymbol{g}}_i$, and when $\ell < k$ an additional vector which contains all independent random variables $z_j$, $j > \ell$ along with a mean-zero Gaussian vector. Then we can project each vector to a scalar using unit random vectors and check for independence using the Independence Score defined in 2. When $\ell = k$, then all we need are the $j = 1, \ldots, k$ projections on the $k$ directions identified via ICA in Eq A.47.

As an example, we conduct an experiment (Figure A.3), where we fix a mixing matrix $B$ using the same generating mechanism in Section 5. Now we create vectors $\mathbf{q}$ which interpolate between $B(:, 1)$ and a fixed arbitrary vector orthogonal to $B(:, 1)$. As the interpolation changes, we plot the independence score for direction $\mathbf{q}$. The dataset is the 9-dimensional dataset, which has independent components from many different distributions (see Section 5).

This plot clearly shows that there is a clear negative correlation between the score and the dot product of a vector with the column $B(:, 1)$. To be concrete, when $\mathbf{q}$ has a small angle with $B(:, 1)$, the independence score is small, and the error bars are also very small. However, as the dot product decreases, the score grows and the error bars become larger.

---

**Algorithm 3** Independence Score after extracting $\ell$ columns. $U$ and $V$ are running estimates of $B$ and $B^{-1}$ upto $\ell$ columns and rows respectively.

---

**Input**
$\qquad X \in \mathbb{R}^{n \times k}, U \in \mathbb{R}^{k \times \ell}, V \in \mathbb{R}^{\ell \times k}$, Number of random projections $M$
$k_0 \leftarrow \min(\ell + 1, k)$
**for** $j$ in range$[1, \ell]$ **do**
$\qquad Y_j \leftarrow X \left(U(:, j)V(j, :)\right)^T$
**end for**
**if** $\ell < k$ **then**
$\qquad Y_{\ell+1} \leftarrow X \left(I - V^T U^T\right)$
**end if**
**for** $j$ in range$[1, M]$ **do**
$\qquad \boldsymbol{t} \leftarrow$ random unit vector in $\mathbb{R}^k$
$\qquad$ **for** $a$ in range$[1, k_0]$ **do**
$\qquad\qquad W(:, j) \leftarrow Y_j \boldsymbol{t}$
$\qquad$ **end for**
$\qquad$ Let $\boldsymbol{\alpha} \in \mathbb{R}^{1, k_0}$ represent a row of $W$
$\qquad \tilde{S} \leftarrow \mathrm{cov}(W)$
$\qquad \gamma \leftarrow \sum_{i,j} \tilde{S}_{ij}$
$\qquad \boldsymbol{\beta} \leftarrow \sum_i \tilde{S}_{ii}$
$\qquad s(j) = |\hat{\mathbb{E}} \exp(\sum_j i\alpha_j) \exp(-\boldsymbol{\beta}) - \prod_{j=1}^{k_0} \hat{\mathbb{E}} \exp(i\alpha_j) \exp(-\gamma)|$
**end for**
Return $\mathrm{mean}(s), \mathrm{stdev}(s)$

---

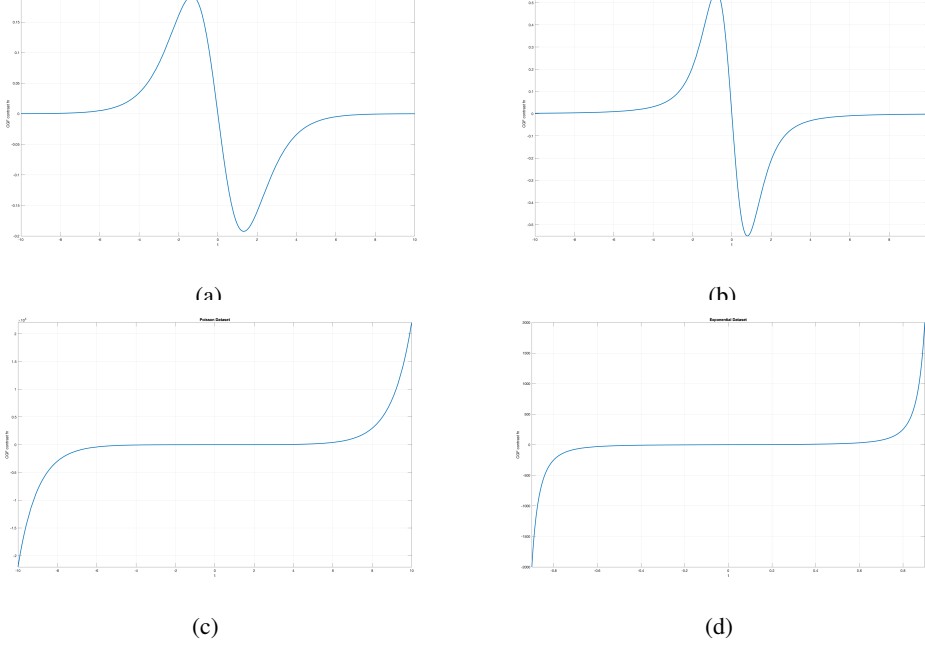

Figure A.4: Plots of the third derivative of the CGF-based contrast function for different datasets - Bernoulli($p = 0.5$) A.4a, Uniform($\mathcal{U}\left(-\sqrt{3}, \sqrt{3}\right)$) A.4b, Poisson($\lambda = 1$) A.4c and Exponential($\lambda = 1$) A.4d. Note that the sign stays the same in each half-line

We refer to this function as $\Delta\left(X, U, V, \ell\right)$ which takes as input, the data $X$, the number of columns $\ell$ and matrices $U, V$ which are running estimates of $B$ and $B^{-1}$ upto $\ell$ columns and rows respectively.

## A.4   More details for third derivative condition in theorem 3

Theorem 3 requires the condition "The third derivative of $h_X(u)$ does not change the sign in the half line $[0, \infty)$ for the non-Gaussian random variable considered in the ICA problem.". In this section, we provide sufficient conditions with respect to contrast functions and datasets where this holds. Further, we also provide an interesting example to demonstrate that our Assumption 1-(d) might not be too far from being necessary.

**CGF-based contrast function.** Consider the cumulant generating function of a random variable $X$, i.e. the logarithm of the moment generating function. Now consider the contrast function

$$g_X(t) = CGF(t) - \text{var}(X)t^2/2.$$

We first note that it satisfies Assumption 1(a)-(d). Next, we observe that it is enough for a distribution to have all cumulants of the same sign to satisfy Assumption 1(d) for the CGF. For example, the Poisson distribution has all positive cumulants. Figure A.4 depicts the third derivative of the contrast function for a Bernoulli(1/2), Uniform, Poisson, and Exponential.

**Logarithm of the symmetrized characteristic function.** Figure A.5 depicts the third derivative of this contrast function for the logistic distribution where Assumption 1(d) holds. Although the Assumption does not hold for a large class of distributions here, we believe that the notion of having the same sign can be relaxed up to some bounded parameter values instead of the entire half line, so that the global convergence results of Theorem 3 still hold. This belief is further reinforced by Figure A.6 which shows that the loss landscape of functions where Assumption 1(d) doesn't hold is still smooth.

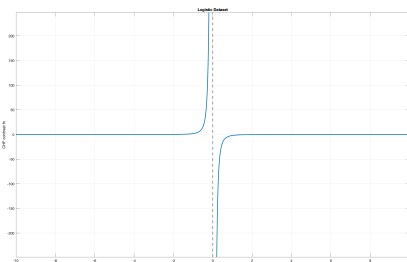

Figure A.5: Plots of the third derivative of the CHF-based contrast function for Logistic$(0, 1)$ dataset.

### A.4.1 Towards a necessary condition

Consider the probability distribution function (see [36]) $f(x) = ke^{-\frac{x^2}{2}}(a + b\cos(c\pi x))$ for constants $a, b, c > 0$. For $f(x)$ to be a valid pdf, we require that $f(x) \geq 0$, and $\int_{-\infty}^{\infty} f(x)\, dx = 1$. Therefore, we have,

$$\int_{-\infty}^{\infty} ke^{-\frac{x^2}{2}}(a + b\cos(c\pi x))\, dx = 1 \tag{A.48}$$

Noting standard integration results, we have that,

$$\int_{-\infty}^{\infty} e^{-\frac{x^2}{2}}\, dx = \sqrt{2\pi} \quad \text{and} \quad \int_{-\infty}^{\infty} e^{-\frac{x^2}{2}}\cos(c\pi x)\, dx = \sqrt{2\pi}e^{-\frac{c^2\pi^2}{2}} \tag{A.49}$$

Therefore, $k = \dfrac{1}{\sqrt{2\pi}(a+be^{-\frac{c^2\pi^2}{2}})}$. Some algebraic manipulations yield the following:

$$MGF_{f(x)}(t) = \frac{1}{(a + be^{-\frac{c^2\pi^2}{2}})}e^{\frac{t^2}{2}}\left(a + be^{-\frac{c^2\pi^2}{2}}\cos(c\pi t)\right)$$

Therefore, $\ln(MGF_{f(x)}(t)) = -\ln(a + be^{-\frac{c^2\pi^2}{2}}) + \frac{t^2}{2} + \ln(a + be^{-\frac{c^2\pi^2}{2}}\cos(c\pi t))$.

We now compute the mean, $\mu$, and variance $\sigma^2$.

The mean, $\mu$ can be given as :

$$\mu = \int_{-\infty}^{\infty} f(x)x\, dx = 0 \text{ since } f(x) \text{ is an even function.}$$

The variance, $\sigma^2$ can therefore be given as :

$$\sigma^2 = k\left(a\int_{-\infty}^{\infty} e^{-\frac{x^2}{2}}x^2\, dx + b\int_{-\infty}^{\infty} e^{-\frac{x^2}{2}}x^2\cos(c\pi x)\, dx\right)$$

Now, we note that $\int_{-\infty}^{\infty} e^{-\frac{x^2}{2}}x^2\, dx = \sqrt{2\pi}$ and $\int_{-\infty}^{\infty} e^{-\frac{x^2}{2}}x^2\cos(c\pi x)\, dx = e^{-\frac{c^2\pi^2}{2}}\sqrt{2\pi}(1 - c^2\pi^2)$
Therefore,

$$\sigma^2 = k\left(a\sqrt{2\pi} + be^{-\frac{c^2\pi^2}{2}}\sqrt{2\pi}(1 - c^2\pi^2)\right) = 1 - \frac{bc^2\pi^2 e^{-\frac{c^2\pi^2}{2}}}{a + be^{-\frac{c^2\pi^2}{2}}}$$

The symmetrized CGF can therefore be written as

$$\operatorname{sym}CGF(t) := \ln(MGF_{f(x)}(t)) + \ln(MGF_{f(x)}(-t))$$
$$= -2\ln(a + be^{-\frac{c^2\pi^2}{2}}) + t^2 + 2\ln(a + be^{-\frac{c^2\pi^2}{2}}\cos(c\pi t))$$

Therefore, $g(t) = \operatorname{sym}CGF(t) - (1 - \frac{bc^2\pi^2 e^{-\frac{c^2\pi^2}{2}}}{a+be^{-\frac{c^2\pi^2}{2}}})t^2$ can be written as :

$$g(t) = -2\ln(a + be^{-\frac{c^2\pi^2}{2}}) + 2\ln(a + be^{-\frac{c^2\pi^2}{2}}\cos(c\pi t)) + \frac{bc^2\pi^2 e^{-\frac{c^2\pi^2}{2}}}{a + be^{-\frac{c^2\pi^2}{2}}}t^2$$

Finally, $g'(t)$ can be written as :

$$g'(t) = \frac{d}{dt}\left(2\ln(a + be^{-\frac{c^2\pi^2}{2}}\cos(c\pi t)) + \frac{bc^2\pi^2 e^{-\frac{c^2\pi^2}{2}}}{a + be^{-\frac{c^2\pi^2}{2}}}t^2\right)$$

$$= -\frac{2bc\pi e^{-\frac{c^2\pi^2}{2}}\sin(c\pi t)}{a + be^{-\frac{c^2\pi^2}{2}}\cos(c\pi t)} + \frac{2bc^2\pi^2 e^{-\frac{c^2\pi^2}{2}}}{a + be^{-\frac{c^2\pi^2}{2}}}t$$

$g''(t)$ can be written as :

$$g''(t) = -2bc^2\pi^2 e^{-\frac{c^2\pi^2}{2}}\frac{a\cos(c\pi t) + be^{-\frac{c^2\pi^2}{2}}}{(a + be^{-\frac{c^2\pi^2}{2}}\cos(c\pi t))^2} + \frac{2bc^2\pi^2 e^{-\frac{c^2\pi^2}{2}}}{a + be^{-\frac{c^2\pi^2}{2}}}$$

and $g'''(t)$ can be evaluated as:

$$g'''(t) = 2bc^3\pi^3 e^{-\frac{c^2\pi^2}{2}}\sin(c\pi t)\frac{\left(a^2 - 2b^2 e^{-c^2\pi^2} - abe^{-\frac{c^2\pi^2}{2}}\cos(c\pi t)\right)}{(a + be^{-\frac{c^2\pi^2}{2}}\cos(c\pi t))^3}$$

Lets set $a = 2, b = -1, c = \frac{4}{\pi}$. Then, we have that the pdf, $f(x) = ke^{-\frac{x^2}{2}}(2 - \cos(4x)) = ke^{-\frac{x^2}{2}}(1 + 2\sin^2(2x))$. The corresponding functions are :

$$E[\exp(tX)] = \frac{1}{(2 - e^{-8})}e^{\frac{t^2}{2}}\left(2 - e^{-8}\cos(4t)\right)$$

$$g(t) := \mathrm{sym}\,CGF(t) - \mathrm{var}(X)t^2 = -2\ln(2 - e^{-8}) + 2\ln(2 - e^{-8}\cos(4t)) - \frac{16e^{-8}}{2 - e^{-8}}t^2$$

$$g'(t) = \frac{8e^{-8}\sin(4t)}{2 - e^{-8}\cos(4t)} - \frac{32e^{-8}}{2 - e^{-8}}t$$

$$g''(t) = 32e^{-8}\frac{2\cos(4t) - e^{-8}}{(2 - e^{-8}\cos(4t))^2} - \frac{32e^{-8}}{2 - e^{-8}}$$

$$g'''(t) = -128e^{-8}\sin(4t)\frac{(4 - 2e^{-16} + 2e^{-8}\cos(4t))}{(2 - e^{-8}\cos(4t))^3}$$

**Closeness to a Gaussian MGF:** It can be seen there that $g'''(t)$ changes sign in the half-line. However, the key is to note that the MGF of this distribution is very close to that of a Gaussian and can be made arbitrarily small by varying the parameters $a$ and $b$. This renders the CGF-based contrast function ineffectual. This leads us to believe that Assumption 1(d) is not far from being necessary to ensure global optimality of the corresponding objective function.

## A.5  Surface plots of the CHF-based contrast functions

Figure A.6 depicts the loss landscape of the Characteristic function (CHF) based contrast function described in Section 3.3 of the manuscript. We plot the value of the contrast function evaluated at $B^{-1}\boldsymbol{u}, \boldsymbol{u} = \frac{1}{\sqrt{x^2+y^2}}\begin{pmatrix} x \\ y \end{pmatrix}$ for $x, y \in [-1, 1]$. As shown in the figure. We have rotated the data so that the columns of $B$ align with the $X$ and $Y$ axes. The global maxima occur at $\boldsymbol{u}$ aligned with the columns of $B$.

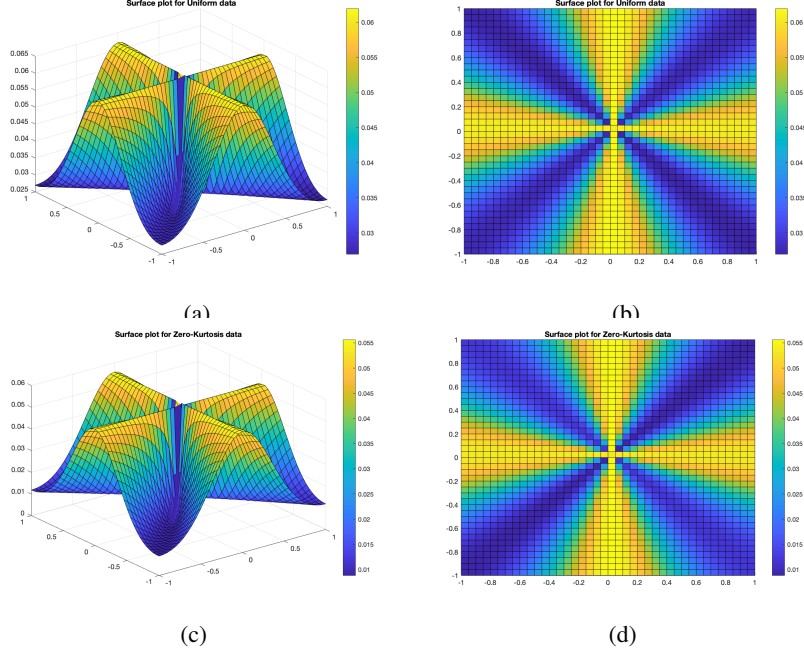

(a)          (b)

(c)          (d)

Figure A.6: Surface plots for zero-kurtosis (A.6c and A.6d) and Uniform (A.6a, A.6b) data with $n = 10000$ points, noise-power $\rho = 0.1$ and number of source signals, $k = 2$.

