# OpenReview forum: "Nonparametric Evaluation of Noisy ICA Solutions"
_NeurIPS.cc/2024/Conference — NeurIPS 2024 poster_

### Official Review · Reviewer_uqTT · 2024-07-06

**Soundness:** 2
**Presentation:** 1
**Contribution:** 2
**Rating:** 4
**Confidence:** 3

**Summary:**

The authors consider the noisy ICA problem. They first propose a numerical objective function that can be used as a guide to assess the quality of an existing ICA algorithm. This function is not suitable for optimization and therefore, the authors propose to use it in a meta-algorithm, where the purpose is to select the best ICA solutions out of several, where each solution is possibly produced by a separate algorithm. Then, they propose new contrast functions that are specifically suited for the noisy ICA problem. Finally, they study critical points of contrast functions so as to validate their use in noisy ICA problems. They provide numerical experiments to back up their claims.

**Strengths:**

The paper consider the noisy ICA problem, which is a challenging problem and is arguably more realistic than the noise-free ICA problem. The paper also demonstrates via numerical experiments that the proposed meta algorithm and the contrast functions can uncover the components in an image mixing problem.

**Weaknesses:**

The paper touches on too many things and is not very readable. For instance, I don't see the connection between the meta algorithm and the new contrast functions. Similarly, I would have welcomed more detail on the development of the meta algorithm, and the choice of randomization in its computation.
Occasionally, something is introduced out of nowhere and it's hard to understand the motivation. For instance, I don't see the development for the new contrast functions.
Overall, I'd have preferred a paper that coherently and clearly explores a single idea. Currently, it reads like a collection of not clearly-developed ideas.

**Questions:**

- line 123, "We propose to adapt the CHFICA objective using estimable parameters to the noisy ICA setting" : I don't understand this sentence. What are the "estimable parameters" here? Is it $S$?
- eqn 2 : What's the motivation for the additional terms (the second factors in both "JOINT" and "PRODUCT")?
- eqn 2 : In the last term, is a $t$ missing?
- Thm 2 : How does knowing this bound help? Please discuss how this theorem is useful in the text.
- line 175, "...in this section them, ..." : typo?
- line 214 : What is a pseudo-Euclidean space?

---

> ### Author Rebuttal · Authors · 2024-08-07
>
> Thank you for acknowledging the challenging nature and wider applicability of the noisy ICA problem and for your kind words regarding our experiments with the Meta algorithm. All typographical and grammatical errors will be corrected in the revised manuscript and are not addressed individually here.
>
> **[Re: Connection between the meta-algorithm and the new contrast functions]** - Most methods for noisy ICA center around designing an appropriate contrast function. Lines 87-103 in the paper, under “Individual shortcomings of different contrast functions and fitting strategies,” explain the different issues with different types of existing contrast functions. As explained in lines 104-110, this leads to two challenges: a) how do we adaptively pick an appropriate contrast function for the dataset at hand, and b) can we design better contrast functions? Our paper gives a full pipeline for a new adaptive algorithm for choosing from a set of contrast functions, which includes established ones and the new ones we introduce.
>
> **[Re: Development of the meta-algorithm and the choice of randomization in its computation]**
> Lines 115-124 build intuition for the score used in the Meta algorithm using the noiseless case. For the noisy case, one needs to account for the characteristic function of the Gaussian noise, whose covariance matrix is unknown. The second terms in JOINT and PRODUCT essentially cancel out the additional terms using \textit{estimable} parameters, i.e. the covariance matrix S of the data (estimated via the sample covariance).
>
> Lines 148-151 discuss why we choose not to use the independence score as an optimization objective but instead use it to choose the best algorithm among candidates. This leads to the meta-algorithm in Algorithm 1.
>
> Remark 2 explains the choice of randomization in the Meta algorithm. Essentially, all related papers that use the characteristic functions in ICA or for Gaussianity testing (see the citations in Remark 2, line 155) average over $t$ sampled from a spherical Gaussian.
>
> **[Re: Development of contrast functions]** Section 3.2 (lines 163-168) provides the mathematical properties useful in a contrast function for designing provable optimization algorithms for ICA. Lines 169-172 explain what each of these properties implies. Section 3.3 then develops the new contrast functions that satisfy these important properties.
>
> **[Re: Estimable parameters]** Yes, S is the estimable parameter here. We will clarify it in the revised manuscript.
>
> **[Re: Second-factors in JOINT and PRODUCT]** See the response on the development of the meta-algorithm above.
>
> **[Re: Bound in Theorem 2]:** This bound shows that uniformly over $F$, the empirical average $\mathbb{E}_{\mathbf{t}\in \mathcal{N}(0,I_k)}\Delta(\mathbf{t},F|\hat{P})$ is close to the population score. This guarantees that as long as the difference between the population scores of the two candidate algorithms is not too small, the meta-algorithm can pick up the better score. We will clarify this in the manuscript.
>
> **[Re: Pseudo-Euclidean Space]** A pseudo-Euclidean space is a generalization of Euclidean space used by Voss et. al. where the produce between vectors $u, v \in \mathbb{R}^{d}$ is given as $u^T A v$ where $A$ does not need to be positive definite. See also [a] for details.
>
> References:
>
> [a] Wikipedia contributors. "Pseudo-Euclidean Space." Wikipedia, The Free Encyclopedia. Accessed August 6, 2024.

---

> > ### Comment · Reviewer_uqTT · 2024-08-12
> > **Post author rebuttal comments**
> >
> > I thank the authors for their responses. They do help clarify the specific points I raised. I'd be willing to slightly increase my score. However, my main criticism, that the paper contains a number of disparate ideas which may as well be split, still stands -- to be fair, it may not be easy to address that with a minor revision.

---

> > > ### Author Response · Authors · 2024-08-13
> > > **Clarification on how the different ideas are tied together**
> > >
> > > Dear Reviewer uqTT,
> > >
> > > We are grateful for your response and for giving us another opportunity to explain why we believe that the different ideas or components in this paper really belong together. To show the efficacy of our meta-algorithm, we need to have a candidate pool of contrast functions whose weaknesses complement each other. For example, kurtosis is a widely used cumulant-based method for noisy ICA that has been previously shown to suffer in the presence of heavy-tailed distributions or distributions with small kurtosis. So, we also need to have contrast functions that are not cumulant-based and are less sensitive to heavy-tailed source signals. While there are many such contrast functions for noiseless ICA, there are not many with provable guarantees for noisy ICA. This is why we designed our CHF-based contrast function and developed a theoretical framework for analysis under the noisy ICA model. We explained this between lines 87-110. We will be happy to add a longer discussion about this at the beginning of the paper to better address your remarks.

---

### Official Review · Reviewer_c27m · 2024-07-07

**Soundness:** 3
**Presentation:** 3
**Contribution:** 3
**Rating:** 6
**Confidence:** 4

**Summary:**

This paper proposes a nonparametric score to adaptively pick the best noisy ICA algorithm from a set of candidates. This “independence score” is based on the characteristic function-based objective (CHF) introduced by Eriksson&Koivunen in 2003.

In practice, this independence score evaluates the inverse mixing matrix obtained by an ICA algorithm without requiring accessing to the true sources.

In addition, the paper proposes some new contrast functions and algorithms and present simulation results showing the effectiveness of the proposed independence score.

**Strengths:**

•	Solid theoretical approach that justifies the proposed independence score and the new derived objective functions

•	Good review of relevant previous studies on ICA and noisy ICA.

•	Experimental results include synthetically generated sources aswell as real data (images, MNIST) synthetically mixed.

**Weaknesses:**

•	The proposed approach focused only in the recovery of the proper unmixing matrix (B^{-1}) while in noisy ICA, the final goal is to recover the clean sources. There is no discussion about how to recover the sources after the proper unmixing matrix is obtained.

•	The experimental part focused on synthetically mixed signals only. The paper would be greatly benefited from the inclusion of some real-world source separation problem.

•	Theorem 2 assumes Subgaussian mixtures, which ensures the concentration of the sample covariance matrix. This assumtion is rather strong and it would be good to analyze what happen if it is not met.

**Questions:**

Regarding assumption of Theorem 2 (sub-Gaussianity). I understand that assumption is needed to guaranty the concentration of the norm operator, which is a rather strong assumption. Do you think is possible to relax that assumption and keep the method working? Have you make any experiments with super-Gaussian sources, for example?

**Limitations:**

Yes, the experimental results section discuss some limitations of the proposed algorithm in highly noisy scenarios.

---

> ### Author Rebuttal · Authors · 2024-08-07
>
> Thank you for your kind words regarding our theoretical approach, literature review, and experiments. We address your comments, suggestions, and questions below:
>
> **[Re: Recovery of source signals]** We address this concern in lines 60-65 of our paper. Under the noisy ICA model (Eq 1), both $\mathbf{x} = B(\mathbf{z}+\mathbf{g’}) + \mathbf{g}$ and $\mathbf{x} = B\mathbf{z} + (B\mathbf{g’} + \mathbf{g})$ for Gaussian $\mathbf{g}’$ are indistinguishable (see Voss et al (2015) pg 3). This makes recovery of the source signals impossible in the noisy ICA setting, even if the mixing matrix is known exactly because part of the noise could always be interpreted as part of the signal while preserving the model definition.  Voss et al. (2015) propose learning the source signals by maximizing the Signal-to-Interference-plus-Noise ratio (SINR). Typically, in all related papers, algorithms are compared using the accuracy of estimating the $B$ matrix. But, with the estimated $B$ matrix one can obtain an SINR optimal estimation of the signals as in the previous citation, which we will clarify.
>
> **[Re: Experiments with real-world source separation data]** Like many previous methodological papers, we focused on synthetically mixed real data. Having access to the ground truth signals and mixing matrix allows us to better understand the quality of the solutions of the algorithms. For example, many of the available sound-based (see e.g. [a]) datasets mix different sound sources synthetically.
>
> **[Re: Subgaussianity assumption in Theorem 2]:** Yes, as long as the sample covariance matrix concentrates around the population covariance in the operator norm, our proof holds. For example, [b] shows that if $\mathbb{E}[|X^T u|^q]\leq L^q$ is bounded for all unit vectors $u \in \mathbb{R}^{d}$ (Eq 1.6), then the covariance matrix concentrates at a rate $(\frac{d}{n})^{\frac{1}{2}-\frac{1}{q}}$. Under such distributions, which are more general than sub-gaussians, Theorem 2 will hold with a different error rate. We will clarify this. Our experiments include exponential source signals, which do not follow the subgaussian assumption. We have added new experimental results (see pdf) with other super-gaussian sources (Laplace, Student’s t with 3, 5 degrees of freedom, and exponential) to answer your question. The Meta algorithm closely follows the best candidate algorithm even in the presence of many super-gaussian signals, which reflects that the sub-gaussianity assumption in Theorem 2 is not critical.
>
> References:
>
> [a] Massachusetts Institute of Technology. "Independent Component Analysis (ICA) Benchmark." MIT Media Lab. Accessed August 6, 2024.
>
> [b] Vershynin, Roman. "How close is the sample covariance matrix to the actual covariance matrix?." Journal of Theoretical Probability 25, no. 3 (2012): 655-686.

---

> > ### Comment · Reviewer_c27m · 2024-08-10
> > **Thanks**
> >
> > I appreciate the authors' responses to my comments. However, despite the weaknesses not being serious, they still persist, so I will maintain my original score.

---

> > > ### Author Response · Authors · 2024-08-12
> > >
> > > Dear reviewer c27m,
> > >
> > > Thank you - we are grateful for your review and response.
> > >
> > > We just wanted to clarify that the first weakness (re: recovery of source signals) is not exactly a weakness because it is a property of the noisy ICA model. We addressed this in lines 60-65 of our paper, and we are happy to add a more detailed discussion. Similarly, the third point (re: sub-gaussianity assumption in theorem 2) can be replaced with a much weaker condition on finite moments (as pointed out in our rebuttal).

---

### Official Review · Reviewer_N8km · 2024-07-12

**Soundness:** 3
**Presentation:** 3
**Contribution:** 2
**Rating:** 6
**Confidence:** 3

**Summary:**

The presented paper focuses on the problem of noisy ICA which remains a significant challenge in classical machine learning.
The authors introduces a nonparametric independent score extending the work in [21] to evaluate the estimation of the demixing matrices without requiring any prior knowledge of underlying noise distribution parameters.
Further. the authors propose some new contrast functions and provides a very interesting discussion on the convergence for the presented contrast functions of noisy ICA which can be also applied for cumulant based contrast functions.

**Strengths:**

The introduction of a nonparametric independence score is innovative.
The paper provides an extensive theoretical analysis, including the development of new contrast functions and a detailed discussion on convergence properties.

**Weaknesses:**

Two main weaknesses stood out to me in the paper:
1.  The proposed study heavily rely on Gaussian noise assumptions. The performance of suggested score might not hold in cases with different noise characteristics. Similarly, the proposed contrast functions could have limited scope of applicability in real life scenarios..
2. The meta algorithm's effectiveness is contingent on the pool of candidate algorithms it selects from.

**Questions:**

In equation 2 in the exponent of the product term \mathbf{t} seems to be missing?

**Limitations:**

The methods focus primarily on Gaussian noise, which might limit their applicability in scenarios where noise distributions do not conform to this assumption.
The global convergence analysis heavily depends on the assumption that the noise is Gaussian. This dependency raises questions about the generalizability of the convergence results to other types of noise.

---

> ### Author Rebuttal · Authors · 2024-08-07
>
> Thank you for your kind words regarding our independence score, contrast functions, and our theoretical analysis. We address your questions and suggestions below.
>
> **[Re: Gaussian noise assumption]** The classical noisy ICA problem adds Gaussian noise to a mixture of non-gaussian independent components. This is an illustrious problem with applications ranging from signal processing and image analysis to biomedical data analysis. Nearly all the literature we found focuses on Gaussian noise. While it will be interesting to analyze the role of non-gaussian noise, that is outside the scope of this paper.
>
> **[Re: effectiveness of Meta]** You are right. We hope that given the multitude of ICA methods we can include in the candidate pool, one can use this property as a strength of our algorithm.
>
> **[Re: $\mathbf{t}$ missing]**: Yes, that is a typographical error. We will fix it in the revised manuscript.

---

### Official Review · Reviewer_Dz1S · 2024-07-15

**Soundness:** 4
**Presentation:** 2
**Contribution:** 3
**Rating:** 6
**Confidence:** 4

**Summary:**

This work proposes a modification of the original CHFICA characteristic function to the noisy ICA case without requiring knowledge of the noise distribution parameters.
The modified independence score is then used to select the best out of multiple ICA methods based on the score assigned to their solutions, in a per dataset basis. In practice, an average score is evaluated over random directions t, which renders the score computationally inefficient for direct Minimization.
The empirical estimate of the new independence score is shown to converge to the population score if the data covariance is concentrated.
Empirical evidence is shown that the new empirical score correlates well with the well-established Amari error.

This work also introduces two computationally efficient contrast functions to be Maximized (CHF-based and CGF-based) for BSS, both of which do not require higher moments and, thus, work well on data with near-zero kurtosis, like Bernoulli(p) with low p.
Theoretical analyses of local and global convergence are also included for a large class of smooth contrast functions that meet certain desirable properties (Assumption 1). Properties (a) and (c) are critical as they yield contrast functions that are not influenced by additive independent Gaussian noise.
The CHF contrast only requires a finite second moment and is suitable for heavy-tailed sources.
The CGF contrast is not appropriate for heavy-tailed sources.
Global convergence: The Hessian of both contrasts is shown to have the form C = BDB^T. The maxima of the Hessian-adjusted contrasts is shown to occur at B^-1 u = e_i for positive elements of D. Fast sequential power method optimization in a pseudo-Euclidean space defined by C^-1 was shown by [48]. Before optimization, C need only be computed once for some random vector u and just reused during the power iteration updates. A requirement of no sign change of the third derivative of the contrast over the half-lines is required to sufficiently distinguish between Gaussian and near-0 kurtosis source distributions.
Local convergence: Linear convergence is attainable without the third derivative requirement. Geometric convergence is attained at some low epsilon based on how non-gaussian the source is.
Contrary to prior work on cumulant-based methods, convergence of contrasts meeting Assumption 1 was established based on the characterization of the contrasts in the pseudo-Euclidean space and not on the convergence of the power method.

Experimental results support Meta's ability to select the optimal solution over various algorithms, showing variance reduction in the estimates, uniform performance across high and near-0 kurtosis.
Evaluation of the CHF and CGF contrasts show that CHF outperforms other methods at higher noise, but only at sample size n > 20000.
The Meta algorithm picks up the slack of CHF by selecting the competing PFICA solution when n < 20000.

**Strengths:**

Originality :
- The work introduces a somewhat novel combination of known ideas in a reasonably novel formulation that adapts previous work to the case of additive Gaussian noise.
- The work differs from and extends previous contributions, dealing with near-zero kurtosis sources and additive Gaussian noise.
- The manuscript cites related work on contrast functions for BSS and noiseless ICA methods, adequately indicating the sources of inspiration.

Quality:
 - The work is technically sound, with thorough proofs.
 - The theoretical claims are well supported by the experiments.
 - Strengths and weaknesses are discussed well.

Clarity:
 - The work is written well overall, focusing on the key points and contributions.

Significance :
 - The results are important as they establish a consistent framework to select the optimal solution over a range of BSS methods for noisy ICA based on the proposed modified independence score.

**Weaknesses:**

Quality:
 - The code is not friendly to readers as there is no 1-to-1 correspondence with the notation in the paper. Suggestion: improve documentation and tidy up the codebase (especially comments and function arguments) for readability.

Clarity:
 - Despite the well-written manuscript and thorough methodology and proofs, the paper contains many typos (both in text and proofs) and inconsistencies in notation (especially between Appendix and main manuscript) that limit the clarity. Besides fixing these typos, explicitly include some of the "obvious" steps currently omitted in the derivation; that would greatly improve readability. Also add some high-level intuitions as noted below.
 - Clarify the origin of new contrast functions and how they came about. It is unclear what motivated or led to these otherwise "semi-arbitrary" functions.
 - Define O_P(.)
 - It appears that the new independence score does not meet some of the properties in Assumption 1. Please, clarify if that is indeed the case and, if so, which properties specifically.
 - Appendix A.3: Clarify that Algo 2 is NOT specific to contrasts based on power iteration only, but any sequential method. It is suggested to make one separate sub-section covering how the projection is done and another for the Algo description itself. Also clarify what the "Total" (non-sequential) Algo would look like, for comparison.
 - Lines 119-122: t.T should be bold face.
 - Eq. 2: missing last t in the PRODUCT term.
 - Section 3.2: Change "Properties of Contrast Functions 1." to "Assumption 1. (Properties of Contrast Functions)"
 - Line 175: remove "them"
 - Line 182: "note" --> "not"
 - Footnote 2 (page 5): add "and remains a linear combination of z"
 - Theorem 4, line 254: Assumption 1(d) is NOT about the third derivative.
 - Line 271: Define the value of k in this experiment. I see 11 mentioned in passing in line 786...
 - Line 277: Fix contradicting statements: "CHF and CGF are initialized using the B estimated via PFICA" and "CHF and CGF is based on a single random initialization".
 - Line 283: "also used" --> "also be used"
 - Table 1 caption: Clarify that since median is being reported, Meta is better than the best method “on average”, but on any individual experiment it is identical to the best method.
 - Line 299: refs to 2b and 2c are in flipped order.
 - Line 315: In the noise power experiment, it is stated that "the difference between the two leading algorithms is small." Therefore, the same acknowledgement should be indicated in line 315 after "CHF dominates, and one can see that Meta starts following CHF"
 - Line 456: Assumption 1(d) is NOT about the 3rd derivative constraint...
 - Line 464: Missing ' in g.
 - Line 472: For consistent notation, it should be Delta(t,F | P) = 0, not Delta_F(t) = 0
 - Lines 474-477: there is an extra transpose symbol at the last t (several times).
 - Line 510: (e) --> (d)
 - Notation and conditions in lines 524-529 is not consistent with Theorem 3 in the main manuscript.
 - Line 777: w_i is not defined, g_i should be ~g_i.
 - Lines 784-789: the r defined here has no relation to the r in Algo 2. Please pick another letter.
 - Algo 2:
	 - Clarify why it is necessary to evaluate Y_j per component and why it would not be equivalent to just compute W = X V^T U^T r
	 - for j in range[1, M] : M is not defined, but I think it would be l + 1? Also, the letter j is used for many different things inside the same loop. Change to " for p in range[1, l + 1] ", and s(j) --> s(p).
	 - " W(:,j) ← Y_j r " --> " W(:,a) ← Y_a r "
	 - I suspect r is equivalent to t in Eq. 2, if so change it to t.
 - Appendix A.4: Verify that all occurrences of Assumption 1 (d) are correct (i.e., referring to symmetry not the third derivative)

**Questions:**

Questions were included along with the weaknesses.

**Limitations:**

Yes, the authors addressed the limitations of the work.

---

> ### Author Rebuttal · Authors · 2024-08-07
>
> Thank you for your kind words regarding our work's originality, quality, clarity, and significance. We appreciate your detailed comments and suggestions, which we address below. All typographical and grammatical errors will be corrected in the revised manuscript and are not addressed individually here.
>
> **[Re: Code]** We will reorganize the codebase to enhance readability for the revised submission
>
> **[Re: Clarity]** In the revised version of the manuscript, we will fix typographical and notation errors, and add further details in the derivations to enhance readability and clarity
>
> **[Re: $O_P$]** $O_P$ refers to the Big-Oh notation in probability. We will include a formal definition in the revised manuscript
>
> **[Re: Occurrences of Assumption 1 (d)]** We will double-check and ensure that occurrences of Assumption 1(d) refer to the symmetry of source distributions, not the third-derivative condition
>
> **[Re: Properties of Independence Score]** The new independence score does not satisfy property (a) of Assumption 1, so we do not consider it as an optimization objective but rather treat it as a diagnostic score for the Meta algorithm
>
> **[Re: Appendix A.3]** We will reorganize the section to improve the clarity of Algorithm 2 and add further details about the total algorithm that would look like which uses this to choose the best candidate. Thank you for pointing out the wider applicability of the algorithm apart from power-method-based iterative approaches.
>
> **[Re: Y_j per component]** Thank you for pointing it out. It is indeed true that the evaluation of Y_j can be done as suggested.
>
> **[Re: Choice of k for experiments]** The value of k depends on the particular experiment and is specified in lines 289 and 297.

---

> > ### Comment · Reviewer_Dz1S · 2024-08-13
> >
> > Thank you for your response.
> > I have some follow up questions:
> >
> > 1. How is O_P different from just O?
> >
> > 2. Include a note that "The new independence score does not satisfy property (a) of Assumption 1"
> >
> > 3. Clarify which one is the 11-dimensional dataset mentioned in line 786... (line 297 says k=9).
> >
> > 4. Line 277: So which is the correct statement: "CHF and CGF are initialized using the B estimated via PFICA" or "CHF and CGF is based on a single random initialization"?
> >
> > 5. Origin of new contrast functions: How do you arrive at Eq. (4) from the CHF? Likewise, how do you arrive at Eq. (4) from the CGF? Although the conditions in Assumption 1 are satisfied by the new contrast functions, the conditions do not specify what f() should be. Currently the train of thought/rationale/derivation from CHF -> f() is missing (likewise from CGF -> f() ). Assumption 1 is insufficient to clarify this point.

---

> > > ### Author Response · Authors · 2024-08-14
> > >
> > > Thank you for your questions and for giving us an opportunity to further clarify. We address them below:
> > >
> > > **Regarding $O_P$ and $O$**: The $X_n=O_P(a_n)$ implies that $X_n/a_n$ is stochastically bounded (or more technically uniformly tight), i.e. $\forall \epsilon>0$, there exists a finite $M$ such that $\sup_n P(|X_n/a_n|>M)\leq \epsilon$ (See [A]). We will clarify this.
> > >
> > > **Regarding Independence Score and Property (a)**: We will do so.
> > >
> > > **Regarding dimension of dataset**: It should be the 9-dimensional dataset. We will fix this typographical error.
> > >
> > > **Regarding random initialization of CGF and CHF**: Both are correct. Recall that, for the power iteration-based algorithm of Voss et. al (2015) we need both a quasi-orthogonalization matrix which is an estimator of a matrix of the form $BDB^T$, and a unit vector $u$. The quasi-orthogonalization matrices for CHF and CGF use the $B$ matrix estimated via PFICA. But the initial vector $u$ for the power iteration is a random unit vector. We will clarify this further in the revised manuscript.
> > >
> > > **Regarding the origin of new contrast functions**: We started from the fact that, like the cumulants, the cumulant generating function (and similarly the CHF-based counterpart) satisfies Assumption 1a). In order to satisfy 1c), we wish to subtract out the part resulting from the Gaussian noise, which led to the additional terms involving $u^{\top}Su$. We will add this explanation to better address your remark.
> > >
> > > References :
> > > [A] Vaart, A. W. van der. 1998. Asymptotic Statistics. Cambridge Series in Statistical and Probabilistic Mathematics. Cambridge University Press.

---

### Author Rebuttal · Authors · 2024-08-06

We want to first thank all the reviewers for their valuable suggestions and insightful feedback. We believe we have addressed nearly all of their main technical questions. In what follows, we will address some important points each reviewer has raised. We will correct all the typographical issues pointed out and will not address them here.

### **Re: Clarity (Reviewer Dz1S)**

 Thank you for noting that our manuscript, proofs, and methodology are well-written and thorough. In the revised version of the manuscript, we will fix typographical and notational errors and add further details in the derivations to enhance readability and clarity.

### **Re: Gaussian noise assumption (Reviewer N8km)**

 The classical noisy ICA problem adds Gaussian noise to a mixture of non-gaussian independent components. This illustrious problem has applications ranging from signal processing and image analysis to biomedical data analysis. Nearly all the literature we found focuses on Gaussian noise. While it will be interesting to analyze the role of non-gaussian noise, that is outside the scope of this paper.

### **Re: Recovery of source signals (Reviewer c27m)**

 We address this concern in lines 60-65 of our paper. Under the noisy ICA model (Eq 1), both $\mathbf{x} = B(\mathbf{z}+\mathbf{g’}) + \mathbf{g}$ and $\mathbf{x} = B\mathbf{z} + (B\mathbf{g’} + \mathbf{g})$ for Gaussian $\mathbf{g}’$ are indistinguishable (see Voss et al (2015) pg 3). This makes recovery of the source signals impossible in the noisy ICA setting, even if the mixing matrix is known exactly because part of the noise could always be interpreted as part of the signal while preserving the model definition.  Voss et al. (2015) propose learning the source signals by maximizing the Signal-to-Interference-plus-Noise ratio (SINR). Typically, in all related papers, algorithms are compared using the accuracy of estimating the $B$ matrix. But, with the estimated $B$ matrix one can obtain an SINR optimal estimation of the signals as in the previous citation, which we will clarify.


### **Re: Subgaussianity assumption in Theorem 2] (Reviewer c27m)**

Yes, as long as the sample covariance matrix concentrates around the population covariance in operator norm, our proof holds. For example, [a] shows that if $\mathbb{E}[|X^T u|^q]\leq L^q$ is bounded for all unit vectors $u \in \mathbb{R}^{d}$ (Eq 1.6), then the covariance matrix concentrates at a rate $(\frac{d}{n})^{\frac{1}{2}-\frac{1}{q}}$. Under such distributions, which are more general than sub-gaussians, Theorem 2 will hold with a different error rate. We will clarify this. Our experiments include exponential source signals, which do not follow the subgaussian assumption. To answer your question, we have now added new experimental results (see pdf) with other super-gaussian sources (Laplace, Student’s t with 3, 5 degrees of freedom, and exponential). The Meta algorithm closely follows the best candidate algorithm even in the presence of many super-gaussian signals, which reflects that the sub-gaussianity assumption in Theorem 2 is not critical.

### **Re: Connection between the meta-algorithm and the new contrast functions] (Reviewer uqTT)**

Most methods for noisy ICA center around designing an appropriate contrast function. Lines 87-103 in the paper, under “Individual shortcomings of different contrast functions and fitting strategies,” explain the different issues with different types of existing contrast functions. As explained in lines 104-110, this leads to two challenges: a) how do we adaptively pick an appropriate contrast function for the dataset at hand, and b) can we design better contrast functions? Our paper gives a full pipeline for a new adaptive algorithm for choosing from a set of contrast functions, which includes established ones and the new ones we introduce.

### **Re: Development of the meta-algorithm and the choice of randomization in its computation] (Reviewer uqTT)**

 Lines 115-124 build intuition for the score used in the Meta algorithm using the noiseless case. For the noisy case, one needs to account for the characteristic function of the Gaussian noise, whose covariance matrix is unknown. The second terms in JOINT and PRODUCT essentially cancel out the additional terms using \textit{estimable} parameters, i.e. the covariance matrix $S$ of the data (estimated using the sample covariance).

Lines 148-151 discuss why we do not use the independence score as an optimization objective and instead use it to choose the best algorithm amongst candidates. This leads to the meta-algorithm in Algorithm 1.

Remark 2 explains the choice of randomization in the Meta algorithm. Essentially, all related papers that use the characteristic functions in ICA or for Gaussianity testing (see the citations in Remark 2, line 155) average over $t$ sampled from a spherical Gaussian.

### **Re: Origin and development of contrast functions (Reviewer Dz1S)**

  Section 3.2 (lines 163-168) provides the mathematical properties that are useful in a contrast function for designing provable optimization algorithms for ICA. Lines 169-172 explain what each of these properties imply. Section 3.3 then develops the new contrast functions that satisfy these important properties.

References:

[a] Vershynin, Roman. "How close is the sample covariance matrix to the actual covariance matrix?." Journal of Theoretical Probability 25, no. 3 (2012): 655-686.

---

### Decision · Program_Chairs · 2024-09-25

**Decision:**

Accept (poster)

**Comment:**

This paper presents a modification of the noiseless characteristic function-based ICA to the noisy ICA without requiring the knowledge of the Gaussian noise distribution parameters. Two computationally efficient contrast functions are introduced in the paper. Reviewers claimed that the origin of these new contrast functions was not clear, but during the reviewer-author discussion period it has been clarified up to a certain point. One of serious concerns is that the paper seems to a collection of not clearly-developed ideas, rather than exploring coherently and clearly a single idea. I  believe this is partially true. During the AC-reviewer discussion period, there was a mix of support and concern. One reviewer with negative comments agreed that he/she would not mind accepting the paper even though he/she wanted to keep his/her rating as a minority. Consider all the comments raised by reviewers, I think that paper has a contribution in the direction of noisy ICA. I would like to ask the authors to revise the paper carefully, improving the clarity in the final version.